# MedChain: Bridging the Gap Between LLM Agents and Clinical Practice with Interactive Sequence

**Jie Liu**[1], **Wenxuan Wang**[2], **Zizhan Ma**[3], **Guolin Huang**[4], **Yihang Su**[3],
**Kao-Jung Chang**[6,7], **Haoliang Li**[1], **Linlin Shen**[4,5], **Michael Lyu**[3], **Wenting Chen**[1*]

[1]The City University of Hong Kong  [2]Renmin University of China
[3]The Chinese University of Hong Kong  [4]Shenzhen University
[5] Guangdong Provincial Key Laboratory of Intelligent Information Processing
[6]National Yang Ming Chiao Tung University [7]Taipei Veterans General Hospital
Project: https://github.com/ljwztc/MedChain

## Abstract

Clinical decision making (CDM) is a complex, dynamic process crucial to healthcare delivery, yet it remains a significant challenge for artificial intelligence systems. While Large Language Model (LLM)-based agents have been tested on general medical knowledge using licensing exams and knowledge question-answering tasks, their performance in the CDM in real-world scenarios is limited due to the lack of comprehensive benchmark that mirror actual medical practice. To address this gap, we present MedChain, a dataset of 12,163 clinical cases that covers five key stages of clinical workflow. MedChain distinguishes itself from existing benchmarks with three key features of real-world clinical practice: personalization, interactivity, and sequentiality. Further, to tackle real-world CDM challenges, we also propose MedChain-Agent, an AI system that integrates a feedback mechanism and a MedCase-RAG module to learn from previous cases and adapt its responses. MedChain-Agent demonstrates remarkable adaptability in gathering information dynamically and handling sequential clinical tasks, significantly outperforming existing approaches.

## 1 Introduction

At the intersection of artificial intelligence and healthcare lies one of medicine's most complex challenges: Clinical Decision Making (CDM). In healthcare delivery, CDM demands not only the integration of diverse data sources and continuous assessment of evolving clinical scenarios, but also evidence-based judgments for diagnosis and treatment [46]. While crucial for optimal patient care, this intricate process imposes significant cognitive demands on healthcare professionals, making it an ideal candidate for AI assistance [42].

Recent advances in Large Language Model (LLM)-based agents [34, 49, 15, 44, 16, 71] have emerged as an effective solution for complex decision-making tasks, from software development [38] to office automation [57]. In the medical domain, these LLMs have demonstrated impressive performance on medical licensing exams [45, 36] and knowledge-based assessments [13, 10, 30]. While LLMs have consistently scored well above passing thresholds in these benchmark [45], it is crucial to recognize that these assessments fall short of capturing the complexity of real-world CDM, where errors can cascade through multiple decision stages, as illustrated in Figure 1. Based on our analysis, CDM exhibits three key characteristics.

---

[*]Wenting Chen is the corresponding author.

39th Conference on Neural Information Processing Systems (NeurIPS 2025) Track on Datasets and Benchmarks.

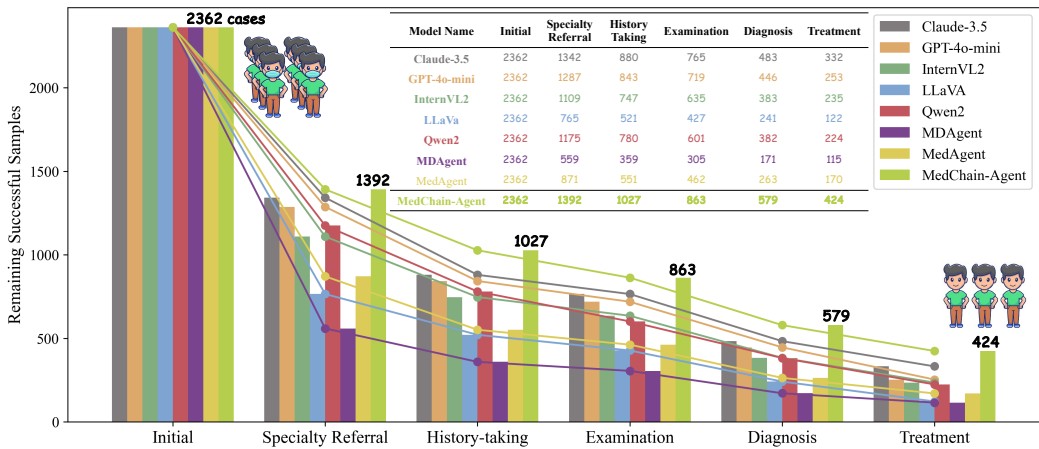

Figure 1: **Demonstration of error propagation of CDM in MedChain.** Starting with 2,362 initial cases, the diagram illustrates how diagnostic errors cascade through five clinical stages. Cases with incorrect diagnoses carry forward problematic information to subsequent stages, leading to a cumulative decrease in accuracy. After completing the treatment phase, we count cases that maintain correctness through each consecutive phase up to the each stage. Our MedChain-Agent achieves best performance in CDM comparing with other SOTA methods.

**Firstly**, these benchmarks rarely account for patient-specific information such as past medical history and present illness [36], which significantly influence clinical decisions in real clinical scenarios. This omission fails to capture the nuanced context that often shapes *personalized* diagnosis. **Secondly**, unlike real clinical scenarios where decisions build upon previous steps, existing benchmarks present clinical tasks as independent problems [40], missing the critical interdependencies in the diagnostic process. In reality, clinical decision-making is a *sequential* process where each step is contingent upon the preceding ones, and an error in one stage can profoundly impact subsequent decisions. **Thirdly**, most benchmarks present all relevant information upfront, providing a static, and comprehensive dataset [51]. However, real clinical workflow demand multiple rounds of dynamic information gathering through ongoing patient *interaction*.

**MedChain:** To address these critical gaps, we introduce MedChain, a novel benchmark designed to evaluate LLM-based agents in real-world clinical scenarios. Specifically, MedChain comprises 12,163 diverse cases spanning 19 medical specialties and 156 sub-categories, including 7,338 medical images with corresponding reports. Each case progresses through five crucial stages: specialty referral, history-taking, examination, diagnosis, and treatment. Unlike existing benchmarks, MedChain uniquely emphasizes three key features. 1) *Personalization:* Each case incorporates detailed patient-specific information. At first, agents are provided with only the patient's chief complaint and basic information. 2) *Interactivity:* Information must be actively gathered through dynamic consultation from patient. 3) *Sequentiality:* Decisions at each stage influence subsequent steps. Only after agent sequentially completes all five stages, the overall diagnostic process is evaluated.

**MedChain-Agent:** Given the novel features and challenges presented by this benchmark, existing agent frameworks struggle to address these issues adequately. To overcome these limitations, we propose MedChain-Agent, a multi-agent collaborative framework that enables LLMs with feedback mechanism and MedCase-RAG to dynamically gather information and handle sequential clinical tasks. Specifically, MedChain-Agent facilitates a synergistic interplay among three specialized agent types: General Agents for task-specific expertise, a Summarizing Agent for insight synthesis, and a Feedback Agent for iterative refinement. This multi-layered, iterative approach ensures decisions are products of thorough analysis and diverse perspectives. Additionally, to address the multifaceted nature of CDM, which demands the integration of evidence-based research, and patient-specific factors, we incorporate a novel MedCase-RAG module into our MedChain-Agent framework. Unlike conventional medical RAG methods, MedCase-RAG dynamically expand its database and employs a structured approach to data representation, mapping each medical case into a 12-dimensional feature vector. This system enables efficient retrieval of relevant cases and helps the model make informed decisions.

Our contributions are summarized as follows:

- We represent the first effort to propose a CDM benchmark, MedChain, providing a holistic assessment of diagnostic capabilities of LLLM-based agents, closely reflecting real-world patient care.

- We propose a multi-agent framework based on the characteristics of CDM, called MedChain-Agent. This system enables efficient retrieval of relevant cases and helps the model make informed decisions.

- Through extensive experiments, we compare the performance of existing works on Med-Chain and the superiority of MedChain-Agent in CDM and realibility.

## 2 Related Works

### 2.1 Evaluation of LLM in Medcine

Benchmarking plays a vital role as a key performance indicator, directing model improvements, pinpointing weaknesses, and shaping the course of model evolution. The evaluation of LLMs in medicine has primarily focused on testing general medical knowledge through structured assessments [33]. Leading benchmarks such as MultiMedQA [45] integrate various medical QA datasets (e.g., MedQA [19], MedMCQA [36]), emphasizing performance on medical licensing examination materials. Other benchmarks like PubMedQA [20] focus on research-oriented queries, while several Chinese medical benchmarks [55, 2] evaluate models through multiple-choice questions from medical licensing exams. While these benchmarks effectively assess general medical knowledge, they fail to capture three critical aspects of real-world clinical decision-making (see Appendix Table 6), i.e., personalization in patient care, the interactive nature of clinical consultations, and the sequential dependency of medical decisions, where each step builds upon previous findings.

Recently, several benchmarks [59, 35, 28, 26, 24, 8] have been proposed to evaluate LLMs across diverse clinical scenarios and tasks, including information extraction, text summarization, and clinical outcome prediction. However, these benchmarks primarily consist of independent question-answer pairs, where each task is evaluated in isolation. Sequential decision-making is critical in medical practice, as each patient experiences a continuous journey from initial triage through treatment to recovery. Therefore, it is essential to evaluate how LLMs perform throughout this entire clinical pipeline. Our work distinguishes itself from existing benchmarks by focusing on sequential decision-making within interactive environments, specifically evaluating LLM performance in realistic clinical scenarios that require executing the complete patient care workflow.

### 2.2 LLM-based Agent in Medicine

LLM-based agents have demonstrated significant potential across various medical applications [53, 32], encompassing tasks such as medical examination questions, clinical diagnoses, and treatment plans. Recent research has explored different approaches: Agent Hospital [25] provides medical scenario simulation, while several frameworks [66, 47, 22] focus on specific medical stages with multi-agent architectures. Some works target specialized aspects, such as CoD [4] for interpretable diagnostics and Ehragent [43] for electronic health records (EHRs) analysis. Others, like Almanac Copilot [67], assist clinicians with EMR-specific tasks. AI Hospital [11] explores interactive clinical scenarios, but it falls short in handling multi-modal medical imaging and lacks a comprehensive benchmark for evaluating multi-agent performance. To enhance these agents' capabilities, researchers have integrated Retrieval-Augmented Generation (RAG), as demonstrated by MIRAGE [64]'s search-enhanced framework and Medical Graph RAG [61]'s knowledge-based approach.

However, current approaches face two major limitations. First, existing frameworks focus on isolated medical tasks rather than providing seamless integration across different clinical stages, making them insufficient for complex scenarios requiring effective inter-stage communication [14]. Second, current medical RAG systems' reliance on chunk-based indexing leads to context inconsistency and computational inefficiencies [18, 12], highlighting the need for more sophisticated approaches to medical knowledge integration.

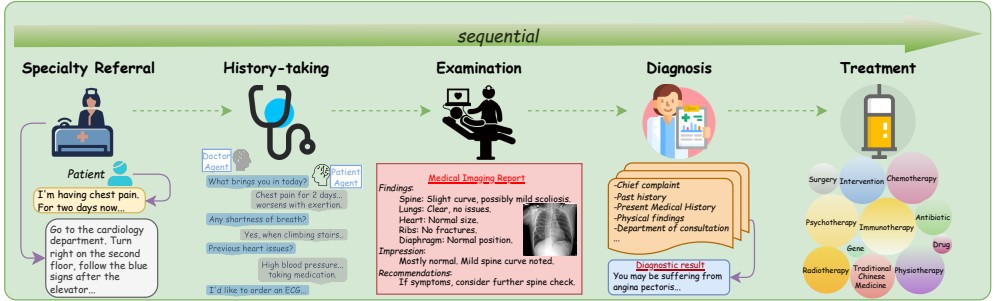

Figure 2: **MedChain Pipeline.** The MedChain is composed of a sequential medical process, including specialty referral, history-taking , examination, diagnosis, and treatment.

## 3 MedChain Benchmark

**Overview.** We introduce MedChain, a comprehensive clinical decision-making benchmark designed to simulate real-world scenarios. Built upon 12,163 Electronic Health Records (EHRs) spanning 19 medical specialties and 156 sub-categories, including 7,338 medical images with reports, MedChain uniquely emphasizes three key characteristics:

- *Personalization*: Each case includes detailed patient profiles that influence decision-making
- *Sequentiality*: Cases involve multiple interconnected decision-making stages
- *Interactivity*: Information must be actively gathered through dynamic consultation

### 3.1 Data Collection

**Data Source and Processing.** Our dataset is sourced from the Chinese medical website "iiYi" [2], which provides over 20,000 validated clinical cases spanning 28 disease categories. These cases are verified by professional doctors and have undergone de-identification to ensure patient privacy. We obtained formal permission from the website administrators to use the data for scientific research purposes. Each case typically contains the patient's chief complaint, medical history, examination results, treatment process, and other relevant information, which insure the **personalization** of MedChain. Following the government standards [3] and Medical Subject Headings [4], we extracted and organized key information including patient basics, chief complaints, specialty referrals, examinations, imaging reports, diagnoses, and treatments. Cases with incomplete information were removed, resulting in 12,163 high-quality cases.

**Quality Control.** To ensure the highest standards of data integrity and clinical relevance in our benchmark, we implemented a rigorous quality control process involving a panel of five senior physicians, each with over 10 years of clinical experience. Our evaluation process examined a random sample of 6,000 cases (49.3% of the dataset). We developed a standardized scoring system that presents physicians with comprehensive case information alongside six binary quality dimensions: disease prevalence, clinical relevance, accuracy of patient history, appropriateness of diagnostic procedures, correctness of diagnosis, and suitability of treatment recommendations. Physicians evaluate each dimension through yes/no responses, with cases satisfying all dimensions considered valid.

The quality assessment yielded strong results, with 94.7% of evaluated cases meeting or exceeding our quality thresholds. Dimension-specific quality rates ranged from 92.9% to 97.2%, demonstrating consistently high standards across all evaluation criteria. Inter-rater reliability analysis produced a Cohen's kappa coefficient of 0.82, indicating substantial agreement among our expert reviewers. Cases that failed to meet the thresholds (5.3%) were either revised or excluded from the final dataset to maintain benchmark integrity.

---

[2]https://www.iiyi.com

[3]National Standards for Medical Items

[4]https://www.ncbi.nlm.nih.gov/mesh/1000048

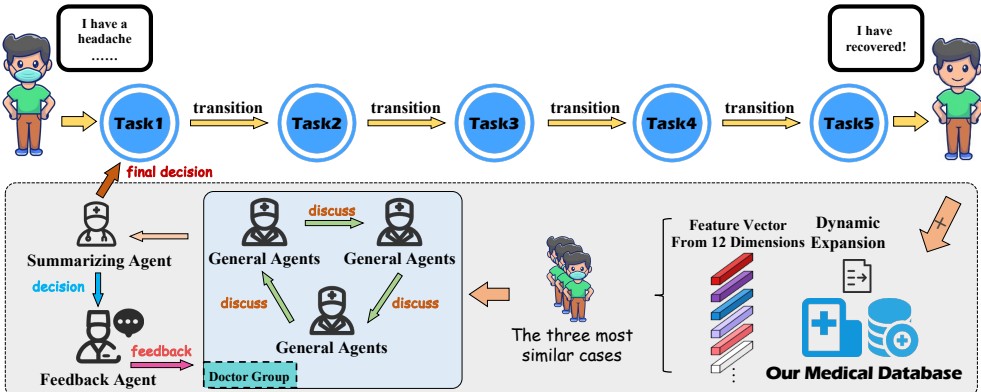

Figure 3: **MedChain-Agent framework.** Depicts a cyclical feedback medical multi-task system, where decisions are supported by retrieving similar past cases from a medical database.

## 3.2 Clinical Workflow Simulation

**Sequential Stages.** MedChain simulates the complete clinical workflow, comprising five sequential tasks, each representing a different stage of the clinical decision-making process, as shown in Figure 2. The results from each stage serve as inputs for the subsequent stage, creating a dependency where later decisions are influenced by the quality of earlier ones. This design guarantees the **sequentiality** of MedChain, mimicking the interconnected nature of real-world clinical decision-making processes. The pipeline consists of: *1) Specialty Referral:* Assessment of case urgency and appropriate department selection; *2) History-taking:* Dynamic information gathering through doctor-patient dialogue; *3) Examination:* Medical image analysis and report generation; *4) Diagnosis:* Comprehensive diagnosis based on accumulated information; *5) Treatment:* Treatment plan formulation considering patient-specific factors.

The construction pipeline of MedChain is sketched out here. The standardization process and the format of each pipeline can be refer to Appendix A. We demonstrate an example in Appendix Figure 4 and Figure 5.

## 3.3 Interaction Environment

To simulate authentic doctor-patient consultations, we developed an **interactive** environment where LLM-based agents must actively gather information through dynamic interactions. We employ Gemma2 (9b) [50] as the patient agent, initializing it with pre-defined case information while withholding the actual diagnosis. Each tested LLM serves as a doctor simulator. This setup enables the agent to provide symptom information and respond to inquiries in a manner that mirrors real patient experiences, drawing inspiration from standardized patients in medical education [1]. We conduct systematic evaluation with senior physician to validate the effectiveness of patient agent. The details can be found in Appendix A.2.2.

## 3.4 Benchmark Evaluation

Given the complexity of medical decision-making, we developed a comprehensive evaluation framework that goes beyond simple binary assessments. The standard answer for each stage comprises multiple key points. The evaluated agent's score increases with the number of standard points addressed, while irrelevant content leads to score reduction. For each task, we employ specific evaluation metrics: Task 1 (Specialty Referral) uses accuracy and Intersection over Union (IoU) to compare predictions with ground truth. Task 2 (History-taking) calculates IoU between predicted and ground truth examination items. Task 3 leverages DocLens [63] for assessing image interpretation quality. Task 4 employs a carefully designed prompt based on National Health Commission of China guidelines to evaluate diagnostic accuracy. Task 5 measures IoU between predicted and ground truth examination items. We elaborate the evaluation metrics in Appendix A.

# 4 MedChain-Agent Framework

## 4.1 Multi-agent System with Feedback

The MedChain-Agent framework introduces a multi-agent system that simulates the complex, interconnected nature of medical decision-making. This system integrates specialized agents that contribute distinct expertise to the diagnostic and treatment process. We sketch the main content here and more details please refer to Appendix B.

### 4.1.1 Agent Roles

Our framework comprises three agent types:

**General Agents:** Task-specific agents recruited based on domain requirements. For specialty referral tasks, these agents possess expertise in medical specialties and triage protocols, enabling effective patient routing. They engage in collaborative discussions that mirror real-world medical consultations.

**Summarizing Agent:** Acts as a central coordinator, consolidating insights from general agents into coherent decisions. This agent synthesizes collective expertise and delivers final recommendations, similar to a senior physician's role in clinical settings.

**Feedback Agent:** Maintains decision quality through continuous evaluation and refinement. This agent assesses outputs, provides targeted feedback, and initiates improvements when necessary, creating a self-correcting decision process.

### 4.1.2 Decision Making with Feedback

The decision-making process begins with general agents analyzing patient data and engaging in structured discussions. The summarizing agent then consolidates these assessments into preliminary decisions, which undergo review by the feedback agent. When issues are identified, the feedback agent initiates an iterative refinement cycle, with general agents reconsidering their assessments and the summarizing agent adjusting decisions accordingly. This process continues until reaching consensus or completing a preset number of iterations.

## 4.2 MedCase-RAG

To enhance decision-making capabilities, we developed MedCase-RAG, a specialized Retrieval-Augmented Generation system for medical applications. Unlike traditional medical RAG systems, our approach employs structured case representation using feature vectors that encode 12 key clinical dimensions: Age, Sex, Chief Complaint, Symptoms, Medical History, Physical Examination, Laboratory Tests, Imaging Reports, Diagnosis, Treatment Plan, Primary Department, and Secondary Department. This feature vectors is extracted by text-embedding-ada-002 from openai. When processing new cases, MedCase-RAG performs similarity searches, identifying the three most similar cases using cosine similarity metrics. This approach provides agents with comprehensive reference points for decision-making. The MedCase-RAG also features dynamic knowledge base expansion, incorporating resolved cases as pseudo-data. This mechanism enables continuous learning from new clinical experiences, adapting to evolving medical knowledge and practices.

# 5 Experiments

## 5.1 Experimental Setup

We split the dataset into training, validation, and testing sets with a ratio of 7:1:2. Our study evaluates both single-agent and multi-agent systems. For base LLM, we test two closed-source models (gpt-4o-mini [34], and claude-3.5-sonnet [49]) and four open-source models (InternVL2-8b [6], llava-llama-3-8b-v1_1 [27], HuaTuoGPT [68], and Qwen2-7B-Instruct [52]), with model weights obtained from official Hugging Face repositories. In the single-agent evaluation, we compare with zero-shot manner, few-shot manner, CoT [58], Self-consistency [56] and RAG [23]. In the multi-agent evaluation, we compare MedChain-Agent against DyLAN [31], AutoGen [62], MedAgent [48] and MDAgent [21]. All agent framework are based on InternVL2-8b [6]. The deployment was conducted using the LMDeploy framework [7]. All tests executed on NVIDIA A100 GPUs featuring

Table 1: **Evaluation of various LLM-based agent in MedChain.** The best performance for each task is highlighted in **bold**.

| Framework | Methods | Specialty Referral | | History-taking | Examination | Diagnosis | Treatment | Average |
|---|---|---|---|---|---|---|---|---|
| | | Level 1 | Level 2 | | | | | |
| Base LLM | GPT-4o-mini [34] | 0.5449 | 0.2871 | 0.3399 | 0.5112 | 0.4177 | 0.3930 | 0.4156 |
| | GPT-o3 | 0.5495 | 0.2866 | 0.3493 | - | 0.4891 | 0.3343 | 0.4017 |
| | Claude-3.5-sonnet [49] | 0.5681 | 0.3050 | 0.3562 | 0.5018 | 0.4207 | 0.4053 | 0.4262 |
| | MedGemma [41] | 0.5063 | 0.1975 | 0.3315 | 0.6324 | 0.4668 | 0.3558 | 0.4105 |
| | LLaVA [27] | 0.3240 | 0.0730 | 0.3182 | 0.5165 | 0.3667 | 0.1060 | 0.2841 |
| | Qwen2 [52] | 0.4975 | 0.2215 | 0.4226 | 0.4829 | 0.4530 | 0.2193 | 0.3828 |
| | InternVL2 [52] | 0.4811 | 0.1935 | 0.4645 | 0.4490 | 0.4367 | 0.2903 | 0.3859 |
| | Baichuan [9] | 0.2959 | 0.0532 | 0.3264 | - | 0.4297 | 0.2591 | 0.2728 |
| | HuaTuoGPT [68] | 0.0707 | 0.0207 | 0.3465 | - | 0.4163 | 0.1780 | 0.2064 |
| | MedReason [60] | 0.4602 | 0.1258 | 0.3315 | - | 0.4741 | 0.2993 | 0.3381 |
| | FineMedLM-o1 [65] | 0.1502 | 0.0124 | 0.3969 | - | 0.2772 | 0.1345 | 0.1942 |
| Single-agent | Zero-shot | 0.4811 | 0.1935 | 0.3450 | 0.4994 | 0.4572 | 0.2817 | 0.3763 |
| | Few-shot | 0.5584 | 0.2481 | 0.4870 | 0.3758 | 0.4382 | 0.3553 | 0.4105 |
| | CoT [58] | 0.5698 | 0.1484 | 0.3750 | 0.6396 | 0.4382 | 0.3421 | 0.4189 |
| | RAG [23] | 0.5927 | 0.2467 | 0.4732 | 0.6408 | 0.4167 | 0.4524 | 0.4704 |
| | Self-consistency [56] | 0.5143 | 0.2319 | 0.4213 | 0.4144 | 0.4198 | 0.3441 | 0.3910 |
| Multi-agent | DyLAN [31] | 0.4415 | 0.1731 | 0.4434 | 0.4980 | 0.3863 | 0.2983 | 0.3734 |
| | AutoGen [62] | 0.5228 | 0.2347 | 0.4528 | 0.4559 | 0.4250 | 0.3590 | 0.4084 |
| | MedAgent [48] | 0.3830 | 0.2039 | 0.4454 | 0.4456 | 0.4102 | 0.3673 | 0.3759 |
| | MDAgent [21] | 0.2398 | 0.1343 | 0.4240 | 0.4983 | 0.3973 | 0.3620 | 0.3426 |
| | MDAgent + RAG [21] | 0.4754 | 0.2063 | 0.4412 | 0.5346 | 0.4198 | 0.4371 | 0.4190 |
| | **MedChain-Agent** | **0.5873** | **0.3505** | **0.5836** | **0.6566** | **0.4807** | **0.4613** | **0.5200** |

Table 2: **Performance Comparison of LLM-based Agents across Other Diagnosis Benchmarks.** Implementation of agent frameworks based on InternVL2, evaluated on multiple medical benchmarks including MedQA [19], PubMedQA [20], PathVQA [17], and MedBullets [3].

| Framework | Method | MedQA | PubMedQA | PathVQA | MedBullets | Average |
|---|---|---|---|---|---|---|
| Single-agent | Zero-shot | 0.426 | 0.668 | 0.449 | 0.490 | 0.508 |
| | Few-shot | **0.477** | 0.648 | 0.448 | **0.503** | 0.519 |
| | CoT | 0.470 | 0.714 | 0.465 | 0.500 | 0.537 |
| | Self-consistency | 0.460 | 0.688 | 0.482 | 0.500 | 0.533 |
| Multi-agent | MedAgents | 0.501 | 0.622 | 0.569 | 0.435 | 0.532 |
| | MDAgents | 0.435 | 0.744 | 0.582 | 0.422 | 0.546 |
| | AutoGen | 0.395 | 0.656 | 0.568 | 0.448 | 0.517 |
| | DyLAN | 0.414 | 0.610 | 0.540 | 0.448 | 0.503 |
| | MedChain-Agent | 0.462 | **0.746** | **0.621** | 0.474 | **0.576** |

80GB of memory. To enhance output stability and reliability across all experiments, we consistently set the temperature parameter to 0. The experiments were conducted in Chinese.

## 5.2 Benchmark Performance Results

The results of our evaluation in the MedChain are presented in Table 1. Our analysis yields two significant insights:

*(1) Sequential decision-making tasks continue to pose significant challenges, even for advanced models.* For instance, within the single-agent frameworks, GPT-4o-mini and InternVL2 achieve average scores of 0.4156 and 0.3859, respectively. These results indicate that despite their sophistication, these models struggle to maintain consistent performance across the sequential stages of clinical decision-making, highlighting the inherent difficulty of these tasks.

*(2) The integration of the MedChain-Agent framework with open-source LLMs demonstrates significant superiority over proprietary models like GPT-4o-mini.* The substantial performance gain observed with MedChain-Agent (average score of 0.5200) implies that our framework can leverage the strengths of open-source LLMs to achieve superior outcomes. This suggests that open-source

Table 3: **Ablation Study for Key Components for MedChain-Agent.** This table presents the performance impact of sequentially removing the Feedback mechanism and MedCase-RAG from the full MedChain-Agent framework.

| Feedback | MedCase-RAG | Specialty referral | | History-taking | Examination | Diagnosis | Treatment | Average |
|---|---|---|---|---|---|---|---|---|
| | | Level 1 | Level 2 | | | | | |
| | | 0.5523 | 0.2228 | 0.3285 | 0.6369 | 0.4724 | 0.3915 | 0.4341 |
| ✓ | | 0.5739 | 0.2906 | 0.4222 | 0.6377 | 0.4299 | 0.4209 | 0.4692 |
| | ✓ | **0.5928** | 0.3353 | 0.5801 | 0.6488 | 0.4699 | 0.4568 | 0.5140 |
| ✓ | ✓ | 0.5873 | **0.3505** | **0.5836** | **0.6566** | **0.4804** | **0.4613** | **0.5200** |

Table 4: **Generalizability of MedChain-Agent with Various Base LLM.** We apply the MedChain-Agent framework to different base LLMs to validate its generalizability. Performance changes are highlighted with light green for improvements and light red for decreases. Results demonstrate that the MedChain-Agent framework consistently brings performance gains across different base LLMs.

| Base LLM | Specialty Referral | | History-taking | Examination | Diagnosis | Treatment | Average |
|---|---|---|---|---|---|---|---|
| | Level 1 | Level 2 | | | | | |
| GPT-4o-mini | 0.6065 (+0.0616) | 0.2302 (-0.0569) | 0.6205 (+0.2806) | 0.6279 (+0.1167) | 0.4975 (+0.0798) | 0.4329 (+0.0399) | 0.5026 (+0.0870) |
| HuaTuoGPT | 0.2554 (+0.1847) | 0.0097 (-0.0110) | 0.3722 (+0.0257) | - | 0.4159 (-0.0004) | 0.1450 (-0.0330) | 0.2396 (+0.033) |
| Qwen2 | 0.5818 (+0.0843) | 0.2781 (+0.0566) | 0.5962 (+0.1736) | 0.6534 (+0.1705) | 0.4628 (+0.0098) | 0.4315 (+0.2122) | 0.5006 (+0.1178) |
| InternVL2 | 0.5873 (+0.1062) | 0.3505 (+0.1570) | 0.5836 (+0.1191) | 0.6566 (+0.2076) | 0.4806 (+0.0439) | 0.4613 (+0.1710) | 0.5200 (+0.1341) |

models, when enhanced with our framework, are not only competitive but can also excel in handling intricate medical decision-making tasks.

## 5.3 Evaluation in Existing Diagnosis Datasets

Moreover, we evaluate MedChain-Agent on several well-established medical QA datasets, including MedQA [19], PubMedQA [20], PathVQA [17], and MedBullets [3]. As shown in Table 2, our framework demonstrates strong performance compared to both single-agent baselines and multi-agent alternatives. MedChain-Agent achieves the highest average score (0.576). This consistent superiority across various medical QA benchmarks further validates the effectiveness of our framework, even on simpler, more structured tasks that differ from our real-world clinical scenarios.

## 5.4 Ablation Studies and Discussion

*(1) Ablation Study for Key components in MedChain-Agent:* To assess our framework's components, we conduct ablation studies as shown in Table 3. Both the Feedback mechanism and MedCase-RAG module demonstrate significant individual contributions to overall performance. The Feedback mechanism alone improves the average score from 0.4341 to 0.4692, with notable gains in History-taking and Level 2 Specialty referral. MedCase-RAG shows stronger individual impact, boosting the average score to 0.5140, with substantial improvements in History-taking and Specialty referral tasks. While MedCase-RAG excels in diagnostic phases, the Feedback mechanism appears more beneficial for Treatment tasks. When combined, these components show synergistic effects, achieving the highest average performance (0.5200) and optimal scores across five of six evaluated tasks, validating their complementary roles in enhancing clinical reasoning capabilities.

*(2) Ablation Study for Three Key Characteristics in MedChain*: To validate the effectiveness of personalization, interactivity, and sequentiality within our benchmark, we conduct an ablation study as shown in Table 5. We systematically remove each characteristic and observe its impact on model performance across Diagnosis and Treatment tasks. 'w/o Person' means all detailed patient profiles are omitted from the input, resulting in a lack of personalized information that makes correct diagnosis more challenging for the model. 'w/o Seq' means we use the ground truth from the previous stage as input to the next stage, rather than using the model's previous output, which simplifies the benchmark. 'w/o Inter' means we directly provide all patient examination results as input without requiring the agent to autonomously inquire about the patient's condition, also simplifying the benchmark. Removing patient-specific information (*w/o Person.*) consistently degrades performance across all models in diagnosis tasks (with drops ranging from 2.51% to 10.31%), demonstrating that personalized information is crucial for accurate clinical decision-making.

Table 5: **The ablation study for three key characteristics in MedChain.** This table presents the impact of personalization, interactivity, and sequentiality on Diagnosis and Treatment tasks. The arrows ↑↓ next to settings indicate expected performance change direction. The arrows next to results show actual changes (highlighted in gray when matching expectations).

| Setting | Model | Diagnosis | Treatment |
|---|---|---|---|
| Full | MedAgent | 0.4106 | 0.3673 |
| | MDAgent | 0.3959 | 0.3620 |
| | gpt-4o-mini | 0.4157 | 0.3930 |
| | InternVL2 | 0.4378 | 0.4472 |
| | MedChain-Agent | 0.4802 | 0.4613 |
| a) w/o Person. ↓ | MedAgent | 0.3075 ↓ | 0.3754 ↑ |
| | MDAgent | 0.3283 ↓ | 0.3109 ↓ |
| | gpt-4o-mini | 0.3906 ↓ | 0.3406 ↓ |
| | InternVL2 | 0.3702 ↓ | 0.2527 ↓ |
| | MedChain-Agent | 0.4159 ↓ | 0.4310 ↓ |
| b) w/o Seq. ↑ | MedAgent | 0.4030 ↓ | 0.4456 ↑ |
| | MDAgent | 0.4497 ↑ | 0.4418 ↑ |
| | gpt-4o-mini | 0.4522 ↑ | 0.4423 ↑ |
| | InternVL2 | 0.4481 ↑ | 0.2903 ↓ |
| | MedChain-Agent | 0.4807 ↑ | 0.4743 ↑ |
| c) w/o Inter. ↑ | MedAgent | 0.3129 ↓ | 0.4109 ↑ |
| | MDAgent | 0.3998 ↑ | 0.3627 ↑ |
| | gpt-4o-mini | 0.4663 ↑ | 0.4003 ↑ |
| | InternVL2 | 0.4550 ↑ | 0.3173 ↓ |
| | MedChain-Agent | 0.4634 ↓ | 0.5207 ↑ |

Interestingly, when removing the sequential dependency between stages (*w/o Seq.*), most models show improved performance, indicating that sequential decision-making poses greater challenges that better reflect real-world clinical scenarios. Similarly, the improved performance observed after removing interactive information gathering (*w/o Inter.*) confirms the effectiveness of interactivity in our benchmark design. These results collectively suggest that while both sequentiality and interactivity make the benchmark more challenging, they are essential components that better simulate the complexity of real-world clinical decision-making processes.

*(3) Generalizability for MedChain-Agent:* To evaluate the generalizability of our framework, we apply MedChain-Agent to various base LLMs as shown in Table 4. The results demonstrate substantial performance improvements across most models and tasks. InternVL2 shows the most significant enhancement with a 13.41% average improvement, followed by Qwen2 (11.78%) and GPT-4o-mini (8.70%). Even with HuaTuoGPT, which has relatively lower baseline performance, our framework still achieves a 3.33% improvement. Notably, the improvements are particularly pronounced in complex tasks such as History-taking and Examination, where gains of up to 28.06% are observed. These consistent enhancements across diverse models and tasks strongly validate the framework's robust generalizability and effectiveness in medical decision-making scenarios.

# 6 Conclusion

In this paper, we introduced MedChain, a novel benchmark for evaluating LLM-based agents in clinical decision-making that authentically reflects real-world medical practice through three essential characteristics: personalization, interactivity, and sequentiality. Our comprehensive dataset encompasses 12,163 diverse clinical cases across 19 medical specialties, including 7,338 medical images with corresponding reports, providing a robust foundation for evaluating AI systems in complex healthcare scenarios. To address the challenges presented by this benchmark, we also introduced MedChain-Agent, a multi-agent framework enhanced by feedback mechanisms and MedCase-RAG, which demonstrates superior performance across various clinical tasks. This work establishes new benchmarks for evaluating medical AI systems and provides practical solutions for enhancing their clinical decision-making capabilities. As AI continues to evolve in healthcare applications, frameworks that can navigate the complexities of real-world clinical scenarios will be increasingly valuable for improving patient care and supporting healthcare professionals.

# 7 Limitations

This paper has two primary limitations that offer avenues for future research:

1) Data Source Diversity: The MedChain benchmark is constructed from 12,163 electronic health records obtained from the Chinese medical website "iiYi." Although this dataset is extensive and covers 19 medical specialties and 156 sub-categories, it is derived from a single source. Additionally,

there exists a notable imbalance in data distribution across different medical specialties, with some departments having significantly more cases than others. This imbalance may introduce bias in model evaluation and limit performance in underrepresented specialties. In our future work, we will incorporate additional data sources from different regions or healthcare systems to further enrich the dataset and address the specialty imbalance.

2) Patient Interaction Simulation: In our interactive environment, the patient responses during the history-taking stage are generated by the Gemma 2 language model. While this approach ensures consistency and control in evaluating the LLM-based agent, the real patient interactions can be more varied and complex. Future work could explore more advanced patient simulators or incorporate real dialogue data to capture a wider range of communication styles and behaviors.

## Acknowledgments and Disclosure of Funding

This work was supported by the National Natural Science Foundation of China under Grant 82261138629 and 12326610, and Guangdong Provincial Key Laboratory under Grant 2023B1212060076

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

# Appendix for MedChain

**Abstract.**

Appendix A describes the process of standardizing and organizing the dataset for the MedChainbenchmark.

Appendix B provides a detailed explanation of the MedChain-Agent framework, including its implementation, feedback mechanism, and the novel Retrieval-Augmented Generation (RAG) approach used to enhance decision-making.

Appendix C lists the additional details for experiment.

Appendix D discuss the difference between MedChain and several similar related works.

## A    Benchmark Construction and Evaluation

### A.1    Dataset Standardization

We employed a combination of large language models and human verification to label data across different tasks. Our methodology involves task-specific prompt construction and output matching to ensure data quality and diversity while maintaining alignment with real clinical case scenarios.

To ensure consistency and comparability across the benchmark, we standardized the classification of examination items into two main categories: Physical Examinations and Auxiliary Examinations. Physical Examinations include evaluations of various body systems and general health indicators, while Auxiliary Examinations encompass different imaging techniques and laboratory tests. We utilized GPT-4o to extract and classify examination items from each case, followed by manual verification to ensure accuracy. For medical imaging, we classified images into seven types, and manual review ensured the correctness of the classifications. Additionally, treatment items were extracted and categorized from each case. This standardization process ensures that the dataset is consistent, facilitating accurate and comparable evaluations of LLM performance. Figure 4 and Figure 5 demonstrate a case after standardization in English and Chinese. Figure 6 shows the statistics of different departments.

### A.2    Tasks Details

This section provides a comprehensive description of the five specific tasks that make up the MedChainbenchmark. Each subsection elaborates on a particular task, detailing its input, output, and evaluation methods. Figure 9 visualizes the differences between MedChainand other methods.

#### A.2.1    Task 1: Specialty referral

The specialty referral task evaluates the LLM's ability to assess the urgency of a patient's condition and determine the appropriate department based on the patient's chief complaint. The input consists of the patient's chief complaint, and the output space includes 19 first-level departments and 156 second-level departments. The LLM must first assign the patient to one of the 19 first-level departments, then to one or more of the 156 second-level departments based on the primary symptoms.

**Evaluation metrics** include accuracy for first-level department assignment and Intersection over Union (IoU) for second-level department assignment. IoU is used for second-level departments to account for the possibility of multiple correct assignments and to reflect partial correctness, which can occur in complex cases. This metric better captures the nuanced nature of departmental referrals in clinical practice.

#### A.2.2    Task 2: History-taking

The history-taking task is designed to simulate doctor-patient communication, where the goal is to obtain relevant information and infer necessary examination items. We employ a multi-agent system to evaluate this process:

```
{
    "tags": {
        "department": ["Pediatrics", "Pediatric Immunology"],
        "disease": ["Uterine fibroids", "Constipation"]
    },
    "[Case Introduction]": {
        "Physical Examination": {
            "General Examination": "Heart and lungs normal",
            "Urogenital System Examination": "Cervical erosion grade II, uterus enlarged to the size of a duck egg, left
adnexa palpable mass size of a goose egg"
        },
        "Auxiliary Examinations": {
            "CT": "CT plain scan: A mixed-density lesion in the middle of the pelvis, about 10.0cm×7.0cm×6.5cm, with
uneven internal density, CT value -146~45HU, and a round high-density focus, CT value 623HU (Fig 1, 2), with smooth
edges. Uterus enlarged, rounded, protruding slightly anteriorly, with uneven internal density, CT value 32~48HU.
Intrauterine device visible in the uterine cavity, bladder compressed and displaced. Imaging diagnosis: Uterine fibroids,
ovarian teratoma.",
            "Ultrasound": "Color Doppler: A 10.0cm×6.0cm cystic-solid mass visible anterior to the uterus, with
heterogeneous internal echo. A medium-echo mass visible on the right side, with a strong echo focus inside, rich color
blood flow can be drawn out inside and around. The left side is cystic, with chaotic internal echo, cord-like strong
echo visible. Right ovary normal",
            "Blood Test": "Alkaline phosphatase: 89u/L (normal range 100~290u/L)",
            "Pathological Examination": "Tumor diameter 8.0cm, intact capsule, smooth surface; partly cystic diameter
about 3.0cm, containing fat and hair, solid area gray-yellow on section, soft and brittle. Another uterine upper segment
5.0cm×5.0cm×4.0cm, intramural tumor tissue about 2.0cm in diameter, yellowish-white on section, hard and tough.
Microscopy: Rich fibrous stroma separating long, wavy, parallel cell bands composed of columnar cells; elongated nuclei
perpendicular to the long axis of the cell bands (Fig 4). Intracystic lipid seen, cyst wall with squamous epithelium and
sebaceous glands (Fig 3). Another tumor cells spindle-shaped, rod-shaped nuclei, interlaced arrangement, cells without
atypia (Fig 5). Immunohistochemistry: inhibin (-), ck (+), cy (+). Pathological diagnosis: Ovarian carcinoid (trabecular
type) with mature cystic teratoma, uterine leiomyoma."
        },
        "Chief Complaint": ["Patient is 8 years old", "Constipation for nearly 1 year, weight loss of 20 pounds in the
past half year. Regular menstruation, no frequent urination"],
        "Images": [
            {"filename": "133.png", "category": ["Pathological Examination"]},
            {"filename": "131.png", "category": ["CT"]},
            {"filename": "132.png", "category": ["Pathological Examination"]}
        ],
        "Imaging Reports": {
            "CT": "CT plain scan: A mixed-density lesion in the middle of the pelvis, about 10.0cm×7.0cm×6.5cm, with
uneven internal density, CT value -146~45HU, and a round high-density focus, CT value 623HU (Fig 1, 2), with smooth
edges. Uterus enlarged, rounded, protruding slightly anteriorly, with uneven internal density, CT value 32~48HU.
Intrauterine device visible in the uterine cavity, bladder compressed and displaced. Imaging diagnosis: Uterine fibroids,
ovarian teratoma.",
            "Pathological Examination": "Tumor diameter 8.0cm, intact capsule, smooth surface; partly cystic diameter
about 3.0cm, containing fat and hair, solid area gray-yellow on section, soft and brittle. Another uterine upper segment
5.0cm×5.0cm×4.0cm, intramural tumor tissue about 2.0cm in diameter, yellowish-white on section, hard and tough.
Microscopy: Rich fibrous stroma separating long, wavy, parallel cell bands composed of columnar cells; elongated nuclei
perpendicular to the long axis of the cell bands (Fig 4). Intracystic lipid seen, cyst wall with squamous epithelium and
sebaceous glands (Fig 3). Another tumor cells spindle-shaped, rod-shaped nuclei, interlaced arrangement, cells without
atypia (Fig 5). Immunohistochemistry: inhibin (-), ck (+), cy (+). Pathological diagnosis: Ovarian carcinoid (trabecular
type) with mature cystic teratoma, uterine leiomyoma."
        },
    "[Diagnosis and Treatment Process]": {
        "Initial Diagnosis": ["Uterine fibroids, ovarian teratoma."],
        "Treatment Course": ["Intramural fibroid nodules visible in the uterus; left ovary enlarged, multilocular, with
intact and smooth capsule. Left fallopian tube normal, right adnexa normal. Subtotal hysterectomy and left adnexectomy
performed. Pathological examination: Tumor diameter 8.0cm, intact capsule, smooth surface; partly cystic diameter about
3.0cm, containing fat and hair, solid area gray-yellow on section, soft and brittle. Another uterine upper segment
5.0cm×5.0cm×4.0cm, intramural tumor tissue about 2.0cm in diameter, yellowish-white on section, hard and tough.
Microscopy: Rich fibrous stroma separating long, wavy, parallel cell bands composed of columnar cells; elongated nuclei
perpendicular to the long axis of the cell bands (Fig 4). Intracystic lipid seen, cyst wall with squamous epithelium and
sebaceous glands (Fig 3). Another tumor cells spindle-shaped, rod-shaped nuclei, interlaced arrangement, cells without
atypia (Fig 5). Immunohistochemistry: inhibin (-), ck (+), cy (+). Pathological diagnosis: Ovarian carcinoid (trabecular
type) with mature cystic teratoma, uterine leiomyoma."]
    },
    "[Treatment Items]": ["Surgery"]
    }
}
```

Figure 4: **Case Report** "77_Ovarian Carcinoid with Mature Cystic Teratoma: A Case Report."

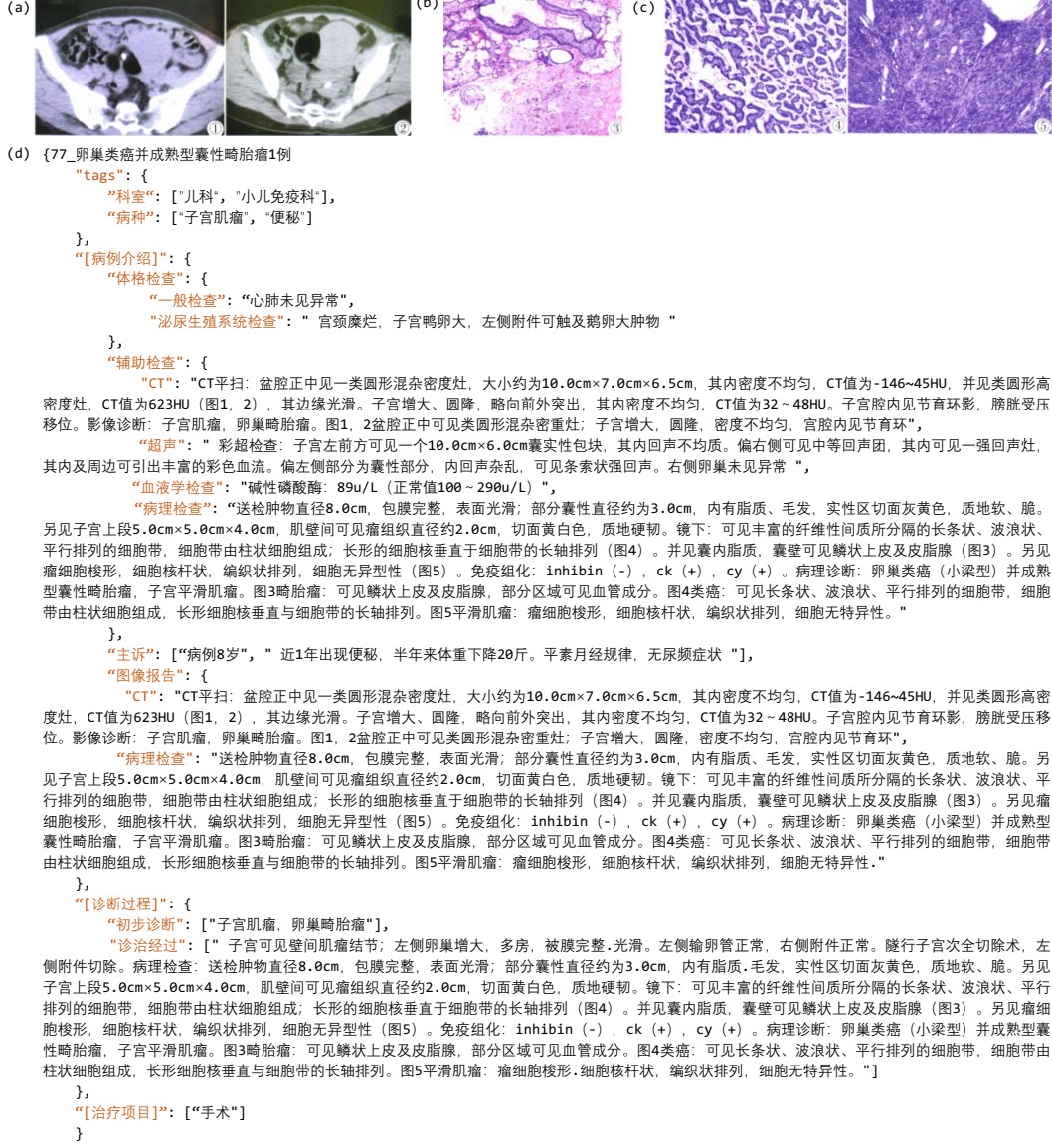

Figure 5: **Case Report Chinese Version with Corresponding Medical Imaging.** "77_Ovarian Carcinoid with Mature Cystic Teratoma: A Case Report." (a), (b) and (c) Medical Imaging. (d) Chinese version.

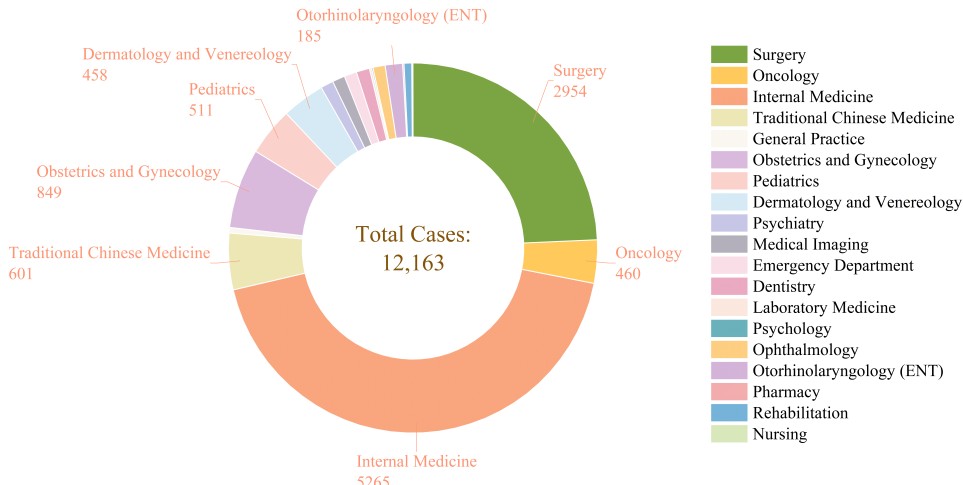

Figure 6: Multi-departmental Distribution

- Doctor Agent: The LLM being evaluated plays this role, asking questions and suggesting examinations based on the patient's responses.
- Patient Agent: A local large model (such as Google's gemma2/9b) simulates the patient, responding based on pre-defined case information. This includes the patient's chief complaint, medical history, and examination results.

**Prompt for Patient**: *You are to role-play as a Standardized Patient. Here is your case information: {patient information}. You need to answer the doctor's questions directly based on the case information (do not fabricate doctor-patient dialogues). Note that unless the doctor explicitly asks about physical examination and auxiliary examination findings, please do not proactively mention or inquire about physical examination and auxiliary examination related content. If the doctor asks about content that does not exist in the case, please indicate that you don't know and avoid fabricating information. At all times, remember that you are only playing the role of a standardized patient.*

**Prompt for Doctor**: *You are a doctor. The patient's chief complaint is as follows: {chief complaints}. You need to gather more information through conversation with the patient. Physical examination includes: general examination (including height, weight, temperature, blood pressure, pulse, etc.), head, eyes, ears, nose, and throat examination, neck examination (including thyroid, cervical lymph nodes), chest examination (including lungs, heart), abdominal examination, spine and limb examination, skin examination, neurological examination, and genitourinary system examination. Auxiliary examinations include: X-ray, MRI, CT, ultrasound, nuclear medicine imaging, hematological tests, urine tests, stool tests, endoscopic examination, and pathological examination. After obtaining certain information (such as past history and present illness history), please select the physical examinations and auxiliary examinations to inquire about based on the patient's condition. Ask about only one or two items per conversation round. Please inquire about as many examination items as possible (ask at least one item each for physical examination and auxiliary examination) until you can determine the condition. At the end of the conversation, please say "Wishing you a speedy recovery."*

**Evaluation metrics** is the IoU between the predicted examination items and the ground truth set. This patient-agent design simulates a realistic clinical history-taking process, allowing the doctor-LLM to demonstrate its ability to ask relevant follow-up questions, interpret patient responses, and determine appropriate examinations. The use of a local large model as the Patient Agent ensures consistency in evaluations and improves the reproducibility of the benchmark.

**Systematic Evaluation**: Furthermore, we conducted a systematic evaluation with a senior physician (10+ years clinical experience) across 10 simulated cases to validate the patient agent's effectiveness. The evaluation focused on three key dimensions: medical history accuracy (match rate between simulated and original EHR data), symptom consistency (temporal coherence of symptom pro-

Evaluation Prompt

Extract Reference Claims

```
    Next, I will give you an imaging report. Please divide it into several statements, ensuring there are no
omissions or repetitions between the statements. Output the resulting statements in the form of a list. The output
format should be: ["Statement 1", "Statement 2", "Statement 3", ...]
    Imaging report:
```

Entailment Evaluation

```
    Please evaluate whether the following AI assistant-provided imaging report can fully include each of the
following statements. Also, explain your answer. For each statement, output "1" or "0", where "1" means the statement
can be fully included in the imaging report, and "0" means the statement contains information not included in the
imaging report. Output the answers as a list composed of dictionaries in text form, not JSON. The output format should
be: [{"Statement": "Original Statement 1", "Explanation": "Reason for the prediction", "Inclusion Prediction": 1 or 0
(whether the statement is included in the imaging report)}, {"Statement": "Original Statement 2", "Explanation": "Reason
for the prediction", "Inclusion Prediction": 1 or 0 (whether the statement is included in the imaging report)}, ...]
    Statements:
    Imaging report:
```

Figure 7: **The prompt for Evaluation in Task3 Examination.**

gression), and treatment response fidelity (accuracy of responses to medications and interventions). Each dimension was rated on a 5-point Likert scale (1=poor, 5=excellent). Results demonstrated strong performance across all metrics, with mean scores of 4.0 (variance=1.0) for match rate, 3.9 (variance=1.49) for temporal coherence, and 3.7 (variance=0.81) for treatment response accuracy.

### A.2.3 Task 3: Examination

The examination task assesses the LLM's ability to analyze medical images and generate corresponding image reports. The input consists of the medical images from each case, and the output is a free-text image report. This task tests the LLM's capability to interpret visual medical data and articulate findings in a clear, professional manner.

**Evaluation metrics:** Given the complexity of medical reports, conventional NLP metrics such as BLEU [37] and BERTScore [69] are inadequate for evaluating this task, as they do not capture whether the generated report aligns with the ground truth in terms of medical claims. Instead, we adopt the Claim Recall metric proposed in DocLens [63] to evaluate the completeness of the generated report. First, GPT-4o-mini is used to extract a list of claims from the original ground truth report. Then, GPT-4o-mini assesses whether the generated report entails these reference claims, and the recall score measures the proportion of claims that are correctly reflected in the generated report. The prompts used for this evaluation can be found in Figure 7.

### A.2.4 Task 4: Diagnosis

The diagnosis task evaluates the LLM's ability to identify the patient's illness based on the provided symptoms and examination results. The input includes the patient's chief complaint, medical history, examination results, and the department visited. The output is a diagnosis in free-text format.

**Evaluation metrics:** We use a five-level grading system to evaluate the accuracy of the diagnosis, ranging from 1 to 5. The prompt we used in LLM-based evaluation is followed the instruction of People's Medical Publishing House [5]. The system accounts for the complex nature of medical diagnoses, where multiple factors may contribute to the patient's condition. The prompt is shown in Figure 8.

### A.2.5 Task 5: Treatment

The treatment task involves formulating a treatment plan based on the patient's diagnosis and examination results. The input includes the patient's chief complaint, medical history, examination results, and diagnosis. The output is a set of proposed treatment items. This task tests the LLM's ability to synthesize the gathered information and apply medical knowledge to develop an appropriate treatment plan, considering factors such as drug interactions, treatment protocols, and patient-specific considerations.

---

[5]Diagnostics. 9th Ed. Beijing: People's Medical Publishing House; 2018

Figure 8: **The prompt for Evaluation in Task4 Diagnosis.**

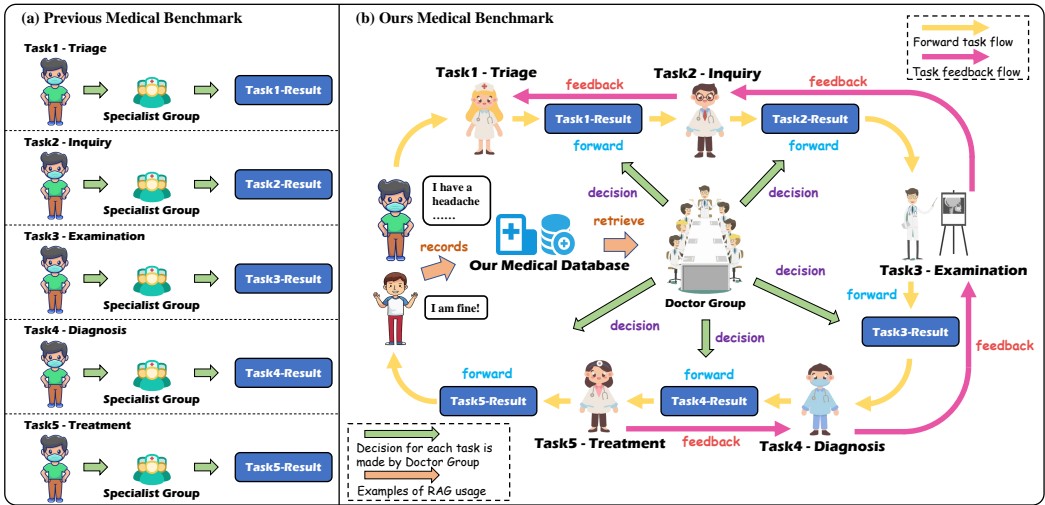

Figure 9: **Comparison between our benchmark method and previous benchmarks.** In Figure (a), the benchmark methods used in previous work are shown, which are based on specific domain problems and cannot achieve a complete end-to-end medical task workflow. In Figure (b), our proposed benchmark is presented, where we effectively accomplish the full workflow of medical tasks through our proposed MedCase-RAG method and the ACFM mechanism.

**Evaluation metrics** is the IoU between the proposed treatments and the ground truth treatment set. Since multiple treatment options may be appropriate, IoU allows for partial credit when the LLM suggests a subset of the recommended treatments or proposes additional reasonable treatments that are not part of the ground truth.

### A.3 Examination Items in Task 2

**Physical Examination:** General examination (including height, weight, temperature, blood pressure, pulse, etc.), head, eyes, ears, nose and throat examination, neck examination (including thyroid, cervical lymph nodes), chest examination (including lungs, heart), abdominal examination, spine and limb examination, skin examination, neurological examination, urogenital system examination.

**Auxiliary Examinations:** X-ray, MRI, CT, ultrasound, nuclear medicine imaging, blood tests, urine tests, stool tests, endoscopy, pathological examination.

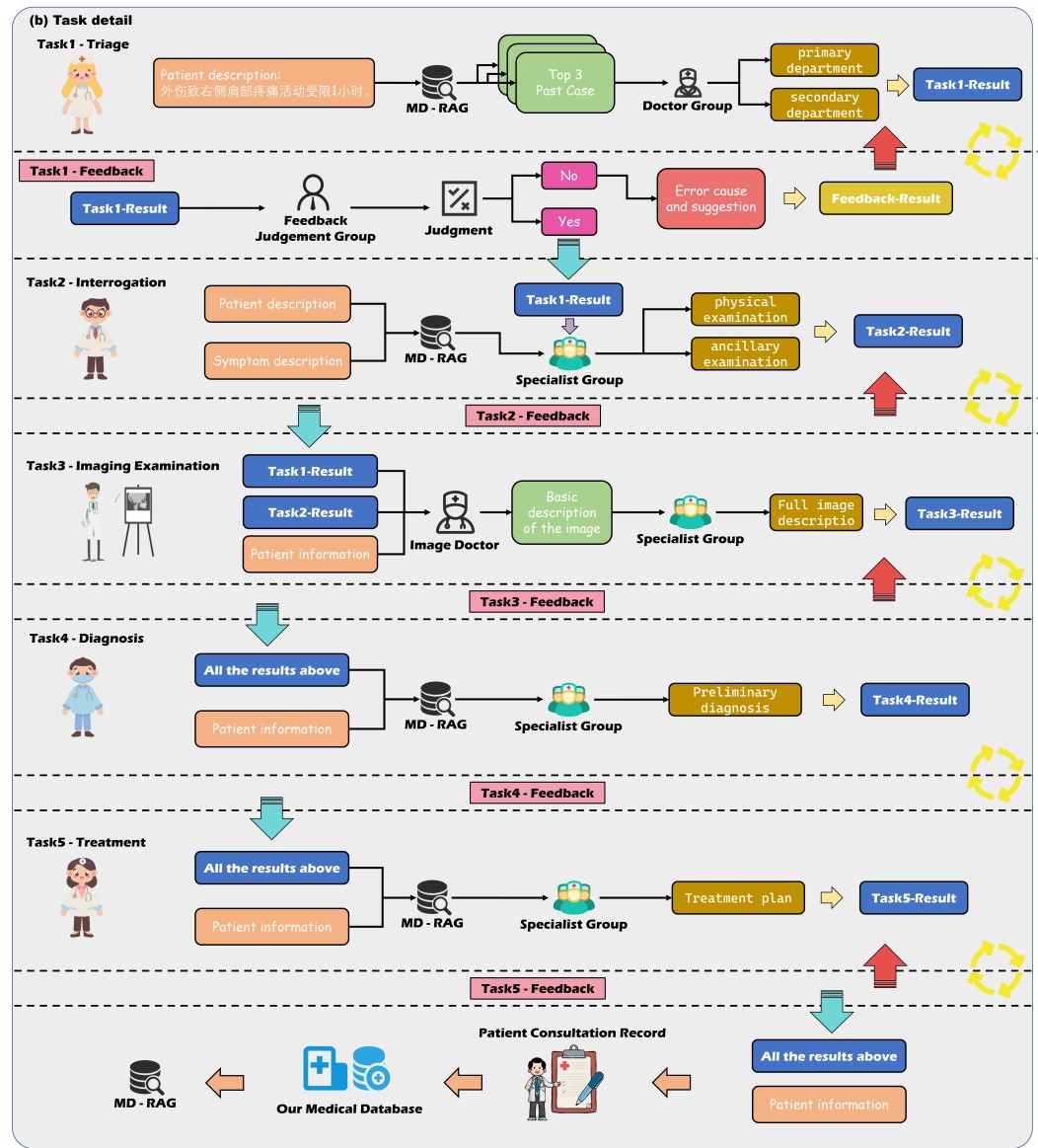

Figure 10: **The specific agent design for each task in our MedChain-Agent and the ACFM of the entire process are presented.**

## A.4    Treatment Items in Task 5

Surgery, interventional therapy, medication, chemotherapy, antibiotic therapy, radiation therapy, physical therapy, immunotherapy, psychological therapy, traditional Chinese medicine, gene therapy.

## B    MedChain-Agent

### B.1    Implementation Details

The overall process and output flow of our framework are illustrated in Figure 10. The patient first describes their condition, after which Task 1 performs specialty referral, assigning the patient to the appropriate primary and secondary departments. Once Task 1 is completed, the doctor group consists of specialists from the corresponding primary department, enabling better analytical decision-making. Task 2 involves these specialists prescribing suitable medical examinations based on the patient's

current situation. Task 3 focuses on analyzing the patient's imaging data to generate a detailed and comprehensive imaging report. Task 4 consists of the specialists conducting a comprehensive analysis based on the results from Task 2 and the imaging report from Task 3 to provide an initial diagnosis for the patient. Finally, Task 5 involves the specialists formulating an appropriate treatment plan based on all the previous tasks' results and analyses.

Additionally, in the aforementioned description, the decision-making and analytical processes for all tasks are conducted through discussions within the Doctor Group. Specifically, three specialists from the same department first engage in discussions and analysis, after which they summarize their findings and results to the final medical generalist (primary care physician) for the final decision. At each step, relevant similar cases are retrieved from our medical database to assist in decision-making. For Task 1, we will divide the tasks of identifying primary and secondary departments into two subtasks based on the Chain of Thought (CoT) approach: first identifying the primary department, followed by identifying the secondary department. Similarly, for Task 2, we first prescribe specific tests before additional examinations, optimizing the decision-making process.

Once the patient completes the entire medical process, a comprehensive treatment follow-up record will be compiled, including treatment outcomes and reports from each stage, along with the patient's information. This data will then be reintroduced into our medical database. Since these data have not been fully validated (we do not know how reasonable the decisions made for the current cases are), we will treat this data as pseudo-data based on the principles of semi-supervised learning and assign it a lower priority. This means that we will first retrieve data from completely validated sources, and if retrieval fails or data is insufficient, we will resort to retrieving from the pseudo-data.

## B.2    Feedback Mechanism Details

To date, no work has proposed a multi-agent framework for simulating the entire medical process. Although existing frameworks demonstrate efficiency in specific medical tasks, their performance is suboptimal when linking various medical stages together. In sequential task scenarios, simply concatenating individual tasks is not feasible. Therefore, our full-process task can be regarded as a multi-sequence task, which necessitates a deeper exploration of the issue of error propagation. When a problem arises in the first task, subsequent tasks will analyze and make decisions based on erroneous results, potentially leading to severe impacts on the entire process.

To address this issue, we have introduced a feedback mechanism within the full-process framework, tightly connecting the current task with all subsequent tasks. After the current task is completed, the output results are evaluated by the physicians of the subsequent tasks (Feedback Judgement Group). Only when the results are confirmed to be accurate will the process advance to the next task; if issues are identified, the reasons for the errors and improvement suggestions will be output and fed back to the current task for re-discussion and decision-making. This process will continue iteratively until consensus is reached on the results or the maximum number of discussion rounds is achieved.

By implementing a feedback mechanism between each task, we can effectively manage the propagation of errors while significantly enhancing collaboration and communication among agent groups. This mechanism encourages agents to share information in real-time, improving mutual understanding and allowing for rapid strategic adjustments when issues are identified, thereby enhancing the overall flexibility and adaptability of the system. Each agent can better respond to changes in a dynamic environment, facilitating more efficient medical services. This feedback-driven collaborative model lays the foundation for the efficient operation of multi-agent systems, aiding in the provision of precise and reliable medical decision support in complex clinical environments.

## B.3    RAG Details

We performed data restructuring on the original dataset, where each case is mapped into feature vectors from 12 dimensions. Among these, "Symptom Description" is identified as the most representative feature of the current patient and is processed through a Text Embedding model for quantification, which is stored in the database for subsequent dense retrieval tasks.

Previous Medical-RAG methods relied on medical question-answering (QA) databases and predominantly used chunked indexing for retrieval. Our approach differs in several ways. Firstly, the content of our foundational medical database is distinct; while most methods have built their databases

using medical QA data, ours utilizes a tree structure. We initially categorize patient information according to primary medical departments (in the experimental section, we categorized into 19 primary departments) and then extract and map patient information into a two-dimensional feature representation. This storage design allows for better retention of the patient's crucial clinical information to assist in decision-making. Secondly, our retrieval method also diverges from theirs. In our retrieval process, we simulate the everyday practice of physicians. When faced with challenging cases that require decision-making, doctors often refer to past cases as references to enhance current decisions and judgments. We treat these two-dimensional features as the minimal unit of a case, using "Symptom description" as the basis for retrieval. When a new patient arrives, we extract and refine their "Symptom description" feature (approximately 70 characters). We employ direct quantitative matching without tokenizing this feature, using a Text Embedding model for quantification. The resulting feature vector is then compared to each case in our database corresponding to the relevant department through cosine similarity calculations, selecting the top three cases with the highest symptom similarity as the current retrieval results. These results, combined with the current case features, are forwarded to subsequent agents for comprehensive decision-making and judgment, enhancing the agents' output. Since our retrieval method does not utilize document chunking but rather employs direct matching, it retains more matching information, leading to improved matching accuracy.

### B.4 Task Prompts

Figure 11 ~ page 30 showcase prompts for each task and stage within our MedChain-Agent framework.

Table 6: **Comparison of Benchmarks for LLM-based Agent.** The MedChaincomposed of 12, 163 case with five sequential stages of the clinical workflow. It provide interaction environment for LLM-based agent evaluation.

| Benchmark | #Num | Modality | Personal Information | Sequential Evaluation | Interaction |
|---|---|---|---|---|---|
| MedQA [70] | 1,273 | Text | ✓ | ✗ | ✗ |
| PubMedQA [20] | 500 | Text | ✗ | ✗ | ✗ |
| MedMCQA [36] | 193,155 | Text | ✓ | ✗ | ✗ |
| MMedBench [39] | 53,566 | Text | ✗ | ✗ | ✗ |
| MedBench [29] | 300,901 | Text | ✓ | ✗ | ✗ |
| Asclepius [54] | 3,232 | Text, Imaging | ✗ | ✗ | ✗ |
| GMAI-MMBench [5] | 26,000 | Text, Imaging | ✗ | ✗ | ✗ |
| MedChain | 12,163 | Text, Imaging | ✓ | ✓ | ✓ |

## C Supplementary for Experiment

Tasks 1 and 2 are subdivided based on the Chain of Thought (CoT) reasoning approach for decision-making. Consequently, Task 1 is split into "specialty referral (Level 1)" and "specialty referral (Level 2)" to evaluate performance at both primary and secondary department levels. Task 2 is divided into "Physical" and "Ancillary," representing performance in physical examinations and ancillary tests. or both ablation and comparative experiments, we split our dataset into training and test sets in a 7:3 ratio. We extracted 12-dimensional features from each training set case and stored them in our Medical Dataset for subsequent retrieval tasks.

## D Related Work Discussion

We compare MedChain with most of related work in Table 6 and Figure 9. Especially, we analyze MedChain with AI Hospital and CoD as following:

**AI Hospital [11]**: While both AI Hospital and our work aim to evaluate LLMs in clinical scenarios through multi-agent interactions, there are several key distinctions. Firstly, MedChain significantly

Figure 11: **The prompt for Task 1 in the MedChain-Agent .**

Figure 12: **The prompt for Task 1 in the MedChain-Agent in Chinese.**

expands the scale and diversity of medical cases, comprising 12,163 cases across 19 specialties and 156 subspecialties, including 7,338 medical images. This represents a substantial advancement over AI Hospital's dataset of 506 cases with limited specialty coverage, enabling more comprehensive evaluation of LLMs' medical capabilities across diverse clinical scenarios. Secondly, MedChain introduces a more sophisticated multi-agent collaboration mechanism. While AI Hospital employs a basic agent interaction model, our framework incorporates a feedback-driven multi-agent system enhanced by the MedCase-RAG module. This module enables dynamic knowledge base expansion and supports case-based reasoning through structured feature representation.

**CoD [4]** introduces an interactive approach to enhance the interpretability of medical diagnosis. While both CoD and our work incorporate interactive components, they differ substantially in both design objectives and implementation mechanisms. CoD primarily focuses on improving the interpretability of the diagnostic phase through confidence-driven interactions that demonstrate the reasoning process. In contrast, MedChain takes a more comprehensive approach by simulating the complete clinical workflow through a multi-agent collaborative framework. Our framework encompasses five sequential stages from triage to treatment, with each stage's decisions being evaluated and guided by subsequent stages through a novel cross-stage feedback mechanism. This design enables MedChain to capture the interdependent nature of clinical decision-making, where decisions at each stage influence and are influenced by other stages in the workflow.

**Mission1: Physical examination**
sys_prompt: You are a professional [first_room] physician with extensive clinical experience, and you need to prescribe appropriate medical examinations for the patient.
user_input: Task Description:Based on the current case situation and with appropriate reference to past cases, prescribe suitable "[task_obj]" for the current patient. If no physical examination is needed, you can output "[task_obj]: None." Please adhere strictly to the output format.Only output the examinations that need to be performed; no need to provide examination results!!!Past Cases:"$[case1][case2][case3]$"Current Case:"Patient's complaint: [case now]"Output Format:"[task_obj]: {......}"

**Mission2: Mission1-Feedback**
sys_prompt: You are a medical physical examination result evaluator with extensive experience in medical examinations. Please review the physical examination items in the physician's "physical examination" to determine whether they are correct.
user_input: Please review the physical examination items in the physician's "physical examination" to determine whether they are correct. Refer to the "past cases" below to assist in your decision-making.Output requirements: (You must follow the requirements below)Output "Correct" if there are no errors, or "Incorrect" if there are errors, and provide reasons!!!Always answer first, then explain the reasons!Physical Examination: " [result] "Past Cases: "$[past_case] $"Output Example: If correct, the output example is: (No reason needed if correct) " Correct " If incorrect, the output example is: (Must explain the reasons for being incorrect!!!) " Incorrect Reason: {Opinion on the items in the "physical examination" result} "
Feedback_result_example: Here are the "physical examination" items ordered by the medical examination physician:"[result]"Here are the opinions of other doctors regarding the ordered "physical examination" items:"[feedback]"

**Mission4: Auxiliary Examination**
sys_prompt: You are a professional [first_room] doctor with extensive clinical interview experience, and you need to prescribe appropriate medical examinations for the patient.
user_input: Task Description:You need to prescribe appropriate "[task_obj]" for the current patient based on the current case situation, with suitable reference to past cases. If no physical examination is required, you can output "[task_obj]: None." Please adhere strictly to the output format.Only output the required examinations, without providing the examination results.Past Cases: "$[case1][case2][case3] $"Current Case: " Patient's Chief Complaint: [case now] "Output Format: " [task_obj]: {......} "

**Mission5: Mission4-Feedback**
sys_prompt: You are a medical auxiliary examination result evaluator with extensive experience in medical testing. Please review the auxiliary examination items listed by the medical examination doctor to determine their accuracy.
user_input: Please review the auxiliary examination items provided by the medical examination doctor to determine their accuracy. Use the following "past cases" distribution of departments to assist in your decision-making.Output requirements: (You must follow the requirements below for your reasoning)If you believe there are no errors, output "Correct." If there are errors, output "Incorrect" and explain the reason!!!Always respond first, then provide the reason!Auxiliary examination: " [result] "Past cases: "$ [past_case] $"Output examples: If you believe it's correct (no reason needed): " Correct " If you believe it's incorrect (you must explain the reason!!!): " Incorrect Reason: {Your opinion on the "auxiliary examination" items} "
Feedback_result_example: Here are the "auxiliary examination" items prescribed by the medical examination doctor:"[result]"Here are the opinions of other doctors regarding the prescribed "auxiliary examination" items:"[feedback]"

Figure 13: **The prompt for Task 2 in the MedChain-Agent .**

```
Task2 Inquiry Prompt

  Mission1: Physical examination
      sys_prompt: 你是一名专业的[first_room]医生，拥有着丰富的临床问诊经验，您需要为患者开具合适的医学检查。
      user_input: 任务说明：1、你需要根据当前病例的情况，适当参考过往病例，为当前患者开具合适的"[task_obj]"，若无需开局体格检查
则可以输出"[task_obj]：None"，请你严格按照输出格式进行输出。2、只输出需要做的检查即可，无需输出检查结果！！！过往病例：
"$[case1][case2][case3]"$当前病例："病人主诉：[case now]"输出格式："[task_obj]：{......}"
    Mission2: Mission1-Feedback
      sys_prompt: 你是一名医学体格检查结果判别者，拥有着丰富的医学检查经验，请你检查医学检查医生的"体格检查"中的这些体格检查项目是
否正确。
      user_input: 请你检查医学检查医生的"体格检查"中的这些体格检查项目是否正确。参考下面的"过往病例"的科室分配情况，辅助做出决策。
输出要求：（你一定要按下面的要求进行思考）1、认为没有错误则输出"正确"，有错误则输出"错误"，并且说明原因！！！2、一定要先回答，然后
再回答原因！体格检查："[result]"过往病例："$[past_case]"$输出示例：认为正确的输出示例：（认为正确则无需输出原因）"正确"认为错误的输
出示例：（认为错误要说明原因，一定要说明原因！！！）"错误原因：{对"体格检查"项目结果的意见}"
        Feedback_result_example: 这是医学检查医生开具的"体格检查"项目："[result]"这是其他医生对开具的"体格检查"项目的意
见:"[feedback]"
    Mission3: Feedback-Answer
      sys_prompt: 你是一名专业的[first_room]医生，拥有着丰富的临床问诊经验，您需要为患者开具合适的医学检查。
      user_input: 请你合理参考其他医生的关于"体格检查"项目的看法和建议，基于病人情况和诊室医生的意见按照原来的格式重新对病人的"体
格检查"开具合适的检查项目。其他医生的意见不一定正确，请你合理对待。只输出修改后的"体格检查"项目即可，无需输出原因和其他说明。参考下
面的"过往病例"的科室分配情况，辅助做出决策。原来的体格检查项目和其他医生的建议："[feedback]"过往病例："$[past_case]"$输出格式："
体格检查：{......}"
    Mission4: Auxiliary Examination
      sys_prompt: 你是一名专业的[first_room]医生，拥有着丰富的临床问诊经验，您需要为患者开具合适的医学检查。
      user_input: 任务说明：1、你需要根据当前病例的情况，适当参考过往病例，为当前患者开具合适的"[task_obj]"，若无需开局体格检查
则可以输出"[task_obj]：None"，请你严格按照输出格式进行输出。2、只输出需要做的检查即可，无需输出检查结果！！！过往病例：
"$[case1][case2][case3]"$当前病例："病人主诉：[case now]"输出格式："[task_obj]：{......}"
    Mission5: Mission4-Feedback
      sys_prompt: 你是一名医学辅助检查结果判别者，拥有着丰富的医学检查经验，请你检查医学检查医生的"辅助检查"中的这些辅助检查项目是
否正确。
      user_input: 请你检查医学检查医生的辅助检查"中的这些辅助检查项目是否正确。参考下面的"过往病例"的科室分配情况，辅助做出决策。
输出要求：（你一定要按下面的要求进行思考）1、认为没有错误则输出"正确"，有错误则输出"错误"，并且说明原因！！！2、一定要先回答，然后
再回答原因！体格检查："[result]"过往病例："$[past_case]"$输出示例：认为正确的输出示例：（认为正确则无需输出原因）"正确"认为错误的输
出示例：（认为错误要说明原因，一定要说明原因！！！）"错误原因：{对"辅助检查"项目结果的意见}"
        Feedback_result_example: 这是医学检查医生开具的"辅助检查"项目："[result]".这是其他医生对开具的"辅助检查"项目的意
见:"[feedback]"
    Mission6: Feedback-Answer
      sys_prompt: 你是一名专业的[first_room]医生，拥有着丰富的临床问诊经验，您需要为患者开具合适的医学检查。
      user_input: 请你合理参考其他医生的关于"辅助检查"项目的看法和建议，基于病人情况和诊室医生的意见按照原来的格式重新对病人的"辅
助检查"开具合适的检查项目。其他医生的意见不一定正确，请你合理对待。只输出修改后的"辅助检查"项目即可，无需输出原因和其他说明。参考下
面的"过往病例"的科室分配情况，辅助做出决策。原来的辅助检查项目和其他医生的建议："[feedback]"过往病例："$[past_case]"$输出格式："
辅助检查：{......}"
```

Figure 14: **The prompt for Task 2 in the MedChain-Agent in Chinese.**

```
Task3 Examination Prompt

   Mission1: Description of imaging by a professional radiologist
      sys_prompt: You are a professional medical imaging physician, and you need to provide a feature description based
on the input medical images.
      user_input: The following are the "basic patient information" and the patient's [img_class] imaging. Please
provide a concise and comprehensive description of the imaging.

Basic Patient Information:
"
[symptom]
"

Output Example: (The descriptions in both the imaging and conclusion sections should be as detailed and comprehensive as
possible.)
"
Chest CT: (Description should be as detailed and comprehensive as possible)
1. Multiple abnormal density shadows in both lungs, mostly located subpleurally, with a large lesion in the right upper
lobe accompanied by multiple cavitary formations, raising the possibility of infection;
2. Small amount of pericardial effusion and a small amount of pleural effusion on the right side;
3. Slight thickening of both pleurae.

**Imaging Report Conclusion:** (Description should be as detailed and comprehensive as possible)
1. No clear abnormal density shadows were seen in both lungs, bronchial structures appear normal, and no significant
dilation was observed.
2. No obvious enlarged lymph nodes were seen in the mediastinum or at both lung hilum.
3. An approximately 8.3 cm soft tissue density mass was observed in the anterior mediastinum.
4. Other structures within the mediastinum showed no significant displacement, and the heart size and shape were normal.

Recommendations:
1. .......
2. .......
"
   Mission2: Image report generation
      sys_prompt: You are a professional {first_room} radiologist with extensive clinical experience, capable of
generating medical imaging reports effectively.
      user_input: Below is the imaging report generated by the previous radiologist based on the 'patient condition'
and the patient's imaging. Please re-evaluate the 'imaging report' below to check for any omissions, incompleteness, or
errors, and if any are found, provide the output as per the example.Patient condition: ' [now_case] 'Imaging report: '
[img_report] 'Output instructions:If there are no suggestions and you consider it correct, output: ' None 'If there are
supplementary suggestions, output: ' It is recommended to supplement, improve, or modify the following parts of the
"imaging report"... '
   Mission3: Mission2-Feedback
      sys_prompt: You are a specialized {first_room} radiologist with extensive clinical experience in imaging. You
need to evaluate whether the previously analyzed report by the expert physician is reasonable based on the patient's
imaging findings.
      user_input: Task Instructions:Below is the imaging report provided by the radiologist.Based on the report and the
patient's imaging, do you think there are issues with this imaging report?If you think the result is correct, output
"Correct"; if you think it is incorrect, output "Incorrect" and explain the reason!You must answer first and then
provide your reasoning.You only need to output your reason or opinion; do not output the original report result or
information.Imaging Doctor's Report Result:"[result]"Output Example:[Correct/Incorrect]Reason:......
   Mission4: Feedback-Answer
      sys_prompt: You are a specialized {first_room} radiologist with extensive clinical experience in imaging. You
need to evaluate whether the previously analyzed report by the expert physician is reasonable based on the patient's
imaging findings.
      user_input: Task Instructions:Please reasonably consider the clinic doctor's opinions and suggestions regarding
the imaging results for the above patient. Based on the patient's imaging and the opinions of other expert doctors,
modify the "current imaging report" according to the original format.Patient Situation:"[now_case]"Current Imaging
Report:"[img_report]"Expert Doctor's Opinion:"[feedback]"Output Example:"Imaging Report: {......}"
```

Figure 15: **The prompt for Task 3 in the MedChain-Agent .**

```
Task3 Examination Prompt

  Mission1: Description of imaging by a professional radiologist
    sys_prompt: 你是一名专业的医学影像医生，你需要根据输入的医学影像对其进行特征描述。
    user_input: 下面传进来的是"病人的基本情况"和病人的[img_class]影像，请你进行精简、全面的影像描述。病人的基本情况：
"[symptom]"输出示例：（影像描述和结论部分的描述尽量详细！）"胸部CT：（描述尽可能详细、全面)1、两肺内多发异常密度影，大部位于胸膜下，
右肺上叶病变范围较大伴多发空洞形成，考虑感染可能大;2、心包少量积液，右侧胸腔少量积液;3、两侧胸膜稍肥厚影像报告结论：（描述尽可能详细、
全面)1．两肺内未见明确异常密度影，支气管正常，未见明显扩张。2．纵隔内及两肺门未见明显肿大淋巴结影。3．前纵隔内见一直径约8.3cm的软
组织密度肿块。4．余纵隔内诸结构未见明显移位，心影大小形态正常。建议：1．.......2．......."

  Mission2: Image report generation
    sys_prompt: 你是一名专业的{first_room}影像学医生，拥有丰富的临床经验，能够很好地生成医学影像报告。
    user_input: 下面是之前的影像学医生基于"病人情况"和病人的影像生成的影像报告。请你基于"病人情况"和病人的影像再次检查一下下面
的"影像报告"是否有遗漏、不完善或者错误的部分，若有则按照输出示例进行输出。仅做补充说明即可，无需完整回答！！病人情况："[now_case]"
影像报告："[img_report]"输出说明：1．若无建议，认为正确则输出："无"2．若有补充建议则输出："认为"影像报告"中需要补充、完善、修改的
部分......"

  Mission3: Mission2-Feedback
    sys_prompt: 你是一名专业的{first_room}影像医生，拥有丰富的影像学临床经验，您需要基于患者的影像情况，判断之前专家医生分析的
报告是否合理。
    user_input: 任务说明：1、下面是影像医生开具影响报告结果。2、基于下面的报告和病人的影像，你认为这个影像报告结果有问题吗？3、
你认为这个结果正确则输出-"正确"，认为不正确则输出-"错误"，并且说明原因！！！4、你一定要先回答，然后再回答原因。5、你只需要输出你的
原因或观点即可，不要输出原来的报告结果或者信息。影像医生的报告结果："[result]"输出示例：[正确/错误]原因：......

  Mission4: Feedback-Answer
    sys_prompt: 你是一名专业的放射科医生，在影像方面有丰富的临床经验。您需要根据患者的影像学检查结果评估专家医师之前分析的报告
是否合理。
    user_input: 任务说明：请你合理参考诊室医生关于上述病人影像结果的看法和建议，基于病人影像图片和其他专家医生的意见按照原来的
格式修改"当前影像报告"。病人情况："[now_case]"当前影像报告："[img_report]"专家医生的意见："[feedback]"输出示例："影像报告:
{......}"
```

Figure 16: **The prompt for Task 3 in the MedChain-Agent in Chinese.**

```
Task4 Diagnosis Prompt

  Mission1: Medical Diagnosis
    sys_prompt: You are a professional {first_room} physician with extensive clinical diagnostic experience. You need
to output the diagnosis based on the patient's condition.
    user_input: You need to provide an appropriate "diagnosis" for the current patient based on the described
"patient condition" and consider past cases for reference. Only reply with one or more diagnosis names, without any
additional explanations, and strictly follow the output format.Patient condition:"[case now]"Past
cases:"$[case1][case2][case3]$"Output example: (directly output your options without additional content)For a single
diagnosis:"'Ankylosing Spondylitis'"For multiple diagnoses: (separated by commas)"'Rheumatoid Arthritis', 'Fracture'"
    Mission2: Mission1-Feedback
    sys_prompt: You are a professional {first_room} physician with extensive clinical diagnostic experience, and you
need to analyze whether the diagnosis made by the clinic doctor is reasonable based on the patient's condition.
    user_input: Please check whether the diagnosis in the "current patient condition and diagnosis results" is
correct. Refer to the "past cases" below to assist in making your decision.Output requirements: (You must follow the
requirements below in your reasoning)If you believe there are no errors, output "Correct." If there are errors, output
"Incorrect," and explain the reason!!!You must answer first, and then provide your reasoning!Current patient condition
and diagnosis results: "[result]"Past cases: " [case1][case2][case3] "Output example: If you believe it is correct,
output example: (No reason is needed if you think it is correct) " Correct " If you believe it is incorrect, output
example: (If you think it is incorrect, provide the reason; you must explain the reason!!!) " Incorrect Reason: {Your
opinion on the "diagnosis result"} "
    Feedback_result_example: Here is the diagnosis result made by the doctor based on the current patient
condition:"[result]"Here are the opinions of other doctors regarding the diagnosis result:"[feedback]"
    Mission3: Feedback-Answer
    sys_prompt: You are a specialized {first_room} physician with extensive clinical diagnostic experience, and you
need to provide a diagnosis based on the patient's condition.
    user_input: Please reasonably consider other doctors' opinions and suggestions regarding the current case's
"diagnosis." Based on the patient's condition and the opinions of the clinic doctor, revise the patient's "diagnosis"
according to the original format. Other doctors' opinions may not be correct, so please assess them critically. Only
output the modified "diagnosis" without any reasons or additional explanations. You can refer to the "past cases"
diagnosis situation to assist in decision-making.Original diagnosis results and other doctors' suggestions: " [feedback]
"Past cases: "$ [past_case] $"Output example: (directly output your options without additional content) For a single
diagnosis: " 'Spondyloarthritis' " For multiple diagnoses: (separated by commas) " 'Rheumatoid arthritis', 'Fracture' "

Task5 Treatment Prompt

  Mission1: Medical treatment
    sys_prompt: You are a specialized {first_room} doctor with extensive clinical treatment experience, and you need
to provide a treatment plan based on the patient's condition.
    user_input: You need to prescribe appropriate treatment options for the current patient based on the described
"patient condition." You should choose suitable options from the "available treatment options" list below, and you can
select multiple options.Refer to the "past cases" treatment options for assistance in making your decision.Patient
condition: " [case now] "Past cases: "$ [case1][case2][case3] "$Available treatment options: ["Surgery", "Interventional
Therapy", "Medication Therapy", "Chemotherapy", "Antibiotic Therapy", "Radiation Therapy", "Physical Therapy",
"Immunotherapy", "Psychotherapy", "Traditional Chinese Medicine", "Gene Therapy"]Output example: (directly output your
options without additional content) For a single treatment option: " Surgery " For multiple treatment options: (separate
with commas) " Medication Therapy, Antibiotic Therapy, Physical Therapy "
    Mission2: Mission1-Feedback
    sys_prompt: You are a professional {first_room} doctor with extensive clinical treatment experience. You need to
analyze whether the treatment results provided by the clinic doctor are reasonable based on the patient's condition.
    user_input: Please check whether the "treatment items" prescribed by the doctor are correct. Refer to the
department allocation situation of the "past cases" below to assist in making your decision.Output requirements: (You
must think according to the requirements below)If you believe there are no errors, output "Correct." If there are errors,
output "Incorrect," and provide a reason!You must answer first, then provide the reason!Treatment items prescribed by
the doctor based on the patient's condition: " [result] "Past medical records: " [case1][case2][case3] "Output example:
If you believe it is correct, the output example is as follows: (If you believe it is correct, no reason is needed) "
Correct " If you believe it is incorrect, the output example is as follows: (If you believe it is incorrect, you must
explain the reason!!!) " Incorrect Reason: {Opinion on the treatment item results} "
    Feedback_result_example: The following are the "diagnosis results" you provided: "[Result]." Below are the
opinions from other doctors regarding your "diagnosis results": "[Feedback]."
    Mission3: Feedback-Answer
    sys_prompt: You are a specialized {first_room} doctor with extensive clinical treatment experience, and you need
to provide a treatment plan based on the patient's condition.
    user_input: Please reasonably consider the opinions and suggestions of other doctors regarding the "treatment
options" for the current case, and based on the patient's condition and the doctor's opinions, re-prescribe appropriate
"treatment options" for the patient according to the original format. You need to choose appropriate options from the
following "available treatment options," and you can select multiple suitable options. Only output the modified
"treatment options," without providing reasons or additional explanations. You may refer to the "past cases" treatment
options to assist in your decision-making.Original treatment options and other doctors' suggestions: " [feedback] "Past
cases: "$ [past_case] $"Available treatment options: ["Surgery", "Interventional Therapy", "Medication", "Chemotherapy",
"Antibiotic Therapy", "Radiation Therapy", "Physical Therapy", "Immunotherapy", "Psychotherapy", "Traditional Chinese
Medicine", "Gene Therapy"]Output example: (directly output your options, do not output extra content) For a single
treatment option: " Surgery " For multiple options: (separated by commas) " Medication, Antibiotic Therapy, Physical
Therapy "
```

Figure 17: **The prompt for Task 4 and Task5 in the MedChain-Agent .**

**Task4 Diagnosis Prompt**

  Mission1: Medical Diagnosis
    sys_prompt: 你是一名专业的{first_room}医生，有着丰富的临床诊断经验，你需要根据病人的情况，输出诊断结果。
    user_input: 你需要根据下面描述的"病人情况"，适当参考过往病例，为当前患者进行合适的"诊断"。仅回复一个或多个诊断结果的名称，不要包含其他任何解释，请严格按照输出格式进行输出。参考下面的"过往病例"的诊断情况，辅助做出决策。病人情况:"[case now]"过往病例："$[case1][case2][case3]"$输出示例:（直接输出你的选项，不要输出多余的内容)输出为单个诊断："'强直性脊柱炎'"输出为多个诊断:（用逗号分隔开)"'风湿性关节炎','骨折'"
    Mission2: Mission1-Feedback
    sys_prompt: 你是一名专业的{first_room}医生，拥有着丰富的临床诊断经验，您需要从患者的情况分析诊室医生的诊断结果是否合理。
    user_input: 请你检查下面"当前病人情况和诊断结果"中的诊断结果是否正确。参考下面的"过往病例"的科室分配情况，辅助做出决策。输出要求:（你一定要按下面的要求进行思考）1、认为没有错误则输出"正确"，有错误则输出"错误"，并且说明原因！！！2、一定要先回答，然后再回答原因！当前病人情况和诊断结果: "[result]"过往病历: "[case1][case2][case3]"输出示例: 认为正确的输出示例:（认为正确则无需输出原因)"正确"认为错误的输出示例:（认为错误要说明原因，一定要说明原因！！！）"错误原因: {对"体格检查"项目结果的意见}"
    Feedback_result_example: 这是医生基于当前病人情况的诊断结果："[result]".这是其他医生对诊断结果的意见:"[feedback]"
    Mission3: Feedback-Answer
    sys_prompt: 你是一名专业的{first_room}医生，有着丰富的临床诊断经验，你需要根据病人的情况，输出诊断结果。
    user_input: 请你合理参考其他医生的关于当前病例"诊断"的看法和建议，基于病人情况和诊室医生的意见按照原来的格式重新对病人进行合适的"诊断"。其他医生的意见不一定正确，请你合理对待。只输出修改后的"诊断"即可，无需输出原因和其他说明。可以参考下面的"过往病例"的诊断情况，辅助做出决策。原来的诊断结果和诊室医生的建议: "[feedback]"过往病例: "$[past_case]"$输出示例:（直接输出你的选项，不要输出多余的内容)输出为单个诊断: "'强直性脊柱炎'"输出为多个诊断:（用逗号分隔开)"'风湿性关节炎','骨折'"

**Task5 Treatment Prompt**

  Mission1: Medical treatment
    sys_prompt: 你是一名专业的{first_room}医生，有着丰富的临床治疗经验，你需要根据病人的情况，输出治疗方案。
    user_input: 你需要根据下面描述的"病人情况"，为当前患者开具合适的治疗项目。你需要从下面"可选择的治疗项目"中选择合适的项目，可以选择多个合适的项目。参考下面的"过往病例"的治疗项目情况，辅助做出决策。病人情况: "[case now]"过往病例："$[case1][case2][case3]"$可选择的治疗项目: ["手术"，"介入治疗"，"药物治疗"，"化学治疗"，"抗生素治疗"，"放射治疗"，"物理疗法"，"免疫疗法"，"心理治疗"，"中医治疗"，"基因治疗"]输出示例:（直接输出你的选项，不要输出多余的内容)输出为单个治疗项目: "手术"输出为多个科室:（用逗号分隔开)"药物治疗,抗生素治疗,物理疗法"
    Mission2: Mission1-Feedback
    sys_prompt: 你是一名专业的{first_room}医生，拥有着丰富的临床治疗经验，您需要从患者的情况分析诊室医生的治疗结果是否合理。
    user_input: 请你检查医生开具的"治疗项目"是否正确。参考下面的"过往病例"的科室分配情况，辅助做出决策。输出要求:（你一定要按下面的要求进行思考）1、认为没有错误则输出"正确"，有错误则输出"错误"，并且说明原因！！！2、一定要先回答，然后再回答原因！医生基于病人情况开具的治疗项目: "[result]"过往病历: "[case1][case2][case3]"输出示例: 认为正确的输出示例:（认为正确则无需输出原因)"正确"认为错误的输出示例:（认为错误要说明原因，一定要说明原因！！！）"错误原因: {对治疗项目结果的意见}"
    Feedback_result_example: 以下是您提供的"诊断结果": "[结果]。"以下是其他医生对您的"诊断结果"的意见: "[反馈]。"
    Mission3: Feedback-Answer
    sys_prompt: 您是一名专业的第一诊室医生，有丰富的临床治疗经验，需要根据患者的情况提供治疗方案。
    user_input: 您是一名专业的第一诊室医生，有丰富的临床治疗经验，需要根据患者的情况提供治疗方案。请合理考虑其他医生对当前病例"治疗方案"的意见和建议，并根据患者的病情和医生的意见，按照原格式重新为患者开出合适的"治疗方案"。您需要从以下"可用的治疗选项"中选择合适的选项，并且您可以选择多个合适的选项。只输出修改后的"治疗选项"，不提供原因或其他解释。您可以参考"过去的病例"治疗方案来帮助您做出决策。原治疗方案及其他医生建议: "[反馈]"既往病例: "$ [past_case] $"可用治疗方案: ["手术"、"介入治疗"、"药物"、"化疗"、"抗生素治疗"、"放射治疗"、"物理治疗"、"免疫治疗"、"心理治疗"、"中药"、"基因治疗"]输出示例: （直接输出您的选项，不输出额外内容）对于单一治疗方案: "手术"对于多个选项:（以逗号分隔)"药物治疗，抗生素治疗，物理治疗

Figure 18: **The prompt for Task 4 and Task5 in the MedChain-Agent in Chinese.**

| Task 1 - Input: |
| --- |
| Patient Information:
"Chief Complaint: Right shoulder pain and limited mobility due to trauma, lasting 1 hour.
Detailed Situation: Patient fell to the ground while working 1 hour ago, landing on their back. They experienced severe pain in the right shoulder immediately, with limited mobility in the right shoulder and right upper limb. At the time, there was no headache, chest pain, or abdominal pain, and no loss of consciousness. Patient came to our hospital for examination. X-ray shows: right scapular fracture. Proposed admission for right scapular fracture treatment. Since admission, patient appears weak, has not had bowel or bladder movements."
Output Example: (If there are multiple options for Secondary Department, separate them with commas)
"Primary Department: Obstetrics and Gynecology
Secondary Department: Gynecologic Endocrinology, Gynecologic Oncology
Reasoning: Because......" |
| **AutoGen** |
| Role1: Experienced Triage Nurse
"Primary Department: Surgery
Secondary Department: Trauma Orthopedics
Reasoning: Based on the patient's chief complaint of 'right shoulder pain and limited mobility due to trauma for 1 hour,' and the detailed description of 'right scapular fracture,' we can determine that the patient needs further treatment in the Surgery Department. This is because Surgery is the main department for treating trauma and fractures."
Role1: Experienced Triage Nurse
"Primary Department: Surgery
Secondary Department: Trauma Orthopedics
Reasoning: Based on the patient's chief complaint of 'right shoulder pain and limited mobility due to trauma for 1 hour,' and the detailed description of 'right scapular fracture,' we can determine that the patient needs further treatment in the Surgery Department. This is because Surgery is the main department for treating trauma and fractures, and Trauma Orthopedics specifically specializes in treating shoulder fractures."
Role1: Experienced Triage Nurse
"Primary Department: Surgery
Secondary Department: Trauma Orthopedics
Reasoning: Based on the patient's chief complaint of 'right shoulder pain and limited mobility due to trauma for 1 hour,' and the detailed description of 'right scapular fracture,' we can determine that the patient needs further treatment in the Surgery Department. This is because Surgery is the main department for treating trauma and fractures, and Trauma Orthopedics specifically specializes in treating shoulder fractures." |
| **DyLAN** |
| Agent1: Question: Patient's Chief Complaint: Right shoulder pain and limited mobility due to trauma, lasting 1 hour.
•Detailed Situation: Patient fell to the ground while working 1 hour ago, landing on their back. They experienced severe pain in the right shoulder immediately, with limited mobility in the right shoulder and right upper limb. At the time, there was no headache, chest pain, or abdominal pain, and no loss of consciousness. Patient came to our hospital for examination. X-ray shows: right scapular fracture. Since admission, patient appears weak, has not had bowel or bladder movements.
•Primary Department: Surgery
•Secondary Department: Trauma Orthopedics
Agent2: Question: Patient's Chief Complaint: Right shoulder pain and limited mobility due to trauma, lasting 1 hour.
•Detailed Situation: Patient fell to the ground while working 1 hour ago, landing on their back. They experienced severe pain in the right shoulder immediately, with limited mobility in the right shoulder and right upper limb. At the time, there was no headache, chest pain, or abdominal pain, and no loss of consciousness. Patient came to our hospital for examination. X-ray shows: right scapular fracture. Since admission, patient appears weak, has not had bowel or bladder movements.
•Primary Department: Surgery
•Secondary Department: Trauma Orthopedics, Vascular Surgery
Agent Importance Scores: [2.0, 1.0]
Final Answer: "Primary Department": "Surgery", "Secondary Department": ["Orthopedics", "Vascular Surgery"] |
| **MedAgent** |
| Expert Gathering: Trauma Medicine, Pain Management, Orthopedics, Rehabilitation Medicine, Emergency Medicine.
Analysis Proposition: Trauma Medicine: 'Right shoulder pain and limited mobility due to trauma for 1 hour. This description indicates the patient experienced right shoulder trauma within the past hour, resulting in pain and limited mobility. This condition may be caused by various factors, including but not limited to shoulder dislocation, rotator cuff tear... Therefore, it is recommended that the patient seek medical attention promptly for detailed physical examination and imaging studies to determine specific diagnosis and treatment plan.'
Pain Management: 'Right shoulder pain and limited mobility due to trauma for 1 hour. This description indicates the patient experienced right shoulder trauma within the past hour, resulting in pain and limited mobility. This condition may be caused by various factors, including but not limited to muscle strain, ligament tear, bone... If the patient experiences other symptoms such as numbness, tingling, or sensory abnormalities, immediate medical attention should be sought as these symptoms may indicate nerve damage.'
Orthopedics: 'Right shoulder pain and limited mobility due to trauma for 1 hour.'
Rehabilitation Medicine: 'Right shoulder pain and limited mobility due to trauma for 1 hour. This description indicates the patient experienced right shoulder trauma within the past hour, resulting in pain and limited mobility. This condition may be caused by various factors, including but not limited to muscle strain, ligament tear, bone... If the patient experiences other symptoms such as numbness, tingling, or sensory abnormalities, immediate medical attention should be sought as these symptoms may indicate nerve damage.'
Emergency Medicine: 'Right shoulder pain and limited mobility due to trauma for 1 hour. This description suggests possible shoulder injury, specifically shoulder dislocation, rotator cuff tear, subacromial impingement syndrome, etc. Among these, shoulder dislocation is the most severe injury requiring immediate medical attention. Rotator cuff... Additionally, if the patient has other underlying conditions such as heart disease, hypertension, etc., prompt medical attention is needed to avoid complications.'
Report Summarization: 'Key Points: Right shoulder pain and limited mobility due to trauma for 1 hour. The most concerning aspect is shoulder dislocation, as delayed treatment may lead to permanent joint dysfunction. Additionally, if the patient has other underlying conditions such as heart disease, hypertension, etc., prompt medical attention is needed to avoid complications. Overall Analysis: Right shoulder pain and limited mobility due to trauma for 1 hour. The most concerning aspect is shoulder dislocation, as delayed treatment may lead to permanent joint dysfunction. Additionally, if the patient has other underlying conditions such as heart disease, hypertension, etc., prompt medical attention is needed to avoid complications.'
Collaborative Consultation: Trauma Medicine: yes Pain Management: yes Orthopedics: yes Rehabilitation Medicine: yes Emergency Medicine: yes
Decision Making: Prompt: Based on the discussion results from each expert above, choose the most appropriate department for patient triage from the given primary and secondary departments.
Final Answer: '-Primary Department: Emergency Department\n-Secondary Department: Trauma Orthopedics' |
| **MDAgent** |
| Complexity Check: basic
Final Answer: 'Recommended Department Classification:\n-Primary Department: Orthopedics\n-Secondary Department: Trauma Orthopedics' |
| **MedChain-Agent** |
| Primary Department Classification by doctor (Agent): Surgery
Feedback: Correct
Secondary Department Classification by doctor (Agent): (Subdivision under Surgery as the primary department) Orthopedics
Feedback: Correct
Final Answer:
Primary Department: Surgery
Secondary Department: Orthopedics |
| **Ground Truth:** |
| Primary Department: "Surgery"
Secondary Department: "Orthopedics" |

Figure 19: **The case study among different multi-agent for task1.**

| Task 1 - Input: |
|---|
| 病人主诉：
"病人主诉：
外伤致右侧肩部疼痛活动受限1小时。
详细情况：
患者于1小时前干活时不慎摔倒在地，背部着地，当时感右侧肩部疼痛难忍，右侧肩部及右上肢活动受限，当时无头痛、胸痛、腹痛，无昏迷，入我院就诊行x片检查示：右侧肩胛骨骨折，拟右侧肩胛骨骨折收入住院，入院来，患者精神软，大小便未解。"
输出示例： （二级科室若有多个选项则用逗号分开）
"一级科室：妇产科
二级科室：妇科内分泌,妇科肿瘤
判断理由：因为......" |

| AutoGen |
|---|
| role1：经验丰富的分诊护士
"一级科室：外科
二级科室：创伤骨科
判断理由：根据病人的主诉，"外伤致右侧肩部疼痛活动受限1小时"，以及详细情况中描述的"右侧肩胛骨骨折"，可以判断病人需要到外科进行进一步治疗 因为外科是处理创伤和骨折等外伤的主要科室。"
role1：经验丰富的分诊护士
"一级科室：外科
二级科室：创伤骨科
判断理由：根据病人的主诉，"外伤致右侧肩部疼痛活动受限1小时"，以及详细情况中描述的"右侧肩胛骨骨折"，可以判断病人需要到外科进行进一步治疗 因为外科是处理创伤和骨折等外伤的主要科室，而创伤骨科是处理肩部骨折的专业科室。"
role1：经验丰富的分诊护士
"一级科室：外科
二级科室：创伤骨科
判断理由：根据病人的主诉，"外伤致右侧肩部疼痛活动受限1小时"，以及详细情况中描述的"右侧肩胛骨骨折"，可以判断病人需要到外科进行进一步治疗 因为外科是处理创伤和骨折等外伤的主要科室，而创伤骨科是处理肩部骨折的专业科室。" |

| DyLAN |
|---|
| Agent1：问题：病人主诉：外伤致右侧肩部疼痛活动受限1小时。
•详细情况：患者于1小时前干活时不慎摔倒在地，背部着地，当时感右侧肩部疼痛难忍，右侧肩部及右上肢活动受限，当时无头痛、胸痛、腹痛，无昏迷，入我院就诊，行X片检查示：右侧肩胛骨骨折。入院来，患者精神软，大小便未解。
•一级科室：外科
•二级科室：创伤骨科
Agent2：问题：病人主诉：外伤致右侧肩部疼痛活动受限1小时。
•详细情况：患者于1小时前干活时不慎摔倒在地，背部着地，当时感右侧肩部疼痛难忍，右侧肩部及右上肢活动受限，当时无头痛、胸痛、腹痛，无昏迷，入我院就诊，行X片检查示：右侧肩胛骨骨折。入院来，患者精神软，大小便未解。
•一级科室：外科
•二级科室：创伤骨科、血管外科
Agent Importance Scores：智能体重要性评分：[2.0, 1.0]
Final Answer：
"一级科室": "外科",
"二级科室": ["骨科", "血管外科"] |

| MedAgent |
|---|
| expert gethering：创伤医学、疼痛管理、骨科、康复医学、急诊医学
analysis proposistion：创伤医学：'外伤致右侧肩部疼痛活动受限1小时。该描述表明患者在过去1小时内经历了右侧肩部的外伤，导致疼痛和活动受限。这种情况可能由多种原因引起，包括但不限于肩关节脱位、肩袖撕裂、...因此，建议患者尽快就医，进行详细的体格检查和影像学检查，以确定具体的诊断和治疗方案。'
疼痛管理：'外伤致右侧肩部疼痛活动受限1小时。该描述表明患者在过去1小时内经历了右侧肩部的外伤，导致疼痛和活动受限。这种情况可能由多种原因引起，包括但不限于肌肉拉伤、韧带撕裂、骨...患者出现其他症状，如麻木、刺痛或感觉异常，应立即就医，因为这些症状可能表明神经损伤。'
骨科：'外伤致右侧肩部疼痛活动受限1小时。'
康复医学：'外伤致右侧肩部疼痛活动受限1小时。该描述表明患者在过去1小时内经历了右侧肩部的外伤，导致疼痛和活动受限。这种情况可能由多种原因引起，包括但不限于肌肉拉伤、韧带撕裂、骨...患者出现其他症状，如麻木、刺痛或感觉异常，应立即就医，因为这些症状可能表明神经损伤。'
急诊医学：'外伤致右侧肩部疼痛活动受限1小时。该描述提示患者可能发生了肩部损伤，具体可能是肩关节脱位、肩袖撕裂、肩峰下撞击症等。其中，肩关节脱位是最严重的损伤，需要立即就医。肩袖...外，如果患者有其他基础疾病，如心脏病、高血压等，也需要及时就医，以避免并发症的发生。'
report summarization：
'关键点：外伤致右侧肩部疼痛活动受限1小时。最令人担忧的方面是肩关节脱位，因为如果不及时治疗，可能会导致永久性的关节功能丧失。此外，如果患者有其他发病疾病，如心脏病、高血压等，也需要及时就医，以避免并发症的发生。总分析：外伤致右侧肩部疼痛活动受限1小时。最令人担忧的方面是肩关节脱位，因为如果不及时治疗，可能会导致永久性的关节功能丧失。此外，如果患者有其他基础疾病，如心脏病、高血压等，也需要及时就医，以避免并发症的发生。'
Collaborative consultation：创伤医学：yes 疼痛管理：yes 骨科：yes 康复医学：yes 急诊医学：yes
Decision Making：prompt：基于上面各个专家给出的讨论结果，在下述给定的一级科室和二级科室中选择最合适的科室进行病患分诊。
final answer：'-一级科室: 急诊科\n-二级科室: 创伤骨科' |

| MDAgent |
|---|
| Complexity Check：basic
Final Answer：'建议科室分级：\n-一级科室: 骨科\n-二级科室: 创伤骨科' |

| MedChain-Agent |
|---|
| 一级科室分类doctor（Agent）：外科
Feedback：正确
二级科室分类doctor（Agent）：（外科为一级科室的前提下在进行细分）骨科
Feedback：正确
Final Answer：
一级科室：外科
二级科室：骨科 |

| Ground Truth: |
|---|
| 一级科室："外科"
二级科室："骨科" |

Figure 20: **The case study among different multi-agent for task 1in Chinese.**

**Task 2 - Input:**

Patient Information:
"Chief Complaint:
Right shoulder pain and limited mobility due to trauma, lasting 1 hour.
Detailed Situation:
Patient fell to the ground while working 1 hour ago, landing on their back. They experienced severe pain in the right shoulder immediately, with limited mobility in the right shoulder and right upper limb. At the time, there was no headache, chest pain, or abdominal pain, and no loss of consciousness. Patient came to our hospital for examination. X-ray shows: right scapular fracture. Proposed admission for right scapular fracture treatment. Since admission, patient appears weak, has not had bowel or bladder movements."
Output Example:
"Physical Examination: General examination,
Chest examination
Auxiliary Examination: X-ray, Colonoscopy
Reasoning: Because......"

**AutoGen**

Role1: Experienced General Practitioner 1
"Physical Examination: General examination, Head examination, Chest examination, Abdominal examination, Spine, limb joints and motor function examination
Auxiliary Examination: X-ray, Chest X-ray, Abdominal X-ray, Magnetic Resonance Imaging (MRI), Complete Blood Count, Urinalysis
Reasoning: Physical examination: Including general examination, head examination, chest examination, abdominal examination, spine, limb joints and motor function examination. Due to the patient's history of trauma, it is necessary to perform general examination, head examination, chest examination, abdominal examination, spine, limb joints and motor function examination.
Auxiliary Examination: X-ray, Chest X-ray, Abdominal X-ray, Magnetic Resonance Imaging (MRI), Complete Blood Count, Urinalysis. Due to the patient's history of trauma, X-ray, chest X-ray, abdominal X-ray, and MRI examinations are needed to understand the fracture situation. Additionally, complete blood count and urinalysis are needed to check for other complications."
Role2: Experienced General Practitioner 2
"Physical Examination: General examination, Head examination, Chest examination, Abdominal examination, Spine, limb joints and motor function examination
Auxiliary Examination: X-ray, Chest X-ray, Abdominal X-ray, Magnetic Resonance Imaging (MRI), Complete Blood Count, Urinalysis
Reasoning: Due to the patient's history of trauma, it is necessary to perform general examination, head examination, chest examination, abdominal examination, spine, limb joints and motor function examination. Furthermore, due to the fracture, X-ray, chest X-ray, abdominal X-ray, and MRI examinations are needed to understand the fracture situation. Additionally, complete blood count and urinalysis are needed to check for other complications."
Role2: Experienced General Practitioner 2
"Physical Examination: General examination, Head examination, Chest examination, Abdominal examination, Spine, limb joints and motor function examination
Auxiliary Examination: X-ray, Chest X-ray, Abdominal X-ray, Magnetic Resonance Imaging (MRI), Complete Blood Count, Urinalysis
Reasoning: Due to the patient's history of trauma, it is necessary to perform general examination, head examination, chest examination, abdominal examination, spine, limb joints and motor function examination. Furthermore, due to the fracture, X-ray, chest X-ray, abdominal X-ray, and MRI examinations are needed to understand the fracture situation. Additionally, complete blood count and urinalysis are needed to check for other complications."
Role1: Experienced General Practitioner 1
"Physical Examination: General examination, Head examination, Chest examination, Abdominal examination, Spine, limb joints and motor function examination
Auxiliary Examination: X-ray, Chest X-ray, Abdominal X-ray, Magnetic Resonance Imaging (MRI), Complete Blood Count, Urinalysis
Reasoning: Due to the patient's history of trauma, it is necessary to perform general examination, head examination, chest examination, abdominal examination, spine, limb joints and motor function examination. Furthermore, due to the fracture, X-ray, chest X-ray, abdominal X-ray, and MRI examinations are needed to understand the fracture situation. Additionally, complete blood count and urinalysis are needed to check for other complications."

**DyLAN**

### Agent1:
- **Routine examination**: Needed to assess overall condition.
- **Right shoulder and arm examination**: Critical for evaluating injury extent.
- **Chest examination**: Related due to back injury, but lower priority for shoulder issues.
- **Neck examination**: Important step in evaluating potential cervical spine issues.
**Advantages**: - Includes key examination information, such as routine checks, shoulder/arm and neck assessments.
**Disadvantages**: - Not detailed enough, may miss critical neck examination details or more precise shoulder and arm checks.
**Score**: 2.5/5
### Agent2:
- **Routine examination**: Necessary for understanding patient's overall condition.
- **Neck range of motion examination**: Important given shoulder area trauma.
- **X-ray and CT scans**: Necessary for detailed imaging.
- **Right shoulder MRI**: Recommended for better soft tissue injury observation.
**Advantages**: - Detailed explanation of various imaging methods, helpful for structural assessment.
**Disadvantages**: - Over-emphasizes imaging tests, neglects detailed physical examination.
**Score**: 4.0/5
### Detailed Updated Answer:
1. **Physical Examination**:
- **Routine examination**: Covers vital signs, overall physical condition, and awareness of any emergencies. - Upper limb examination: Especially right shoulder and arm.
    - Check right shoulder range of motion (ROM), determine if movement is limited by pain.
    - Note any abnormal alignment or dislocation.
- **Neck examination**: Consider possible cervical injury given back injury history.
- **Neck range of motion examination (ROM)**: Ensure stability and check for any trauma-induced movement restrictions or injuries.
2. **Imaging Studies**:
- **Right shoulder X-ray**: For initial bone integrity assessment.
- **Right shoulder CT**: For detailed examination of bones, joints, and soft tissue.
- **Right shoulder MRI**: Provides comprehensive soft tissue imaging, including rotator cuff, bursa, and ligaments. Particularly important for ligament tears or severe soft tissue injuries not visible on X-ray or CT.
3. **None**
### Consensus Answer:
- **Routine examination**
- **Right shoulder and arm examination**
- **Neck examination and range of motion assessment**
- **Right shoulder and axilla X-rays** (including acromioclavicular view, sagittal lateral and glenohumeral joint)
- **Right shoulder CT scan**
- **Right shoulder MRI**
### Agent Importance Scores:
**Agent Importance Scores**: [2.0, 1.0]
### Final Answer:
Auxiliary Examination: "MRI","CT"
Physical Examination: "General examination", "Neck examination"

**MedAgent**

Expert Gethering:
Orthopedics, Pain Management, Rehabilitation Medicine, Emergency Medicine, Surgery
Analysis Proposition:
Orthopedics: Based on the description, the patient has right shoulder pain and limited mobility due to trauma, which may involve multiple injuries, including but not limited to shoulder dislocation, rotator cuff tear, clavicle fracture, or proximal humeral fracture.\n\nThe most concerning aspect is shoulder dislocation, due to...examination (like X-ray or MRI) for diagnosis, and take appropriate treatment measures based on specific conditions, such as reduction, fixation, or surgery.
Pain Management: Right shoulder pain and limited mobility due to trauma for 1 hour.\n\nThis case describes a patient who developed right shoulder pain and limited mobility within 1 hour. This condition may be caused by multiple factors, including but not limited to shoulder dislocation, rotator cuff injury, subacromial...conduct detailed physical examination and imaging studies to determine if there is shoulder dislocation and take appropriate treatment measures promptly.
Rehabilitation Medicine: Based on the description, the patient has right shoulder pain and limited mobility due to trauma, which may involve multiple injuries, including but not limited to shoulder dislocation, rotator cuff tear, subacromial impingement syndrome, acromioclavicular joint injury.\n\nThe most concerning aspect is shoulder dislocation...perform detailed physical examination and imaging studies (such as X-ray, MRI, etc.) to clarify diagnosis and develop corresponding treatment plan.
Emergency Medicine: This case describes a patient who developed right shoulder pain and limited mobility after trauma. This condition may involve multiple possible diagnoses, including but not limited to shoulder dislocation, rotator cuff injury, subacromial bursitis, acromioclavicular joint injury.\n\nMost...other related symptoms such as swelling, bruising or sensory abnormalities. This information can help doctors diagnose and treat patients more accurately.
Surgery: Right shoulder pain and limited mobility due to trauma for 1 hour.\n\nThis case describes a patient who developed right shoulder pain and limited mobility within 1 hour. This condition may be caused by multiple factors, including but not limited to shoulder dislocation, rotator cuff injury, subacromial...conduct detailed physical examination and imaging studies to determine if there is shoulder dislocation and take appropriate treatment measures promptly.
Report Summarization:
'Key points: Right shoulder pain and limited mobility due to trauma for 1 hour. The most concerning aspect is shoulder dislocation. It is recommended to conduct detailed physical examination and imaging studies, such as X-ray or MRI, as soon as possible to determine whether there is shoulder dislocation or other related injuries.\n\nOverall analysis: This case describes a patient who developed right shoulder pain and limited mobility within 1 hour. This condition may be caused by multiple factors, including but not limited to shoulder dislocation, rotator cuff injury, subacromial bursitis, acromioclavicular arthritis. The most concerning or noteworthy aspect is shoulder dislocation. Shoulder dislocation is a serious injury that can cause shoulder pain, limited mobility and shoulder deformity. If not diagnosed and treated promptly, it may lead to permanent disability. Therefore, for this patient, detailed physical examination and imaging studies, such as X-ray or MRI, should be conducted as soon as possible to determine whether there is shoulder dislocation and take appropriate treatment measures promptly.'
Collaborative Consultation:
Orthopedics: yes
Pain Management: yes
Rehabilitation Medicine: yes
Emergency Medicine: yes
Surgery: yes
Decision Making:
prompt: Based on the discussion results from each expert above, determine appropriate initial physical examination and auxiliary examination.
final answer: '"\nPhysical examination: General examination, Visual inspection, Palpation, Movement examination, Neurological examination\nAuxiliary examination: X-ray, MRI'

**MDAgent**

Complexity Check: basic
Final Answer: "'\nPhysical examination: Oral examination\nAuxiliary examination: CT, Complete blood count\n'"

**MedChain-Agent**

Physical examination doctor group:
doctor1: Physical examination: {"General examination", "Head-eyes-ears-nose-throat examination", "Neck examination", "Chest examination", "Abdominal examination", "Spine and extremity examination", "Skin examination"}
doctor2: Physical examination: {"General examination", "Head-eyes-ears-nose-throat examination", "Neck examination", "Chest examination", "Abdominal examination", "Spine and extremity examination", "Skin examination"}
doctor3: Physical examination: {"General examination", "Head-eyes-ears-nose-throat examination", "Neck examination", "Chest examination", "Abdominal examination", "Spine and extremity examination", "Skin examination"}
Summary: 'Physical examination: {"General examination", "Head-eyes-ears-nose-throat examination", "Neck examination", "Chest examination", "Abdominal examination", "Spine and extremity examination", "Skin examination"}'
Feedback: "Comments on physical examination items: {Physical examination items are incomplete, lacking neurological examination}"
Refine: 'Physical examination: {"General examination", "Head-eyes-ears-nose-throat examination", "Neck examination", "Chest examination", "Abdominal examination", "Spine and extremity examination", "Skin examination", "Neurological examination"}'
Auxiliary examination doctor group:
doctor1: Auxiliary examination: {"X-ray", "CT", "X-ray film"}
doctor2: Auxiliary examination: {"X-ray", "CT"}
doctor3: Auxiliary examination: {"X-ray", "CT"}
Summary: 'Auxiliary examination: {"X-ray", "CT"}'
Feedback: Comments on auxiliary examination: {Auxiliary examination items should be selected based on the patient's specific condition, not generalized.}
Refine: 'Auxiliary examination: {"X-ray", "CT"}'
Final Answer:
'Physical examination: {"General examination", "Head-eyes-ears-nose-throat examination", "Neck examination", "Chest examination", "Abdominal examination", "Spine and extremity examination", "Skin examination", "Neurological examination"}'
'Auxiliary examination: {"X-ray", "CT"}'

**Ground Truth:**

Physical examination: General examination, Spine and extremity examination
Auxiliary examination: X-ray

Figure 21: **The case study among different multi-agent for task2.**

**Task 2 - Input:**
病人情况：
"病人主诉：
外伤致右侧肩部疼痛活动受限1小时。
详细情况：
患者于1小时前干活时不慎摔倒在地，背部着地，当时感右侧肩部疼痛难忍，右侧肩部及右上肢活动受限，当时无头痛、胸痛、腹痛，无昏迷，入我院就诊
行X片检查示'右侧肩胛骨骨折，拟右侧肩胛骨骨折收入住院，入院来，患者精神软，大小便未解。"
输出示例：
"体格检查：一般检查,胸部检查
辅助检查：X-ray,肠镜
判断理由：因为......"

**AutoGen**

role1: 经验丰富的全科医生1
"体格检查：一般检查,头部检查,胸部检查,腹部检查,脊柱、四肢关节及运动功能检查
辅助检查：X-ray,胸部X线片,腹部X线片,磁共振成像(MRI),血常规,尿常规
判断理由：包括一般检查、头部检查、胸部检查、腹部检查、脊柱、四肢关节及运动功能检查，由于患者有外伤史，因此需要进行一般检查、头部检查、胸部检查、腹部检查、脊柱、四肢关节及运动功能检查。
辅助检查：X-ray、胸部X线片、腹部X线片、磁共振成像(MRI)、血常规、尿常规。由于患者有外伤史，需要进行X-ray、胸部X线片、腹部X线片、磁共振成像(MRI)检查，以了解骨折情况。另外，需要进行血常规、尿常规检查，以了解有无其他并发症。"
role2: 经验丰富的全科医生2
"体格检查：一般检查,头部检查,胸部检查,腹部检查,脊柱、四肢关节及运动功能检查
辅助检查：X-ray,胸部X线片,腹部X线片,磁共振成像(MRI),血常规,尿常规
判断理由：由于患者有外伤史，因此需要进行一般检查、头部检查、胸部检查、腹部检查、脊柱、四肢关节及运动功能检查。此外，由于患者有骨折，需要进行X-ray、胸部X线片、腹部X线片、磁共振成像(MRI)检查，以了解骨折情况。另外，需要进行血常规、尿常规检查，以了解有无其他并发症。"
role2: 经验丰富的全科医生2
"体格检查：一般检查,头部检查,胸部检查,腹部检查,脊柱、四肢关节及运动功能检查
辅助检查：X-ray,胸部X线片,腹部X线片,磁共振成像(MRI),血常规,尿常规
判断理由：由于患者有外伤史，因此需要进行一般检查、头部检查、胸部检查、腹部检查、脊柱、四肢关节及运动功能检查。此外，由于患者有骨折，需要进行X-ray、胸部X线片、腹部X线片、磁共振成像(MRI)检查，以了解骨折情况。另外，需要进行血常规、尿常规检查，以了解有无其他并发症。"
role1: 经验丰富的全科医生1
"体格检查：一般检查,头部检查,胸部检查,腹部检查,脊柱、四肢关节及运动功能检查
辅助检查：X-ray,胸部X线片,腹部X线片,磁共振成像(MRI),血常规,尿常规
判断理由：由于患者有外伤史，因此需要进行一般检查、头部检查、胸部检查、腹部检查、脊柱、四肢关节及运动功能检查。此外，由于患者有骨折，需要进行X-ray、胸部X线片、腹部X线片、磁共振成像(MRI)检查，以了解骨折情况。另外，需要进行血常规、尿常规检查，以了解有无其他并发症。"

**DyLAN**

#### Agent1:
- **常规检查**：评估整体状况所需。
- **右肩和手臂检查**：评估受伤范围的关键。
- **胸部检查**：由于背部受伤相关，具有一定的相关性，但对于肩部问题优先级较低。
- **颈部检查**：评估可能涉及颈椎的问题的重要环节。
**优点**：- 包括关键的检查信息，如常规检查、肩膀/手臂和颈部的评估。
**缺点**：- 不够详细，可能遗漏了关键的颈部检查细节或更精确的肩膀和手臂检查。
**评分**：2.5/5
#### Agent2:
- **常规检查**：必要，用于了解患者的整体状况。
- **颈部活动范围检查**：考虑到到肩部区域受到，这是一个重要环节。
- **X光与CT检查**：详细成像所必需。
- **右肩MRI**：建议用于更好地观察软组织损伤。
**优点**：- 详细说明了各种成像方法，对结构性评估很有帮助。
**缺点**：- 过于强调成像检查，忽略了详细的物理检查。
**评分**：4.0/5
### 详细更新后的答案：
1. **身体检查**：
   - **常规检查**：涵盖生命体征、整体身体状况，并警惕任何紧急情况。
   - 上肢检查：特别是右肩和手臂。
     - 检查右肩活动范围（ROM），确定是否存在因疼痛引起的活动受限。
     - 注意任何异常的对齐或脱位现象。
   - **颈部检查**：考虑到有背部受伤的病史，检查是否存在颈椎损伤的迹象。
   - **颈部活动范围检查（ROM）**：确保稳定性，并检查是否存在任何因创伤导致的活动限制或损伤。
2. **影像学检查**：
   - **右肩X光**：用于初步评估骨骼完整性。
   - **右肩CT**：用于详细检查骨骼、关节和软组织。
   - **右肩MRI**：提供软组织的全面图像，包括肩袖、滑囊和韧带。对于X光或CT未能显示的撕裂韧带或严重软组织损伤尤为重要。
3. **无**
### 共识答案：
- **常规检查**
- **右肩和手臂检查**
- **颈部检查和活动范围评估**
- **右肩和腋窝X光检查**（包括肩锁关节视图、矢状侧位和肩盂肱关节）
- **右肩CT检查**
- **右肩MRI**
### Agent Importance Scores:
**智能体重要性评分**：智能体重要性评分：[2.0, 1.0]
### Final Answer:
辅助检查："MRI","CT"
体格检查："一般检查","颈部检查"

**MedAgent**

expert gethering:
骨科、疼痛管理、康复医学、急症医学、外科
analysis proposition:
骨科：根据描述，患者因外伤导致右侧肩部疼痛并活动受限，这种情况可能涉及多种损伤，包括但不限于肩关节脱位、肩袖撕裂、锁骨骨折或肱骨近端骨折等。\n\n最令人担忧的方面是肩关节脱位。因，检查（如X光或MRI）以确诊，并根据具体情况采取适当的治疗措施，如复位、固定或手术。
疼痛管理：外伤致右侧肩部疼痛并活动受限1小时。\n\n该病例描述了一个患者在1小时内出现右侧肩部疼痛和活动受限的情况，这种情况可能由多种原因引起包括但不限于肩关节脱位、肩袖损伤、肩峰下。进行详细的体格检查和影像学检查，以确定是否存在肩关节脱位，并及时采取相应的治疗措施。
康复医学：根据描述，患者因外伤导致右侧肩部疼痛并活动受限，这种情况可能涉及多种损伤，包括但不限于肩关节脱位、肩袖撕裂、肩峰下撞击症、肩锁关节损伤等。\n\n最令人担忧的方面是肩关节脱位...进行详细的体格检查和影像学检查（如X线、MRI等），以明确诊断并制定相应的治疗方案。
急症医学：该病例描述了一个患者在遭受外伤后，出现右侧肩部疼痛和活动受限的情况。这种情况可能涉及多种可能的诊断，包括但不限于肩关节脱位、肩袖损伤、肩峰下撞击症、肩锁关节损伤等。\n\n最...其他相关症状，如肿胀、瘀伤或感觉异常等。这些信息可以帮助医生更准确地诊断和治疗患者。
外科：外伤致右侧肩部疼痛活动受限1小时。\n\n该病例描述了一个患者在1小时内出现右侧肩部疼痛和活动受限的情况。这种情况可能由多种原因引起，包括但不限于肩关节脱位、肩袖损伤、肩峰下。进行详细的体格检查和影像学检查，以确定是否存在肩关节脱位，并及时采取相应的治疗措施。
report summarization:
关键点：外伤致右侧肩部疼痛活动受限1小时，最令人担忧的方面是肩关节脱位。建议尽快进行详细的体格检查和影像学检查，如X线或MRI，以确定是否存在右肩关节脱位或其他相关损伤。\n\n总分析：该病例描述了一个患者在1小时内出现右侧肩部疼痛和活动受限的情况，这种情况可能由多种原因引起，包括但不限于肩关节脱位、肩袖损伤、肩峰下撞击、肩锁关节炎等。最令人担忧或最值得关注的方面是肩关节脱位，肩关节脱位是一种严重的损伤，可能导致肩部疼痛、活动受限和肩部畸形。如果不及时诊断和治疗，可能会导致永久性残疾。因此，对于该患者，应尽快进行详细的体格检查和影像学检查，如X线或MRI，以确定是否存在肩关节脱位，并及时采取相应的治疗措施。'
Collaborative consultation:
骨科：yes
疼痛管理：yes
康复医学：yes
急症医学：yes
外科：yes
Decision Making:
prompt：基于上面各个专家给出的讨论结果，开局合适的体格检查和辅助检查。
final answer: "\n体格检查：一般检查,视诊,触诊,动诊,神经系统检查\n辅助检查：X-ray, MRI'

**MDAgent**

Complexity Check: basic
Final Answer: "\n体格检查：口腔检查\n辅助检查：CT, 血常规\n"

**MedChain-Agent**

体格检查doctor group:
doctor1: 体格检查：{"一般检查","头颅眼耳鼻喉检查","颈部检查","胸部检查","腹部检查","脊柱和四肢检查","皮肤检查"}
doctor2: 体格检查：{"一般检查","头颅眼耳鼻喉检查","颈部检查","胸部检查","腹部检查","脊柱和四肢检查","皮肤检查"}
doctor3: 体格检查：{"一般检查","头颅眼耳鼻喉检查","颈部检查","胸部检查","腹部检查","脊柱和四肢检查","皮肤检查"}
Summary: 体格检查：{"一般检查","头颅眼耳鼻喉检查","颈部检查","胸部检查","腹部检查","脊柱和四肢检查","皮肤检查"}
Feedback: "体格检查"项目的意见{体格检查项目不全，缺少神经系统检查}"
Refine: 体格检查：{"一般检查","头颅眼耳鼻喉检查","颈部检查","胸部检查","腹部检查","脊柱和四肢检查","皮肤检查","神经系统检查"}
辅助检查doctor group:
doctor1: 辅助检查：{"X-ray","CT","X线片"}
doctor2: 辅助检查：{"X-ray","CT"}
doctor3: 辅助检查：{"X-ray","CT"}
Summary: 辅助检查：{"X-ray","CT"}
Feedback: 辅助检查的意见：{辅助检查项目应根据患者的具体情况进行选择，不应一概而论。}
Refine: 辅助检查：{"X-ray","CT"}
Final Answer:
体格检查：{"一般检查","头颅眼耳鼻喉检查","颈部检查","胸部检查","腹部检查","脊柱和四肢检查","皮肤检查","神经系统检查"}
辅助检查：{"X-ray","CT"}

**Ground Truth:**
体格检查：一般检查, 脊柱和四肢检查
辅助检查：X-ray

**Task 3 - Input:**
I need you to analyze these medical images through discussion, and ultimately provide a concise Chinese comprehensive imaging report.
Image path: ['./datasets/MedImg/20807.png']

**AutoGen**
Role 1: Experienced radiologist 1
Based on the provided medical images, the patient may have a certain type of tumor. The images show clear abnormal tissue, with irregular borders that contrast significantly with the surrounding tissue. The imaging report recommends further examinations, such as MRI or CT scans, as well as possible pathological examination, to determine the type and stage of the tumor, and further formulate a treatment plan.
Role 2: Experienced radiologist 2
The patient may have a certain type of tumor. The images show clear abnormal tissue, with irregular borders that contrast significantly with the surrounding tissue. It is recommended to conduct further examinations, such as MRI or CT scans, as well as possible pathological examination, to determine the type and stage of the tumor, and further formulate a treatment plan.
Role 1: Experienced radiologist 1
Based on the provided medical images, the patient may have a certain type of tumor. The images show clear abnormal tissue, with irregular borders that contrast significantly with the surrounding tissue. The imaging report recommends further examinations, such as MRI or CT scans, as well as possible pathological examination, to determine the type and stage of the tumor, and further formulate a treatment plan.
Role 2: Experienced radiologist 2
Based on the provided medical images, the patient may have a certain type of tumor. The images show clear abnormal tissue, with irregular borders that contrast significantly with the surrounding tissue. It is recommended to conduct further examinations, such as MRI or CT scans, as well as possible pathological examination, to determine the type and stage of the tumor, and further formulate a treatment plan.

**DyLAN**
### Agent1:
#### Patient Information
- **Name**: [Patient Name]
- **Gender**: [Patient Gender]
- **Age**: [Patient Age]
- **Examination Date**: [Examination Date]
#### Imaging Description
This imaging examination includes anteroposterior (AP) and lateral X-ray films. The imaging shows the following:
1. **Skeletal Structure**:
   - The bilateral shoulder joints, clavicles, ribs, spine, pelvis, and bilateral femurs are all clearly displayed.
   - No obvious fractures or abnormalities are seen in the clavicles and ribs.
   - The spine shows normal physiological curvature, with no significant scoliosis or abnormalities.
2. **Soft Tissue**:
   - No obvious abnormalities are seen in the soft tissues around the bilateral shoulder and hip joints.
   - The bilateral femoral heads and acetabula appear normal.
3. **Internal Fixation**:
   - Metallic internal fixation devices, including screws and plates, are visible in the proximal right femur.
   - Metallic internal fixation devices are also visible in the proximal left femur, including screws and plates.
#### Diagnosis and Analysis
- **Proximal Right Femur Internal Fixation**:
   - The position of the internal fixation in the proximal right femur is good, with no obvious displacement or loosening.
   - The screws and plates are in normal position, with no signs of foreign body reaction.
- **Proximal Left Femur Internal Fixation**:
   - The position of the internal fixation in the proximal left femur is good, with no obvious displacement or loosening.
   - The screws and plates are in normal position, with no signs of foreign body reaction.
#### Conclusion
- This imaging examination shows that the patient's bilateral proximal femur internal fixation devices are in good position, with no obvious displacement or loosening.
- The skeletal structure and soft tissues show no significant abnormalities.
#### Recommendations
- It is recommended that the patient continue follow-up examinations as instructed by the physician to monitor the condition of the internal fixation devices.
- If the patient experiences any discomfort or changes in symptoms, they should seek medical attention promptly.
#### Reporting Physician
- **Signature**: [Physician Signature]
- **Date**: [Report Date]

**MedAgent**
expert gethering: orthopedics, radiology, imaging, thoracic medicine, spine medicine
analysis proposition: orthopedics, radiology, imaging, thoracic medicine, spine medicine: 'This X-ray shows an image of the patient's left shoulder and left upper arm. The following points can be observed from the image:\n\n1. Fracture: The humerus (i.e. upper arm bone) of the left upper arm has an obvious fracture near the shoulder joint. The fracture line is clearly visible, indicating that the bone is broken here. \n\n2. Internal fixation device: Metal internal fixation devices, including screws and plates, can be seen at the fracture site. This indicates that the patient may have undergone surgery to stabilize the fracture and promote healing. \n\n3. Soft tissue condition: Around the fracture area, the soft tissue density is high, and there may be swelling or hematoma. \n\n4. Shoulder joint: No obvious abnormality is found in the shoulder joint area, but further examination is needed to rule out possible shoulder joint injury. \n\nThe most worrying aspects:\n- Stability of the fracture: Despite the presence of an internal fixation device, the stability of the fracture needs to be confirmed to prevent displacement or re-fracture. \n- Soft tissue condition: Swelling and hematoma may indicate the risk of infection or other complications, which require close observation and treatment. \n- Rehabilitation progress: Patients need to follow the doctor's rehabilitation instructions to ensure that the fracture heals properly and avoid secondary injury. \n\nIt is recommended that patients seek medical attention as soon as possible for detailed examinations and follow-up treatment to ensure the stability of the fracture and the progress of rehabilitation. '
report summarization:'Key points: fracture, internal fixation device, soft tissue condition, shoulder joint, fracture stability, infection risk, rehabilitation progress, detailed examination, follow-up treatment. \n\nTotal analysis: This X-ray shows the image of the patient's left shoulder and left upper arm, in which the humerus of the left upper arm has an obvious fracture near the shoulder joint and has undergone surgery for internal fixation. The soft tissue condition needs further observation to rule out possible infection or other complications. It is recommended that patients seek medical attention as soon as possible for detailed examinations and follow-up treatment to ensure the stability of the fracture and the progress of rehabilitation. '
Collaborative consultation: Orthopedics: yes Radiology: yes Imaging: yes Thoracic medicine: no, Recommendations: 'Revisions: In the report, it is recommended that the patient seek medical attention as soon as possible for detailed examinations and follow-up treatment to ensure the stability of the fracture and the progress of rehabilitation. It is recommended that the specific hospital or clinic that the patient needs to see a doctor can be further clarified, and the patient is advised to perform initial self-care before seeing a doctor, such as ice compresses, elevating the affected limb, etc., to reduce pain and swelling. In addition, it is recommended that the report add recommendations for rehabilitation guidance for patients, such as appropriate physical therapy and rehabilitation training to promote fracture healing and restore function. '
Spinal medicine: no, Recommendations: 'Revisions: In the report, it is recommended that the patient seek medical attention as soon as possible for detailed examinations and follow-up treatment to ensure the stability of the fracture and the progress of rehabilitation. It is recommended that the specific examinations and treatment plans that the patient needs to receive can be further clarified, such as whether physical therapy or rehabilitation training is needed, and whether regular X-rays are needed to monitor fracture healing. In addition, it is recommended to pay attention to the patient's pain management to ensure that the patient can comfortably carry out necessary activities during the rehabilitation process. '

Decision Making:
prompt: Based on the discussion results given by the experts above, start with an appropriate imaging report.
final answer:'Overall analysis: This X-ray shows the patient's left shoulder and left upper arm. The humerus of the left upper arm has an obvious fracture near the shoulder joint and has undergone internal fixation. The soft tissue condition needs further observation to rule out possible infection or other complications. It is recommended that the patient seek medical attention as soon as possible for detailed examination and follow-up treatment to ensure the stability of the fracture and the progress of rehabilitation.'

**MDAgent**
Complexity Check: basic
Final Answer: "X-ray": "Imaging report\n\nPatient number: XX\nExamination date: XX/XX/XX\n\nImage description:\n1. Chest X-ray images show that there is a large amount of gas in the chest cavity, the lung tissue is compressed, and the transparency of the lung tissue increases, indicating the possible presence of pneumothorax.\n\n2. X-ray images of the pelvis and bilateral femurs show that there is a metal foreign body in the proximal end of the right femur, and the foreign body is in contact with the acetabular joint, suggesting a possible surgical internal fixation device.\n\n3. There are metal implants at the upper end of both femurs, which are in the correct position, and no foreign body displacement or breakage is found.\n\n4. Spine images show that the spine is neatly aligned, without obvious deformity or dislocation.\n\n5. No obvious abnormal lesions, such as nodules or masses, are found in the lung field.\n\nIn summary, the images suggest that the patient has pneumothorax and may require further clinical evaluation and treatment. X-ray images of the pelvis and femurs show that there is a surgical internal fixation device, which is in a normal position and no abnormalities are found."

**MedChain-Agent**
Description by a professional imaging doctor:''\nImage description:\n1. The X-ray shows a clear fracture line in the right scapula area, which is oblique and extends from the top to the bottom of the scapula.\n2. The bone density around the fracture line is uneven, and there may be slight callus formation.\n3. The joint space of the shoulder joint is well maintained, and there is no obvious joint effusion or joint destruction.\n4. There is no obvious abnormality in the soft tissue structure of the scapula.\n\nConclusion of the imaging report:\n1. The right scapula is fractured, and the fracture line is oblique, extending from the top to the bottom of the scapula.\n2. The bone density around the fracture line is uneven, and there may be slight callus formation.\n3. The joint space of the shoulder joint is well maintained, and there is no obvious joint effusion or joint destruction.\n4. There is no obvious abnormality in the soft tissue structure of the scapula.\n\nRecommendations:\n1. Patients should avoid strenuous activities to avoid further injury.\n2. It is recommended to further perform CT or MRI examination to evaluate the specific situation of the fracture and the soft tissue injury. \n3. Depending on the specific situation of the fracture, surgical fixation or conservative treatment may be required. \n4. Regular follow-up to observe the fracture healing and functional recovery. \n\n''
Doctor Group:
doctor1: The report content is sufficient, no additional content
doctor2: The report content is sufficient, no additional content
doctor3: The report content is sufficient, no additional content
Feedback: Correct
Final Answer:''\nImage description: \n1. The X-ray shows a clear fracture line in the right scapula area, which is oblique and extends from the top to the bottom of the scapula. \n2. The bone density around the fracture line is uneven, and there may be slight callus formation. \n3. The joint space of the shoulder joint is well maintained, and no obvious joint effusion or joint destruction is found. \n4. There is no obvious abnormality in the soft tissue structure of the scapula. \n\nConclusion of the imaging report: \n1. The right scapula is fractured, and the fracture line is oblique, extending from the top to the bottom of the scapula. \n2. The bone density around the fracture line is uneven, and there may be slight callus formation. \n3. The joint space of the shoulder joint is well maintained, and there is no obvious joint effusion or joint destruction. \n4. There is no obvious abnormality in the soft tissue structure of the scapula. \n\nRecommendations: \n1. Patients should avoid strenuous activities to avoid further injury. \n2. Further CT or MRI examinations are recommended to evaluate the specific situation of the fracture and soft tissue injury. \n3. Depending on the specific situation of the fracture, surgical fixation or conservative treatment may be required. \n4. Regular review to observe the fracture healing and functional recovery. \n\n''

**Ground Truth:**
Comminuted fracture of the right calcaneus, please combine with clinical findings; no obvious fractures were found in the bones of the right ankle joint.

Figure 23: **The case study among different multi-agent for task3.**

| Task 3 - Input: |
| --- |
| 我需要你们通过讨论，对这些医学影像进行分析，最终提供一个中文简短的综合性影像报告。 |
| 影像路径：['./datasets/MedImg/20807.png'] |

**AutoGen**

role1：经验丰富的影像科医生1

根据提供的医学影像，患者可能患有某种类型的肿瘤。影像显示有明显的异常组织，边界不规则，与周围组织对比明显。影像报告建议进行进一步的检查，例如MRI或CT扫描，以及可能的病理学检查，以确定肿瘤的类型和分期，并进一步制定治疗计划。

role2：经验丰富的影像科医生2

患者可能患有某种类型的肿瘤。影像显示有明显的异常组织，边界不规则，与周围组织对比明显。建议进行进一步的检查，例如MRI或CT扫描，以及可能的病理学检查，以确定肿瘤的类型和分期，并进一步制定治疗计划。

role1：经验丰富的影像科医生1

根据提供的医学影像，患者可能患有某种类型的肿瘤。影像显示有明显的异常组织，边界不规则，与周围组织对比明显。影像报告建议进行进一步的检查，例如MRI或CT扫描，以及可能的病理学检查，以确定肿瘤的类型和分期，并进一步制定治疗计划。

role2：经验丰富的影像科医生2

根据提供的医学影像，患者可能患有某种类型的肿瘤。影像显示有明显的异常组织，边界不规则，与周围组织对比明显。建议进行进一步的检查，例如MRI或CT扫描，以及可能的病理学检查，以确定肿瘤的类型和分期，并进一步制定治疗计划。

**DyLAN**

### Agent1
#### 患者信息
- **姓名**：患者姓名患者姓名
- **性别**：患者性别患者性别
- **年龄**：患者年龄患者年龄
- **检查日期**：检查日期检查日期
#### 影像描述
本次影像检查包括前后位（AP）和侧位（Lateral）X线片。影像显示如下：
1. **骨骼结构**：
   - 双侧肩关节、锁骨、肋骨、脊柱、骨盆及双侧股骨均显示清晰。
   - 锁骨和肋骨未见明显骨折或异常。
   - 脊柱显示正常生理弯曲，未见明显侧弯或异常。
2. **软组织**：
   - 双侧肩关节和髋关节周围软组织未见明显异常。
   - 双侧股骨头和髋臼显示正常。
3. **内固定物**：
   - 右侧股骨近端可见金属内固定物，包括螺钉和钢板。
   - 左侧股骨近端亦可见金属内固定物，包括螺钉和钢板。
#### 诊断与分析
- **右侧股骨近端内固定物**：
   - 右侧股骨近端内固定物位置良好，未见明显移位或松动。
   - 螺钉和钢板位置正常，未见明显异物反应。
- **左侧股骨近端内固定物**：
   - 左侧股骨近端内固定物位置良好，未见明显移位或松动。
   - 螺钉和钢板位置正常，未见明显异物反应。
#### 结论
- 本次影像检查显示，患者双侧股骨近端内固定物位置良好，未见明显移位或松动。
- 骨骼结构及软组织未见明显异常。
#### 建议
- 建议患者继续按医嘱进行复查，观察内固定物情况。
- 如有任何不适或症状变化，请及时就医。
#### 报告医师
- **签名**：医师签名医师签名
- **日期**：报告日期报告日期
### Agent2：同上
### Agent Importance Scores：智能体重要性评分：[0.5, 0.5]
### Final Answer："X-ray"："### 影像报告\n\n#### 患者信息\n- **姓名**：[患者姓名]\n- **性别**：[患者性别]\n- **年龄**：[患者年龄]\n- **检查日期**：[检查日期]\n\n#### 影像描述\n本次影像检查包括前后位（AP）和侧位（Lateral）X线片。影像显示如下：\n\n1. **骨骼结构**：\n - 双侧肩关节、锁骨、肋骨、脊柱、骨盆及双侧股骨均显示清晰。\n - 锁骨和肋骨未见明显骨折或异常。\n - 脊柱显示正常生理弯曲，未见明显侧弯或异常。\n2. **软组织**：\n - 双侧肩关节和髋关节周围软组织未见明显异常。\n3. **内固定物**：\n - 右侧股骨近端可见金属内固定物，包括螺钉和钢板。\n - 左侧股骨近端亦可见金属内固定物，包括螺钉和钢板。\n\n#### 诊断与分析\n- **右侧股骨近端内固定物**：\n - 右侧股骨近端内固定物位置良好，未见明显移位或松动。\n - 螺钉和钢板位置正常，未见明显异物反应。\n\n#### 结论\n- 本次影像检查显示，患者双侧股骨近端内固定物位置良好，未见明显移位或松动。\n- 骨骼结构及软组织未见明显异常。\n\n#### 建议\n- 建议患者继续按医嘱进行复查，观察内固定物情况。\n- 如有任何不适或症状变化，请及时就医。\n\n#### 报告医师\n- **签名**：[医师签名]\n- **日期**：[报告日期]"

**MedAgent**

expert gathering：骨科、放射学、影像学、胸腔医学、脊柱医学

analysis proposition：骨科、放射学、影像学、胸腔医学、脊柱医学：'这张X光片显示了患者的左肩和右上臂的影像。从影像中可以观察到以下几点：\n\n1. 骨折：右上臂的肱骨（即上臂骨）在靠近肩关节处有明显的骨折，骨折线清晰可见，表明骨骼在此处断裂。\n2. 内固定装置：在骨折处可以看到金属内固定装置，可能是螺钉和钢板。这表明患者可能已经接受了手术治疗，以稳定骨折并促进愈合。\n3. 软组织情况：在骨折区域周围，软组织密度较高，可能存在肿胀或血肿。\n\n4. 关节：肩关节区域未见明显异常，但需要进一步检查以排除可能的肩关节损伤。\n\n最令人担忧的方面：\n- 骨折的稳定性：尽管有内固定装置，但需要确认骨折的稳定性，以防止移位或再次断裂。\n- 软组织情况：肿胀和血肿可能提示有感染或其他并发症的风险，需要密切观察和处理。\n- 康复进展：患者需要遵循医生的康复指导，确保骨折正确愈合，避免二次损伤。\n- 建议患者尽快就医，进行详细检查和后续治疗，以确保骨折的稳定性和康复进展。'

report summarization：'关键点：骨折，内固定装置，软组织情况，肩关节，骨折稳定性，感染风险，康复进展，详细检查，后续治疗。\n\n总分析：这张X光片显示了患者的左肩和右上臂的影像，其中右上臂的肱骨在靠近肩关节处有明显的骨折，并已进行了内固定装置的手术。软组织情况需要进一步观察，以排除可能的感染或其他并发症。建议患者尽快就医，进行详细检查和后续治疗，以确保骨折的稳定性和康复进展。'

Collaborative consultation：骨科：yes 放射学：yes 影像：yes 胸腔医学：no，建议：'修订：在报告中，建议患者尽快就医，进行详细检查和后续治疗，以确保骨折的稳定性和康复进展。建议可以进一步明确患者需要就医的具体医院或诊所，并建议患者在就医前先进行初步的自我护理，如冰敷、抬高患肢等，以减轻疼痛和肿胀。此外，建议报告中增加对患者进行康复指导的建议，如适当的物理治疗和康复训练，以促进骨折的愈合和恢复功能。'

脊柱医学：no，建议：'修订：在报告中，建议患者尽快就医，进行详细检查和后续治疗，以确保骨折的稳定性和康复进展。建议可以进一步观察患者需要接受的具体检查和治疗方案，例如是否需要进行物理治疗或康复训练，以及是否需要定期复查X光片以监测骨折愈合情况。此外，建议关注患者的疼痛管理，以确保患者在康复过程中能够舒适地进行必要的活动。'

Decision Making:

prompt：基于上面各个专家给出的讨论结果，开具合适的影像报告。

final answer：'总分析：这张X光片显示了患者的左肩和右上臂的影像，其中左上臂的肱骨在靠近肩关节处有明显的骨折，并已进行了内固定装置的手术。软组织情况需要进一步观察，以排除可能的感染或其他并发症。建议患者尽快就医，进行详细检查和后续治疗，以确保骨折的稳定性和康复进展。'

**MDAgent**

Complexity Check: basic

Final Answer："X-ray"："影像报告\n\n患者编号：XX\n检查日期：XX年XX月XX日\n\n影像描述：\n1. 胸片影像显示胸腔内含有大量气体，肺组织被压缩，肺组织透明度增加，提示可能存在气胸。\n\n2. 骨盆及双侧股骨X线影像显示右侧股骨近端存在金属异物，异物位置与髋臼关节接触，提示可能的手术内固定装置。\n\n3. 双侧股骨上端有金属内植入物，位置正确，未见异物移位或断裂。\n\n4. 脊柱影像显示脊柱排列整齐，未见明显畸形或脱位。\n\n5. 肺野中未见明显异常病变，如结节或肿块。\n\n综上所述，影像提示患者存在气胸，可能需要进一步的临床评估及治疗。骨盆及股骨X线影像提示存在手术内固定装置，位置正常，未见异常。"

**MedChain-Agent**

专业影像学医生描述："\n影像描述：\n1. 该X光片显示右侧肩胛骨区域有明显的骨折线，骨折线呈斜行，从肩胛骨的上方延伸至下方。\n2. 骨折线周围的骨质密度不均，可能存在轻微的骨痂形成。\n3. 肩关节的关节间隙保持良好，未见明显关节积液或关节破坏。\n4. 肩胛骨的软组织结构未见明显异常。\n\n影像报告结论：\n1. 右侧肩胛骨骨折，骨折线呈斜行，从肩胛骨的上方延伸至下方。\n2. 骨折线周围的骨质密度不均，可能存在轻微的骨痂形成。\n3. 肩关节的关节间隙保持良好，未见明显关节积液或关节破坏。\n4. 肩胛骨的软组织结构未见明显异常。\n\n建议：\n1. 患者应避免剧烈活动，避免进一步损伤。\n2. 建议进一步行CT或MRI检查，以评估骨折的具体情况及软组织损伤情况。\n3. 根据骨折的具体情况，可能需要手术固定或保守治疗。\n4. 定期复查，观察骨折愈合情况及功能恢复情况。\n\n"

Doctor Group:

doctor1：报告内容充分，无补充内容

doctor2：报告内容充分，无补充内容

doctor3：报告内容充分，无补充内容

Feedback: 正确

Final Answer："\n影像描述：\n1. 该X光片显示右侧肩胛骨区域有明显的骨折线，骨折线呈斜行，从肩胛骨的上方延伸至下方。\n2. 骨折线周围的骨质密度不均，可能存在轻微的骨痂形成。\n3. 肩关节的关节间隙保持良好，未见明显关节积液或关节破坏。\n4. 肩胛骨的软组织结构未见明显异常。\n\n影像报告结论：\n1. 右侧肩胛骨骨折，骨折线呈斜行，从肩胛骨的上方延伸至下方。\n2. 骨折线周围的骨质密度不均，可能存在轻微的骨痂形成。\n3. 肩关节的关节间隙保持良好，未见明显关节积液或关节破坏。\n4. 肩胛骨的软组织结构未见明显异常。\n\n建议：\n1. 患者应避免剧烈活动，避免进一步损伤。\n2. 建议进一步行CT或MRI检查，以评估骨折的具体情况及软组织损伤情况。\n3. 根据骨折的具体情况，可能需要手术固定或保守治疗。\n4. 定期复查，观察骨折愈合情况及功能恢复情况。\n\n"

**Ground Truth:**

右跟骨粉碎性骨折，请结合临床；右侧踝关节诸骨未见明显骨折。

Figure 24: **The case study among different multi-agent for task3 in Chinese.**

| Task 4 - Input: |
| --- |
| The patient's comprehensive report is as follows: |
| "Patient's main complaint: right shoulder pain and limited movement for 1 hour due to trauma. |
| Details: The patient accidentally fell to the ground while working 1 hour ago, and landed on his back. At that time, he felt unbearable pain in the right shoulder, and limited movement of the right shoulder and right upper limb. He had no headache, chest pain, abdominal pain, or coma at the time. He was admitted to our hospital for treatment. X-ray examination showed: right scapula fracture. He was hospitalized for right scapula fracture. Since admission, the patient was weak and had difficulty urinating and defecating." |
| Only reply with the name of one or more diagnosis results, and do not include any other explanations. |
| Output example: |
| "Diagnosis result: fracture, knife wound |
| Judgment reason: because..." |

**AutoGen**

role1: Experienced general practitioner 1
"Diagnosis: right scapula fracture
Judgment reason: X-ray examination shows: right scapula fracture."
role2: Experienced general practitioner 2
"Diagnosis: right scapula fracture
Judgment reason: X-ray examination shows: right scapula fracture."
role2: Experienced general practitioner 2
"Diagnosis: right scapula fracture
Judgment reason: X-ray examination shows: right scapula fracture."
role1: Experienced general practitioner 1
"Diagnosis: right scapula fracture
Judgment reason: X-ray examination shows: right scapula fracture."

**DyLAN**

Agent 1: Shoulder joint
Agent 2: Shoulder dislocation
Agent Importance Scores: [2.0, 1.0]
Final Answer: Diagnosis: Shoulder dislocation

**MedAgent**

expert gethering: Traumatology, Orthopedics, Pain Management, Imaging, Rehabilitation
analysis proposistion: Traumatology: The patient was diagnosed with right shoulder pain and limited mobility due to trauma, and was treated at Traumatology and Orthopedics. X-rays showed degenerative lesions of bilateral shoulder joints and a history of surgery on the right shoulder joint. The most worrying aspect is the history of surgery on the right shoulder joint, which may indicate that the patient has...disease, which may further affect the function of the shoulder joint. Therefore, timely treatment is required to avoid further loss of joint function.
Orthopedics: The patient was diagnosed with right shoulder pain and limited mobility due to trauma, and was treated at Traumatology and Orthopedics. X-rays showed degenerative lesions of bilateral shoulder joints and a history of surgery on the right shoulder joint. The most worrying aspect is the history of surgery on the right shoulder joint, which may indicate that the patient has...disease, which may further affect the function of the shoulder joint. Therefore, timely treatment is required to avoid further loss of joint function.
Pain management: According to the information provided, the patient has right shoulder pain and limited mobility due to trauma, and X-rays show bilateral degenerative lesions of the shoulder joints and a history of surgery on the right shoulder joint. These findings suggest that the patient may have chronic problems with the shoulder joint, such as rotator cuff injury, shoulder degeneration... Treatment options. This may include physical therapy, medication, or surgical intervention, depending on the patient's specific situation and medical history.
Imaging: According to the information provided, the patient has right shoulder pain and limited mobility due to trauma, and X-rays show bilateral degenerative lesions of the shoulder joints and a history of surgery on the right shoulder joint. These findings suggest that the patient may have chronic problems with the shoulder joint, such as rotator cuff injury, shoulder degeneration... Further imaging examinations, such as MRI or CT scans, are required to determine the specific pathological changes and develop an appropriate treatment plan.
Rehabilitation: The patient was treated by trauma orthopedics and orthopedics due to right shoulder pain and limited mobility caused by trauma. X-rays show bilateral degenerative lesions of the shoulder joints and a history of surgery on the right shoulder joint. The most worrying aspect is the surgical history of the right shoulder joint, which may indicate that the patient has a loss of joint function in the next step. It is recommended that the patient seek medical attention as soon as possible for detailed examination and treatment to relieve symptoms and prevent worsening of the condition.
report summarization:'Key points: Traumatic right shoulder pain and limited activity for 1 hour. X-rays show degenerative lesions of bilateral shoulder joints and a history of surgery on the right shoulder joint. The most worrying aspect is the surgical history of the right shoulder joint, which may indicate that the patient has previous shoulder surgery, such as shoulder replacement or revision surgery. Although these surgeries can relieve pain and improve function, they may also increase the risk of postoperative complications such as infection, dislocation, or prosthesis loosening. In addition, the degenerative lesions shown in the X-rays may indicate that the patient has osteoarthritis or other joint diseases, which may further affect the function of the shoulder joint. Timely treatment is required to avoid further loss of joint function. \n\nSummary analysis: The patient was treated in the trauma orthopedics and orthopedics department due to right shoulder pain and limited activity caused by trauma. X-rays show degenerative lesions of bilateral shoulder joints and a history of surgery on the right shoulder joint. The most worrying aspect is the surgical history of the right shoulder joint, which may indicate that the patient has previous shoulder surgery, such as shoulder replacement or revision surgery. Although these surgeries can relieve pain and improve function, they may also increase the risk of postoperative complications, such as infection, dislocation, or loosening of the prosthesis. In addition, the degenerative changes shown in the X-ray may indicate that the patient has osteoarthritis or other joint diseases, which may further affect the function of the shoulder joint. Therefore, timely treatment is required to avoid further loss of joint function. '
Collaborative consultation: Traumatology: yes Orthopedics: yes Pain management: yes Imaging: yes Rehabilitation: yes
Decision Making:
prompt: Based on the discussion results given by the experts above, make a reasonable diagnosis. If no decision can be made, output None.
final answer: None

**MDAgent**

Complexity Check: intermediate
Expert Recruitment:
Agent 1 (👤 1. Orthopedic Surgeon): Specializes in the surgical and non-surgical;
Agent 2 (👤 2. Radiologist): Focuses on the interpretation and diagnosis of medical images such as X;
Agent 3 (👤 3. Traumatologist): Specializes in the diagnosis, treatment, and rehabilitation of trauma
Collaborative Decision Making:
Hierarchy Selection
Participatory Debate
(orthopedic surgeon): Trauma (radiologist): Multiple comminuted fractures of the left tibia and fibula with soft tissue swelling of the right calf and left ankle (traumatologist): Fracture of the left tibia and fibula, fracture of the left patella
Analysis and Synthesis:
== Round 1 == |_Turn 1
Agent 1 (👤 orthopedic surgeon) -> Agent 1 (👤 orthopedic surgeon) : As an orthopedic surgeon who specializes in both surgical and non-surgical treatments of musculoskeletal injuries and disorders, I believe that the patient requires a thorough evaluation of her fractures and potential complications. The opinions from the radiologist and traumatologist are valuable, but an orthopedic surgeon's expertise is crucial in determining the most appropriate management plan, which may include surgical intervention if needed. It's essential to address any potential issues that may arise from the fractures, such as joint instability, infection, or non-union.; Agent 2 (👤 radiologist): No; Agent 3 (👤 traumatologist): No |Turn 2 Agent 1 (👤 orthopedic surgeon): No;
Agent 2 (👤 radiologist): No;
Agent 3 (👤 traumatologist): No
Final Decision: Right scapula fracture

**MedChain-Agent**

Doctor group:
doctor1: fracture
doctor2: scapula fracture
doctor3: fracture
Summary: fracture
Feedback: correct
Final Answer: fracture

**Ground Truth:**

fracture

Figure 25: **The case study among different multi-agent for task4.**

| Task 4 - Input: |
| --- |
| 患者综合报告如下:
"病人主诉: 外伤致右侧肩部疼痛活动受限1小时。
详细情况: 患者1小时前干活时不慎摔倒在地,背部着地,当时感右侧肩部疼痛难忍,右侧肩部及右上肢活动受限,当时无头痛、胸痛、腹痛,无昏迷,入我院就诊,行x片检查示: 右侧肩胛骨骨折,拟右侧肩胛骨骨折收入住院,入院来,患者精神软,大小便未解。"
仅回复一个或多个诊断结果的名称,不要包含其他任何解释。
输出示例:
"诊断结果: 骨折,刀伤
判断理由: 因为......" |

| **AutoGen** |
| --- |
| role1: 经验丰富的全科医生1
"诊断结果: 右侧肩胛骨骨折
判断理由: X片检查示: 右侧肩胛骨骨折。"
role2: 经验丰富的全科医生2
"诊断结果: 右侧肩胛骨骨折
判断理由: X片检查示: 右侧肩胛骨骨折。"
role2: 经验丰富的全科医生2
"诊断结果: 右侧肩胛骨骨折
判断理由: X片检查示: 右侧肩胛骨骨折。"
role1: 经验丰富的全科医生1
"诊断结果: 右侧肩胛骨骨折
判断理由: X片检查示: 右侧肩胛骨骨折。" |

| **DyLAN** |
| --- |
| Agent1:
肩关节
Agent2:
肩关节脱位
Agent Importance Scores:
[2.0, 1.0]
Final Answer:
诊断结果: 肩关节脱位 |

| **MedAgent** |
| --- |
| expert gethering: 创伤骨科、骨科、疼痛管理、影像学、康复
analysis proposition: 创伤骨科: 该患者因外伤导致右侧肩部疼痛和活动受限,就诊于创伤骨科和骨科。X光片显示双侧肩关节的退行性病变和右侧肩关节的手术史。最令人担忧的方面是右侧肩关节的手术史,这可能表明患者有...疾病,这可能会进一步影响肩关节的功能。因此,需要及时处理以避免进一步的关节功能丧失。
骨科: 该患者因外伤导致右侧肩部疼痛和活动受限,就诊于创伤骨科和骨科。X光片显示双侧肩关节的退行性病变和右侧肩关节的手术史。最令人担忧的方面是右侧肩关节的手术史,这可能表明患者有...疾病,这可能会进一步影响肩关节的功能。因此,需要及时处理以避免进一步的关节功能丧失。
疼痛管理: 根据提供的信息,患者因外伤导致右侧肩部疼痛和活动受限,X光片显示双侧肩关节的退行性病变和右侧肩关节的手术史。这些发现表明患者可能存在肩关节的慢性问题,如肩袖损伤、肩关节退...的治疗方案。这可能包括物理治疗、药物治疗或手术干预,具体取决于患者的具体情况和病史。
影像学: 根据提供的信息,患者因外伤导致右侧肩部疼痛和活动受限,X光片显示双侧肩关节的退行性病变和右侧肩关节的手术史。这些发现表明患者可能存在肩关节的慢性问题,如肩袖损伤、肩关节退...要进一步的影像学检查,如MRI或CT扫描,以确定具体的病理变化和制定适当的治疗计划。
康复: 该患者因外伤导致右侧肩部疼痛和活动受限,就诊于创伤骨科和骨科。X光片显示双侧肩关节的退行性病变和右侧肩关节的手术史。最令人担忧的方面是右侧肩关节的手术史,这可能表明患者...步的关节功能丧失。建议患者尽快就医,进行详细的检查和治疗,以缓解症状并防止病情恶化。
report summarization: '关键点: 外伤致右侧肩部疼痛活动受限1小时。X光片显示双侧肩关节的退行性病变和右侧肩关节的手术史。最令人担忧的方面是右侧肩关节的手术史,这可能表明患者有既往的肩部手术,如肩关节置换或修复手术。这些手术虽然可以缓解疼痛和改善功能,但也有可能增加术后并发症的风险,如感染、脱位或假体松动。此外,X光片显示的退行性病变可能表明患者有骨关节炎或其他关节疾病,这可能会进一步影响肩关节的功能。需要及时处理以避免进一步的关节功能丧失。\n\n总分析: 该患者因外伤导致右侧肩部疼痛和活动受限,就诊于创伤骨科和骨科。X光片显示双侧肩关节的退行性病变和右侧肩关节的手术史。最令人担忧的方面是右侧肩关节的手术史,这可能表明患者有既往的肩部手术,如肩关节置换或修复手术。这些手术虽然可以缓解疼痛和改善功能,但也有可能增加术后并发症的风险,如感染、脱位或假体松动。此外,X光片显示的退行性病变可能表明患者有骨关节炎或其他关节疾病,这可能会进一步影响肩关节的功能。因此,需要及时处理以避免进一步的关节功能丧失。'
Collaborative consultation: 创伤骨科: yes 骨科: yes 疼痛管理: yes 影像学: yes 康复: yes
Decision Making:
prompt: 基于上面各个专家给出的讨论结果,进行合理的诊断。若无法做出决策,则输出None。
final answer: None |

| **MDAgent** |
| --- |
| Complexity Check: intermediate
Expert Recruitment:
Agent 1 (👤 1. Orthopedic Surgeon): Specializes in the surgical and non;
Agent 2 (👤 2. Radiologist): Focuses on the interpretation and diagnosis of medical images such as X;
Agent 3 (👤 3. Traumatologist): Specializes in the diagnosis, treatment, and rehabilitation of trauma
Collaborative Decision Making:
Hierarchy Selection
Participatory Debate
(orthopedic surgeon): 创伤 (radiologist): 左胫腓骨多处粉碎性骨折伴右小腿及左踝部软组织肿胀 (traumatologist): 左胫腓骨骨折,左髌骨骨折
Analysis and Synthesis:
== Round 1 ==
|_Turn 1
Agent 1 (👤 orthopedic surgeon) -> Agent 1 (👤 orthopedic surgeon) : As an orthopedic surgeon who specializes in both surgical and non-surgical treatments of musculoskeletal injuries and disorders, I believe that the patient requires a thorough evaluation of her fractures and potential complications. The opinions from the radiologist and traumatologist are valuable, but an orthopedic surgeon's expertise is crucial in determining the most appropriate management plan, which may include surgical intervention if needed. It's essential to address any potential issues that may arise from the fractures, such as joint instability, infection, or non-union.;
Agent 2 (👤 radiologist): No;
Agent 3 (👤 traumatologist): No
|Turn 2 Agent 1 (👤 orthopedic surgeon): No;
Agent 2 (👤 radiologist): No;
Agent 3 (👤 traumatologist): No
Final Decision: 右侧肩胛骨骨折 |

| **MedChain-Agent** |
| --- |
| Doctor group:
doctor1: 骨折
doctor2: 肩胛骨骨折
doctor3: 骨折
Summary: 骨折
Feedback: 正确
Final Answer: 骨折 |

| **Ground Truth:** |
| --- |
| 骨折 |

Figure 26: **The case study among different multi-agent for task4 in Chinese.**

| Task 5 - Input: |
|---|
| The patient's comprehensive report is as follows: |
| "Patient's main complaint: right shoulder pain and limited movement for 1 hour due to trauma. |
| Details: The patient accidentally fell to the ground while working 1 hour ago, and landed on his back. At that time, he felt unbearable pain in the right shoulder, and limited movement of the right shoulder and right upper limb. He had no headache, chest pain, abdominal pain, or coma at the time. He was admitted to our hospital for treatment. X-ray examination showed: right scapula fracture. He was planned to be hospitalized for right scapula fracture. Since admission, the patient was weak and had difficulty urinating and defecating." |
| Only reply with the name of one or more treatment items, do not include any other explanations, and make sure the answer is among the above eleven items!!! |
| If there are multiple options, separate them with commas. |
| Output example: |
| "Treatment items: Traditional Chinese medicine treatment, gene therapy, surgery |
| Judgment reason: because...... " |
| **AutoGen** |
| role1: Experienced general practitioner 1 |
| "Treatment: Surgery |
| Judgment reason: The patient has a scapula fracture and needs surgical treatment." |
| role2: Experienced general practitioner 2 |
| "Treatment: Surgery |
| Judgment reason: The patient has a scapula fracture and needs surgical treatment." |
| role2: Experienced general practitioner 2 |
| "Treatment: Surgery |
| Judgment reason: The patient has a scapula fracture and needs surgical treatment." |
| **DyLAN** |
| Agent1: |
| Treatment: Surgery |
| Agent2: |
| Treatment: Surgery |
| Agent Importance Scores: |
| [0.5, 0.5] |
| Final Answer: |
| Treatment: Surgery |
| **MedAgent** |
| Expert gethering: Traumatology, Orthopedics, Pain Management, Imaging, Rehabilitation Medicine |
| analysis proposisition: Traumatology: The patient had limited right shoulder pain for 1 hour due to trauma, and X-ray examination showed a right scapula fracture. The patient had no history of hypertension, diabetes, coronary heart disease, infectious diseases such as hepatitis and tuberculosis, no history of drug and food allergies, and no history of trauma surgery. After admission, he suffered from... changes and a history of surgery on the right shoulder joint. These lesions and surgical history may lead to further loss of joint function and need to be treated in time. |
| Orthopedics: The patient had limited right shoulder pain for 1 hour due to trauma, and X-ray examination showed a right scapula fracture. The patient had no history of hypertension, diabetes, coronary heart disease, infectious diseases such as hepatitis and tuberculosis, no history of drug and food allergies, and no history of trauma surgery. After admission, he suffered from... changes and a history of surgery on the right shoulder joint. These lesions and surgical history may lead to further loss of joint function and need to be treated in time. |
| Pain management: The patient had limited right shoulder pain and activity for 1 hour due to trauma, and X-ray examination showed a right scapula fracture. The patient had no history of hypertension, diabetes, coronary heart disease, infectious diseases such as hepatitis and tuberculosis, no history of drug and food allergies, and no history of trauma surgery. Admission... Changes and right shoulder joint surgery history. These lesions and surgical history may lead to further loss of joint function and need to be treated in time. |
| Imaging: The patient had limited right shoulder pain and activity for 1 hour due to trauma, and X-ray examination showed a right scapula fracture. The patient had no history of hypertension, diabetes, coronary heart disease, infectious diseases such as hepatitis and tuberculosis, no history of drug and food allergies, and no history of trauma surgery. Admission... Changes and right shoulder joint surgery history. These lesions and surgical history may lead to further loss of joint function and need to be treated in time. |
| Rehabilitation medicine: The patient had limited right shoulder pain and activity for 1 hour due to trauma, and X-ray examination showed a right scapula fracture. The patient had no history of hypertension, diabetes, coronary heart disease, infectious diseases such as hepatitis and tuberculosis, no history of drug and food allergies, and no history of traumatic surgery. After admission, he suffered from... and a history of surgery on the right shoulder joint. These lesions and surgical history may lead to further loss of joint function and need to be treated in time. |
| report summarization: 'Key points: The patient had right shoulder pain and limited activity for 1 hour due to trauma, and X-ray examination showed a right scapula fracture. The patient had no history of hypertension, diabetes, coronary heart disease, infectious diseases such as hepatitis and tuberculosis, no history of drug and food allergies, and no history of traumatic surgery. After admission, the patient was weak in spirit and had unresolved urination and defecation. The most worrying aspect is the degenerative lesions of the bilateral shoulder joints and the history of surgery on the right shoulder joint. These lesions and surgical history may lead to further loss of joint function and need to be treated in time. \n\nTotal analysis: The patient had right shoulder pain and limited activity for 1 hour due to trauma, and X-ray examination showed a right scapula fracture. The patient has no history of hypertension, diabetes, coronary heart disease, infectious diseases such as hepatitis and tuberculosis, no history of drug and food allergies, and no history of trauma surgery. After admission, the patient was weak and had difficulty urinating and defecating. The most worrying aspect is the degenerative lesions of the bilateral shoulder joints and the surgical history of the right shoulder joint. These lesions and surgical history may lead to further loss of joint function and need to be treated in time. ' |
| Collaborative consultation: Traumatology: yes Orthopedics: yes Pain management: yes Imaging: yes Rehabilitation medicine: yes |
| Decision Making: |
| prompt: Please propose the treatment items that need to be taken for this patient based on the patient's chief complaint, past history, current medical history, physical examination results, visiting department and diagnosis results provided below. |
| final answer: Surgery |
| **MDAgent** |
| Complexity Check:intermediate |
| Expert Recruitment:   Agent 1 (🏆 1. Pulmonologist): Specializes in the diagnosis and treatment of respiratory system disorders. Agent 2 (🏆 2. Trauma Surgeon): Focuses on the diagnosis and treatment of injuries caused by trauma. Agent 3 (🏆 3. Orthopedic Surgeon): Specializes in the surgical treatment of musculoskeletal disorders and injuries. Collaborative Decision Making: Hierarchy Selection Participatory Debate (pulmonologist): antibiotic treatment (trauma surgeon): interventional treatment (orthopedic surgeon): surgery, antibiotic treatment, physical therapy |
| Analysis and Synthesis: |
| == Round 1 == ⌊_Turn 1 Agent 1 (🏆 pulmonologist): No Agent 2 (🏆 trauma surgeon): No Agent 3 (🏆 orthopedic surgeon): No |
| Final Decision: Medication |
| **MedChain-Agent** |
| Doctor group: |
| doctor1: surgery, physical therapy |
| doctor2: surgery |
| doctor3: surgery |
| Summary: surgery |
| Feedback: correct |
| Final Answer: surgery |
| Data summary: extract 12-dimensional information from the current case and add it to RAG—DataSet |
| **Ground Truth:** |
| Surgery |

Figure 27: **The case study among different multi-agent for task5.**

| | |
|---|---|
| **Task 5 - Input:** | |
| 患者综合报告如下: | |
| "病人主诉: 外伤致右侧肩部疼痛活动受限1小时。 | |
| 详细情况: 患者于1小时前干活时不慎摔倒在地,背部着地,当时感右侧肩部疼痛难忍,右侧肩部及右上肢活动受限,当时无头痛、胸痛、腹痛,无昏迷,入我院就诊,行x片检查示: 右侧肩胛骨骨折,拟右侧肩胛骨骨折收入住院,入院来,患者精神软,大小便未解。" | |
| 仅回复一个或多个治疗项目的名称,不要包含其他任何解释,并确保回答在上述十一项之中!!! | |
| 若有多个选项,则用逗号隔开。 | |
| 输出示例: | |
| "治疗项目: 中医治疗,基因治疗,手术 | |
| 判断理由: 因为......" | |

**AutoGen**

role1: 经验丰富的全科医生1
"治疗项目: 手术
判断理由: 患者有肩胛骨骨折,需要手术治疗。"
role2: 经验丰富的全科医生2
"治疗项目: 手术
判断理由: 患者有肩胛骨骨折,需要手术治疗。"
role2: 经验丰富的全科医生2
"治疗项目: 手术
判断理由: 患者有肩胛骨骨折,需要手术治疗。"

**DyLAN**

Agent1:
治疗项目: 手术
Agent2:
治疗项目: 手术
Agent Importance Scores:
[0.5, 0.5]
Final Answer:
治疗项目: 手术

**MedAgent**

expert gethering: 创伤骨科、骨科、疼痛管理、影像学、康复医学
analysis proposistion:创伤骨科: 患者因外伤致右侧肩部疼痛活动受限1小时,X片检查示右侧肩胛骨骨折。患者无高血压、糖尿病、冠心病等病史,无肝炎、肺结核等传染病史,无药物及食物过敏史、无外伤手术史。入院后患...变和右侧肩关节的手术史。这些病变和手术史可能导致进一步的关节功能丧失,需要及时处理。
骨科: 患者因外伤致右侧肩部疼痛活动受限1小时,X片检查示右侧肩胛骨骨折。患者无高血压、糖尿病、冠心病等病史,无肝炎、肺结核等传染病史,无药物及食物过敏史、无外伤手术史。入院后患...变和右侧肩关节的手术史。这些病变和手术史可能导致进一步的关节功能丧失,需要及时处理。
疼痛管理: 患者因外伤致右侧肩部疼痛活动受限1小时,X片检查示右侧肩胛骨骨折。患者既往无高血压、糖尿病、冠心病等病史,无肝炎、肺结核等传染病史,无药物及食物过敏史、无外伤手术史。入院...变和右侧肩关节的手术史。这些病变和手术史可能导致进一步的关节功能丧失,需要及时处理。
影像学: 患者无高血压、糖尿病、冠心病等病史,无肝炎、肺结核等传染病史,无药物及食物过敏史、无外伤手术史。入院后患...变和右侧肩关节的手术史。这些病变和手术史可能导致进一步的关节功能丧失,需要及时处理。
康复医学: 患者因外伤致右侧肩部疼痛活动受限1小时,X片检查示右侧肩胛骨骨折。患者无高血压、糖尿病、冠心病等病史,无肝炎、肺结核等传染病史,无药物及食物过敏史、无外伤手术史。入院后患...变和右侧肩关节的手术史。这些病变和手术史可能导致进一步的关节功能丧失,需要及时处理。
report summarization: '关键点: 患者因外伤致右侧肩部疼痛活动受限1小时,X片检查示右侧肩胛骨骨折。患者无高血压、糖尿病、冠心病等病史,无肝炎、肺结核等传染病史,无药物及食物过敏史、无外伤手术史。入院后患者精神软,大小便未解。最令人担忧的方面是双侧肩关节的退行性病变和右侧肩关节的手术史。这些病变和手术史可能导致进一步的关节功能丧失,需要及时处理。\n\n总分析:患者因外伤致右侧肩部疼痛活动受限1小时,X片检查示右侧肩胛骨骨折。患者无高血压、糖尿病、冠心病等病史,无肝炎、肺结核等传染病史,无药物及食物过敏史、无外伤手术史。入院后患者精神软,大小便未解。最令人担忧的方面是双侧肩关节的退行性病变和右侧肩关节的手术史。这些病变和手术史可能导致进一步的关节功能丧失,需要及时处理。'
Collaborative consultation: 创伤骨科: yes 骨科: yes 疼痛管理: yes 影像学: yes 康复医学: yes
Decision Making:
prompt: 请根据以下提供的病人的主诉、既往史、现病史、查体结果、就诊科室以及诊断结果,提出需要对该病人采取的治疗项目。
final answer: 手术

**MDAgent**

Complexity Check: intermediate
Expert Recruitment:
Agent 1 (🏆 1. Pulmonologist): Specializes in the diagnosis and treatment of respiratory system disorders.
Agent 2 (🏆 2. Trauma Surgeon): Focuses on the diagnosis and treatment of injuries caused by trauma.
Agent 3 (🏆 3. Orthopedic Surgeon): Specializes in the surgical treatment of musculoskeletal disorders and injuries.
Collaborative Decision Making:
Hierarchy Selection
Participatory Debate
(pulmonologist): 抗生素治疗 (trauma surgeon): 介入治疗 (orthopedic surgeon): 手术,,抗生素治疗,物理疗法
Analysis and Synthesis:
== Round 1 ==
|_Turn 1
Agent 1 (🏆 pulmonologist): No
Agent 2 (🏆 trauma surgeon): No
Agent 3 (🏆 orthopedic surgeon): No
Final Decision: 药物治疗

**MedChain-Agent**

Doctor group:
doctor1: 手术、物理疗法
doctor2: 手术
doctor3: 手术
Summary: 手术
Feedback: 正确
Final Answer: 手术
Data summary: 把当前病例的情况提取出12维信息,加入RAG—DataSet

**Ground Truth:**

手术

Figure 28: **The case study among different multi-agent for task5 in Chinese.**

Task1
作为一名专业的分诊医生，你需要根据病人的主诉，将其分诊至适当的科室。分诊流程分为两步：
1．首先将病人分诊至唯一的第一级科室。
2．然后将病人分诊至一个或多个对应的第二级科室。
请仅回复将病人分诊的第一级科室和第二级科室的名称，不要包含其他任何解释。科室必须从以下列表中选择：
护理科：基础护理、内科护理、外科护理
药剂科：西药房、药理实验室、中药房、调剂科
口腔科：口腔粘膜科、牙周科、儿童口腔科、口腔修复科、牙体牙髓科、种植科、正畸科、口腔急诊科、口腔预防科、颌面外科
儿科：小儿感染科、儿童保健科、小儿耳鼻喉、小儿免疫科、小儿血液科、小儿肾内科、小儿内科、小儿心外科、小儿呼吸科、小儿骨科、小儿内分泌科、小儿整形科、小儿消化科、小儿泌尿科、小儿精神科、儿童康复科、小儿神经外科、小儿外科、小儿心内科、小儿皮肤科、新生儿科、小儿神经内科、小儿急诊科
医学影像科：MRI室、CT室、B超科、X线室、彩超科、放射科、心超科、超声科、核医学科
眼科：眼外伤、青光眼、眼眶及肿瘤、眼视光学、角膜科、白内障、小儿眼科、眼底
检验科：体格检验、临床检验室、血液检验、生化室、病理科
外科：胰腺外科、创伤骨科、脊柱外科、外伤科、麻醉疼痛科、骨肿瘤科、泌尿外科、胃肠外科、神经外科、心脏外科、器官移植、普外科、乳腺外科、关节骨科、骨科、血管外科、肝胆外科、手外科、甲状腺外科、心胸外科、整形科、烧伤科、胸外科、肛肠外科、微创外科
皮肤性病科：激光室、性病科、皮肤美容、皮肤科
精神科：精神科
全科：全科
耳鼻咽喉科：耳鼻咽喉科
内科：肾脏内科、结核病科、肝病科、营养科、消化内科、内分泌科、神经内科、传染科、过敏反应科、干部诊疗科、呼吸科、免疫科、心血管内科、血液科、老年病科、普通内科、艾滋病科、风湿科、感染科
急诊科：急诊科
肿瘤科：肿瘤妇科、肿瘤外科、放疗科、肿瘤内科
中医科：中医肛肠科、中医消化科、中医男科、推拿科、中医肿瘤科、中医肾脏内科、中医外科、中医感染内科、中医皮肤科、中医妇产科、中医内分泌、中医心内科、中西医结合科、中医血液科、中医骨伤科、中医免疫内科、中医肝病科、中医神经内科、中医按摩科、中医儿科、中医老年病科、中医五官科、中医精神科、中医呼吸科、中医内科、针灸科
康复科：理疗科、康复科
妇产科：妇科内分泌、生殖中心、产前检查科、妇科肿瘤、计划生育科、妇泌尿科、高危产科、产科、妇科
心理科：心理科
病人主诉：{pation_contribution}

Task2
你是一个医生。已知病人的主诉如下：
{chief_complaints}
你需要通过与病人对话来获取更多信息。体格检查包括：一般检查（包括身高、体重、体温、血压、脉搏等）、头颅眼耳鼻喉检查、颈部检查（包括甲状腺、颈部淋巴结）、胸部检查（包括肺部、心脏）、腹部检查、脊柱和四肢检查、皮肤检查、神经系统检查、泌尿生殖系统检查。辅助检查包括：X-ray、MRI、CT、超声、核医学成像、血液学检查、尿液检查、粪便检查、内镜检查、病理检查。在获取一定信息后（如既往史、现病史），请你根据病人情况选择要询问的体格检查和辅助检查情况，每轮对话只问一项或两项，请尽可能多的询问检查项目（体格检查和辅助检查每个至少询问一项），直至可以判定病情。最后结束对话时请说"祝您早日康复"。

Task3
请你扮演一名放射科医生，你将接收到一张或多张同一模态的医学影像，这些影像均来自同一位患者。请你尝试对这些影像进行分析，并提供一个简短的综合性影像报告。请勿包含其他任何解释。影像报告：

Task4
作为一名医生，请根据以下提供的病人的主诉、既往史、现病史、查体结果以及就诊科室，作出对该病人的诊断。如果无法作出诊断，请仅回复"None"。请勿包含其他任何解释。
  - 主诉：{zhusu}
  - 既往史：{jiwangshi}
  - 现病史：{xianbingshi}
  - 查体结果：{chati}
  - 就诊科室：{keshi}

  仅回复一个或多个诊断结果的名称，不要包含其他任何解释。

Task5
作为一名医生，请根据以下提供的病人的主诉、既往史、现病史、查体结果、就诊科室以及诊断结果，提出需要对该病人采取的治疗项目。如果无法提出治疗项目，请仅回复"None"。请勿包含其他任何解释。治疗项目必须从以下十一项中选择：
  1．手术
  2．介入治疗
  3．药物治疗
  4．化学治疗
  5．抗生素治疗
  6．放射治疗
  7．物理疗法
  8．免疫疗法
  9．心理治疗
  10．中医治疗
  11．基因治疗

  - 主诉：{zhusu}
  - 既往史：{jiwangshi}
  - 现病史：{xianbingshi}
  - 查体结果：{chati}
  - 就诊科室：{keshi}
  - 诊断结果：{jieguo}

  仅回复一个或多个治疗项目的名称，不要包含其他任何解释，并确保回答在上述十一项之中。

Figure 29: **The prompt of task 1 to 5 for other baseline.**

