# OpenReview forum: "MedChain: Bridging the Gap Between LLM Agents and Clinical Practice with Interactive Sequence"
_NeurIPS.cc/2025/Datasets_and_Benchmarks_Track — NeurIPS 2025 Datasets and Benchmarks Track spotlight_

### Official Review · Reviewer_rq5Y · 2025-06-15

**Rating:** 5
**Confidence:** 5

**Summary:**

MedChain is a new benchmark dataset for evaluating LLM-based agents in clinical decision-making (CDM), designed to reflect real-world medical practice with features like personalization, interactivity, and sequentiality. It comprises 12,163 clinical cases across 19 specialties, including medical images. The proposed MedChain-Agent, an AI system with a feedback mechanism and MedCase-RAG module, demonstrates superior performance in handling these complex, sequential clinical tasks

**Additional Feedback:**

NA

**Dataset Code Accessibility:**

Yes

**Dataset Code Comments:**

NA

**Ethical Considerations:**

No, there are no or only very minor ethics concerns

**Final Justification:**

The author has acknowledged an oversight in the paper that caused my misunderstanding and has promised to make a correction. I believe the revised paper deserves a higher score.

**Limitations Weaknesses:**

The main weaknesses of this paper are also quite prominent, primarily concerning two points.

First, the novelty of this research is limited. The dataset used in this study was actually scraped from the internet (from the website iiyi.com), and previous work (e.g., AI Doctor) has also utilized data from this site. The main difference is that the current study collected a more comprehensive dataset (12,000 cases compared to the previous 500). Furthermore, the approach of decomposing clinical work into distinct tasks has been explored in prior research, such as arxiv 2405.07960 and earlier work by Fei-Fei Li's group. However, none of these previous studies released their datasets publicly.

Second, there is a discrepancy between the paper's claims and its actual contributions. As I understand it, the term "Interactive Sequence" in the title is highly misleading. The study does not actually build a multi-turn interactive environment for AI and patients (unlike systems such as Google AMIE, AgentClinic, or CRAFT-MD). Instead, the authors have constructed five single-turn question-answering benchmarks. Consequently, there is little that is truly "interactive" here. The authors seem to have overlooked the most critical aspect of an interactive consultation: the dynamic gathering of information through multi-turn interaction when information is incomplete.

**Strengths Contributions:**

The most significant contribution of this paper is the introduction of a large-scale, multimodal EMR text dataset. Compared to commonly used datasets like MIMIC-IV, its key advantage lies in being less biased towards critical care and including a diverse range of imaging results, even encompassing nuclear medicine scans such as MRI and PET.

This dataset is of crucial importance for the development of advanced medical LLM models today. It is worth noting that current medical LLMs are almost exclusively developed based on medical literature, with very few models being evaluated and trained on real-world clinical data. Even for the few that have used such data (like Google's AMIE and MedFound), the datasets have not been made publicly available. To my knowledge, this may be one of the largest publicly available clinical text datasets from real patients that covers a broad spectrum of diseases.

---

> ### Author Rebuttal · Authors · 2025-07-30
>
> **Q1 Novelty Clarification**
>
> A1 1) The aforementioned works only cover partial medical specialties. For instance, AI Hospital encompasses only 6 specialties, while our work covers 19 departments. 2) We decompose sequential decision-making in greater detail, with ground truth established for each stage to verify results. In contrast, AgentClinic (arxiv 2405.07960) only evaluates whether the final diagnosis is correct, without assessing intermediate stages.
>
> **Q2 Interactive Part**
>
> A2 I believe there may be a misunderstanding about our work. We respectfully clarify that our work does include this functionality. As documented in Section 3.3 and Appendix A.2.2, our system employs a patient agent that enables dynamic information gathering through multi-turn interactions, consistent with the approach you mentioned in your comment.

---

> ### Comment · Reviewer_rq5Y · 2025-08-01
>
> I do not believe I have misinterpreted your work.
>
> While you do briefly mention in Section 3.3 and the appendix that you constructed an interactive environment, this description is overly simplistic, especially when compared to similar studies. It fails to provide the necessary details for replication or full understanding. For instance:
>
> What prompts are used in patient simulator?
>
> How is the consultation terminated? Is there a maximum number of interaction rounds, or is a verifier module involved?
>
> For a manuscript with "interactive" as a key term in its title, the near-complete absence of specific details regarding the design of the interactive environment is a significant shortcoming.
>
> Moreover, I have thoroughly examined all the files in the provided GitHub repository and have been unable to locate any code related to the implementation of this interactive environment. If this is an oversight on my part, I would appreciate it if you could point me to the exact files containing the code for the patient simulator, the doctor simulator, and the overall interactive environment.

---

> > ### Author Response · Authors · 2025-08-01
> >
> > Thanks for the further inquiries. We acknowledge that we overlooked certain details regarding the interactive environment. We will respond to each of the specific points you mentioned.
> > 1. Patient simulator: Gemma2 (9b) (as described in line 157)
> > 2. Prompt for Patient:
> >
> > *You are to role-play as a Standardized Patient. Here is your case information:*
> >
> > *{patient_information}*
> >
> > *You need to answer the doctor's questions directly based on the case information (do not fabricate doctor-patient dialogues). Note that unless the doctor explicitly asks about physical examination and auxiliary examination findings, please do not proactively mention or inquire about physical examination and auxiliary examination related content. If the doctor asks about content that does not exist in the case, please indicate that you don't know and avoid fabricating information. At all times, remember that you are only playing the role of a standardized patient.*
> >
> > Note: {patient information} includes Chief Complaint, Past History, History of Present Illness, Physical Examination and Auxiliary Examinations.
> >
> > 3. Doctor simulator: Each tested LLM serves as a doctor simulator.
> >
> > 4. Prompt for Doctor:
> >
> > *You are a doctor. The patient's chief complaint is as follows:*
> >
> > *{chief_complaints}*
> >
> > *You need to gather more information through conversation with the patient. Physical examination includes: general examination (including height, weight, temperature, blood pressure, pulse, etc.), head, eyes, ears, nose, and throat examination, neck examination (including thyroid, cervical lymph nodes), chest examination (including lungs, heart), abdominal examination, spine and limb examination, skin examination, neurological examination, and genitourinary system examination. Auxiliary examinations include: X-ray, MRI, CT, ultrasound, nuclear medicine imaging, hematological tests, urine tests, stool tests, endoscopic examination, and pathological examination. After obtaining certain information (such as past history and present illness history), please select the physical examinations and auxiliary examinations to inquire about based on the patient's condition. Ask about only one or two items per conversation round. Please inquire about as many examination items as possible (ask at least one item each for physical examination and auxiliary examination) until you can determine the condition. At the end of the conversation, please say "Wishing you a speedy recovery."*
> >
> > 5. GitHub Code: We apologize for the oversight regarding this component. The relevant code has been uploaded to the doctor_patient_interaction directory in our GitHub repository. We are very grateful for your meticulous review and for bringing these issues to our attention!
> >
> > These details will be incorporated into Section 3.3 and Appendix A.2.2 in the revised manuscript to ensure comprehensive coverage. We greatly appreciate your careful and thorough review. If you have any additional concerns or questions about this work, we welcome the opportunity to address them.

---

### Official Review · Reviewer_qbgW · 2025-07-02

**Rating:** 4
**Confidence:** 3

**Summary:**

This paper introduces MedChain, a new benchmark dataset designed to evaluate the clinical decision-making capabilities of Large Language Model (LLM)-based agents in a way that more accurately reflects real-world medical practice. The dataset contains 12,163 clinical cases across 19 medical specialties and includes 7,338 medical images. MedChain is built around three core features often missing in other benchmarks: (1) personalization (using patient-specific details), interactivity (requiring agents to actively gather information), and sequentiality where decisions in one stage impact subsequent stages). (2) To tackle the challenges presented by this new benchmark, the authors also propose the MedChain-Agent. This is a multi-agent AI system that simulates a collaborative clinical environment with specialized agents for different tasks, a summarizing agent to consolidate findings, and a feedback agent for iterative refinement. A key component of this framework is the MedCase-RAG module, a Retrieval-Augmented Generation system that retrieves similar past cases from a dynamically expanding knowledge base to inform current decision-making. Experiments show that MedChain-Agent significantly outperforms existing single-agent and multi-agent systems on the MedChain benchmark and also shows strong performance on other existing medical datasets.

**Dataset Code Accessibility:**

Yes

**Ethical Considerations:**

No, there are no or only very minor ethics concerns

**Final Justification:**

Thank you for the response. All my previous concerns have been solved point-by-point and I would like to keep my overall positive score.

**Limitations Weaknesses:**

1. Although MedChain provides a valuable resource, the novelty of MedChain-agent is potentially limited. The RAG in MedChain-agent is very similar to long-term memory in LLM agents.
2. Since the experiments are conducted in Chinese, additional Chinese medical LLMs could be better set of baselines besides Huatuo and general LLMs.
3. Following my previous comment, (medical) reasoning models should also be considered as an important set of baselines.
4. Additional experimental details for ablation study will be helpful for readers to understand the setting of w/o seq and w/o inter.

**Strengths Contributions:**

The paper introduces a comprehensive and challenging benchmark that evaluates LLM agents on the entire clinical workflow, from specialty referral to treatment. By incorporating personalization, interactivity, and sequential decision-making, it addresses significant gaps left by existing medical benchmarks that often rely on static, multiple-choice question-answering formats.

---

> ### Author Rebuttal · Authors · 2025-07-30
>
> **Q1 Novelty Clarification**
>
> A1 The novelty of this work primarily lies in the **sequential decision-making process** in clinical scenarios, which distinguishes it from other works since decision results from each stage affect subsequent stages—a critical aspect in clinical practice. Thus, MedChain is constructed with clinical cases consisting of five phases to benchmark this capability. Apart from the mentioned RAG mechanism in weakness, we further propose another technique to sovel a problem. Since error propagation in this sequential process significantly increases the complexity of problem-solving, we propose MedChain-agent with a **feedback mechanism** that adaptively corrects potential errors from previous steps.
>
> **Q2 Additional Chinese LLMs**
>
> A2 We further compare with chinese LLMs baichuan.
> | Model | Level1 | Level2 | History-taking | Examination | Diagnosis | Treatment | Average |
> |------|--------|--------|----------------|-------------|-----------|-----------|---------|
> | Baichuan | 0.2959 | 0.0532 | 0.3264 | - | 0.4297 | 0.2591 | 0.2728 |
>
> **Q3 Reasoning Model**
>
> A3 GPT-o3, MedReason, and FineMedLM-o are further tested as baselines. Since these are all language models, examinations involving vision cannot be tested.
> | Model | Level1 | Level2 | History-taking | Examination | Diagnosis | Treatment | Average |
> |------|--------|--------|----------------|-------------|-----------|-----------|---------|
> | GPT-o3 | 0.5495 | 0.2866 | 0.3493 | - | 0.4891 | 0.3343 | 0.4017 |
> | MedReason | 0.4602 | 0.1258 | 0.3315 | - | 0.4741 | 0.2993 | 0.3381 |
> | FineMedLM-ol | 0.1502 | 0.0124 | 0.3969 | - | 0.2772 | 0.1345 | 0.1942 |
>
> **Q4 Additional Details for ablation study**
>
> A4 We will further elebrate this information in Sec 5.4 as following:
>
> *'w/o person' means all detailed patient profiles are omitted from the input, resulting in a lack of personalized information that makes correct diagnosis more challenging for the model. 'w/o seq' means we use the ground truth from the previous stage as input to the next stage, rather than using the model's previous output, which simplifies the benchmark. 'w/o inter' means we directly provide all patient examination results as input without requiring the agent to autonomously inquire about the patient's condition, also simplifying the benchmark.*

---

> > ### Comment · Reviewer_qbgW · 2025-08-01
> >
> > Thank you for the response. All my previous concerns have been solved point-by-point and I would like to keep my overall positive score.

---

### Official Review · Reviewer_VtKV · 2025-07-09

**Rating:** 6
**Confidence:** 4

**Summary:**

The manuscript introduces a 12,163-case benchmark that includes five sequential stages of clinical care (specialty referral, history-taking, examination, diagnosis, treatment) and covers 19 specialties and 156 sub-categories. The cases are sourced from a de-identified Chinese clinical repository and are annotated to emphasise personalisation, interactivity, and sequential dependency. To address the benchmark, the authors propose MedChain-Agent, a three-role multi-agent system augmented with the MedCase-RAG retrieval module and an explicit feedback loop; ablation studies demonstrate that both RAG and feedback contribute materially to performance gains. The experiments are conducted on both the new benchmark and existing QA datasets. MedChain-Agent outperforms single-agent and prior multi-agent baselines.

**Dataset Code Accessibility:**

Yes

**Ethical Considerations:**

No, there are no or only very minor ethics concerns

**Final Justification:**

Thanks for your response, which addressed all my concerns. I have increased my score to vote for acceptance.

**Limitations Weaknesses:**

1. Limited baseline comparability
- Comparative agent frameworks are all re-implemented on InternVL-2, whereas MedChain-Agent benefits from external information (additional RAG and feedback). Meanwhile, in the experiments, none of the comparative methods incorporated external information such as RAG or GraphRAG.

2. The comparison with previous works is limited.
- The paper only includes multiple-choice QA benchmarks for comparison; however, many efforts (e.g., [1][2][3]) have explored similar ideas of benchmarking LLMs in clinical practice, covering multiple clinical tasks. For example, [1] includes 11 clinical scenarios and evaluates 22 LLMs; [2] includes 14 clinical scenarios and evaluates 11 LLMs; [3] includes 11 clinical tasks and evaluates 22 LLMs. A more detailed and thorough comparison with previous benchmark works, which is missing in the current version, is strongly encouraged. More thorough and broader reviews exist, such as this Nature article (https://www.nature.com/articles/s44222-025-00279-5).

3. Several important experiments are missing.
- More recent advanced LLMs should be included in the benchmark, such as Claude-4, o1, and o3, as well as medical LLMs (e.g., MedGemma).
- The paper does not provide the statistical significance of the experiments.

4. More analysis should be provided.
- Specialty bias concern: Could you provide the exact case counts per specialty and any weighting scheme used during training? How do you address the data bias concern?
- Ablation on feedback iterations: How many refinement cycles are optimal, and what is the average number of refinement cycles for the whole benchmark and for each specialty?
- No explicit safety, hallucination, or dosage-error metrics are included. No quantitative tracking of harmful or factually incorrect recommendations is provided.

5. Minor:
- References [37] and [38] are repeated.

Refs:

[1] Towards evaluating and building versatile large language models for medicine. npj Digital Medicine, 2025.

[2] CliMedBench: A Large-Scale Chinese Benchmark for Evaluating Medical Large Language Models in Clinical Scenarios. EMNLP, 2024.

[3] Large Language Models Are Poor Clinical Decision-Makers: A Comprehensive Benchmark. EMNLP, 2024.

**Strengths Contributions:**

1. Clearly defined motivation.
2. The proposed method is sound.
3. Extensive experiments and analysis.

---

> ### Author Rebuttal · Authors · 2025-07-30
>
> **Q1 Compared methods with RAG**
>
> A1 Thanks for your suggestion. We have compared with the RAG method in single-agent part in Table 1. To further ensure fair comparison with multi-agent, we re-implement MDAgent with RAG.
> |          | Level1 | Level2 | History-taking | Examination | Diagnosis | Treatment | Average |
> |----------|--------|--------|----------------|-------------|-----------|-----------|---------|
> | MDAgent+RAG | 0.4754 | 0.2063 | 0.4412 | 0.5346 | 0.4198 | 0.4371 | 0.4190 |
>
> Compared to our MedChain-Agent's performance (Average: 0.5200), these results demonstrate that MedChain-Agent's superiority stems from our multi-agent collaboration and feedback mechanisms, not merely from external information access. Even with RAG enhancement, baseline methods still underperform our framework significantly.
>
> **Q2 Further review**
>
> A2 Thanks for your suggestion. We will incorporate more previous works for comparison as following:
>
> *Recently, several benchmarks [1,2,3] have been proposed to evaluate LLMs across diverse clinical scenarios and tasks, including information extraction, text summarization, and clinical outcome prediction. However, these benchmarks primarily consist of independent question-answer pairs, where each task is evaluated in isolation. Sequential decision-making is critical in medical practice, as each patient experiences a continuous journey from initial triage through treatment to recovery. Therefore, it is essential to evaluate how LLMs perform throughout this entire clinical pipeline. Our work distinguishes itself from existing benchmarks by focusing on sequential decision-making within interactive environments, specifically evaluating LLM performance in realistic clinical scenarios that require executing the complete patient care workflow.*
>
> [1] Towards evaluating and building versatile large language models for medicine. npj Digital Medicine, 2025.
>
> [2] CliMedBench: A Large-Scale Chinese Benchmark for Evaluating Medical Large Language Models in Clinical Scenarios. EMNLP, 2024.
>
> [3] Large Language Models Are Poor Clinical Decision-Makers: A Comprehensive Benchmark. EMNLP, 2024.
>
> **Q3 Further Important Experiments**
>
> A3
> 1) More recent advanced LLMs comparison: we compare with several recent LLMs including general LLM GPT-o3, and medical LLM MedGemma, baichuan, medreason and FineMedLM-o for comparison.
> | Model | Level1 | Level2 | History-taking | Examination | Diagnosis | Treatment | Average |
> |------|--------|--------|----------------|-------------|-----------|-----------|---------|
> | GPT-o3 | 0.5495 | 0.2866 | 0.3493 | - | 0.4891 | 0.3343 | 0.4017 |
> | medgemma | 0.5063 | 0.1975 | 0.3315 | 0.6324 | 0.4668 | 0.3558 | 0.4105 |
> | baichuan | 0.2959 | 0.0532 | 0.3264 | - | 0.4297 | 0.2591 | 0.2728 |
> | medreason | 0.4602 | 0.1258 | 0.3315 | - | 0.4741 | 0.2993 | 0.3381 |
> | FineMedLM-ol | 0.1502 | 0.0124 | 0.3969 | - | 0.2772 | 0.1345 | 0.1942 |
>
> 2) Statistical significance of the experiments: We selected the best-performing model from each category and tested the statistical significance between their results and our results. These results shown in following table demonstrate that our approach achieves statistically significant improvements over the strongest baseline models in the majority of evaluation tasks.
> |  | task1 | task2 | task3 | task4 | task5 |
> |---------|-------|-------|-------|-------|-------|
> | Claude3.5 | 0.02656 | 4.67E-32 | 5.76E-22 | 9.22E-11 | 4.31E-09 |
> | RAG | 0.05918 | 1.37E-36 | 0.8807 | 4.99E-15 | 4.87E-33 |
> | MedAgent | 5.17E-42 | 6.00E-57 | 1.20E-35 | 1.34E-27 | 4.44E-07 |
>
> **Q4 More Analysis**
>
> A4
> 1) We have analyzed the specialty statistics in Supplementary Figure 6. Since our framework is training-free and employs a multi-agent workflow to solve clinical tasks, this work does not involve any weighting scheme.
>
> 2) The feedback iteration is decided by the agent itself, with a maximum limit of 5. We analyzed the iteration numbers for each task in the benchmark, and the average number is 2.54. The detailed information is shown in the following table:
> | task1 | task2 | task3 | task4 | task5 | Average |
> |-------|-------|-------|-------|-------|---------|
> | 3.5   | 3.7   | 0.6   | 3.2   | 1.7   | 2.54    |
>
> 3) According to these hallucination studies [4, 5], hallucination can be evaluated by accuracy. Accuracy measures the proportion of correctly answered questions, with lower accuracy indicating a higher degree of hallucination in the generated responses.
>
> [4] Li, Yifan, et al. "Evaluating Object Hallucination in Large Vision-Language Models." Proceedings of the 2023 Conference on Empirical Methods in Natural Language Processing. 2023.
>
> [5] Sahoo, Pranab, et al. "A comprehensive survey of hallucination in large language, image, video and audio foundation models." Proceedings of the 2024 Conference on Empirical Methods in Natural Language Processing. 2024.
>
> **Q5 Repeated Reference**
>
> A5 Thanks for your careful proofreading. This will be revised in the future manuscript.

---

> ### Comment · Reviewer_VtKV · 2025-08-06
>
> Thanks for your further response, which addressed all my concerns. I have increased my score to vote for acceptance. Please incorporate all these responses, along with the comments from other fellow reviewers, in the revision.

---

### Note · Authors · 2025-08-13

Dear AC,

We sincerely appreciate your time and effort in coordinating our submission. We thank all reviewers for recognizing the key strengths of our work, including:

- Comprehensive sequential clinical decision-making benchmark across 19 medical specialties. [reviewers VtKV, qbgW, rq5Y]

- Novel multi-agent framework with feedback mechanism for error propagation handling. [reviewers VtKV, qbgW]

- Large-scale multimodal medical dataset with 12,163 cases and 7,338 medical images. [reviewers VtKV, rq5Y]


During the discussion period, we have addressed all reviewer concerns thoroughly:

**Enhanced Baseline Comparisons (raised by VtKV and qbgW)**

- Added RAG-enhanced multi-agent baselines (MDAgent+RAG) showing our superiority stems from multi-agent collaboration, not just external information
- Included recent LLMs (GPT-o3, MedGemma, Baichuan, MedReason, FineMedLM-o)
- Provided statistical significance tests demonstrating significant improvements (p<0.05) over strongest baselines
- We appreciate Reviewer VtKV's positive feedback and score increase to acceptance

**Novelty Clarification (raised by rq5Y and qbgW):**

- Clarified that our novelty lies in the sequential decision-making process where errors propagate across stages—a critical but overlooked aspect in clinical practice
- Demonstrated broader coverage (19 departments vs. 6 in previous work) and more detailed decomposition with ground truth for each stage
- Introduced feedback mechanism to adaptively correct errors from previous steps, addressing the unique challenge of error propagation in sequential tasks

**Additional Analysis (raised by VtKV and qbgW):**

- Analyzed feedback iterations (average 2.54 rounds across tasks)
- Clarified ablation study settings (w/o seq, w/o inter, w/o person)
- Addressed specialty bias concerns and hallucination evaluation metrics

**Clarified Interactive Environment (raised by rq5Y):**

- Provided detailed patient/doctor simulator prompts and termination criteria
- Uploaded missing interactive environment code to GitHub (doctor_patient_interaction directory)
- Clarified multi-turn interaction implementation details in Section 3.3 and Appendix A.2.2

We commit to incorporating all reviewer feedback into our revised manuscript. Through this rebuttal phase, all reviewers expressed satisfaction with the responses, with one reviewer explicitly mentioning increasing their score to acceptance.

Thank you again for your valuable feedback.

Authors

---

### Decision · Program_Chairs · 2025-09-18

**Decision:**

Accept (spotlight)

**Comment:**

This is a substantial contribution, a large-scale benchmark reflecting sequential clinical workflows,  a well-performing multi-agent baseline. While the claims of interactivity are somewhat overstated, and broader comparisons  the dataset’s scale, accessibility, and practical relevance outweigh these weaknesses.